**EMBO** *reports*

# Gim3 buffers and potentiates de novo mutations that affect fluconazole susceptibility in yeast

Mohammed T Tawfeeq [1,2], Dimitrios Konstantinidis [1,2], Ana Lucia Rocha Iraizos [1,2,5,6,7,8], Wouter Van Genechten [3], Jolien Vreys [3], Lieselotte Vermeersch [1,2], Karin Voordeckers [1,2], Patrick Van Dijck [3,4 ✉] & Kevin J Verstrepen [1,2 ✉]

## Abstract

Gim3 is an evolutionarily conserved component of the prefoldin chaperone complex, involved in protein folding. We previously found that *GIM3* genetically interacts with many de novo mutations in *Saccharomyces cerevisiae*. Removing *GIM3* from mutagenized *S. cerevisiae* cells significantly affected the fitness effect of mutations. This indicates that Gim3 might change the evolutionary impact of de novo mutations by either buffering (hiding) or potentiating (increasing) their phenotypic effects, depending on the environmental or genetic context. Here, we investigated Gim3's role in shaping the evolutionary fate of de novo mutations under fluconazole stress, an antifungal drug used to combat fungal infections. Applying both strong and moderate fluconazole stress in the presence or absence of *GIM3* revealed that Gim3 potentiates fluconazole susceptibility (resistance and tolerance) by enabling mutations to have immediate phenotypic effects. Deleting *GIM3* reduced growth in fluconazole in most mutants, indicating that *GIM3* could be a promising target for new antifungal therapies against drug-resistant infections. Importantly, Gim3 also modulates fluconazole susceptibility of the fungal pathogen *Nakaseomyces glabratus*, further highlighting Gim3's role in fluconazole resistance and tolerance.

**Keywords** Gim3; Evolvability; Mutational Robustness; Drug Susceptibility
**Subject Categories** Evolution & Ecology; Microbiology, Virology & Host Pathogen Interaction

## Introduction

One of the main conundrums in biology is the delicate balancing act between (the accumulation of) detrimental and beneficial mutations (Lenski et al, 2006). Organisms need to protect themselves from harmful mutations, yet they also rely on mutations to evolve and adapt to changing environments and stress factors (Masel and Trotter, 2010; Yaakov et al, 2017). Interestingly, the phenotypic outcome of a mutation does not only depend on the mutation itself but also on how it interacts with an individual's genome (genetic interactions) and the environment (environmental interactions). Such interactions add a layer of complexity that makes it much more difficult to unravel if and how specific mutations lead to a phenotype or disease (Mani et al, 2008).

Previous studies have uncovered a special case of genetic interaction where specific genes, termed "buffer genes", influence the phenotypic effects of many mutations or alleles across the genome (Queitsch et al, 2002; Jarosz and Lindquist, 2010; Rutherford and Lindquist, 1998). Reducing the activity of these buffer genes reveals "cryptic genetic variation", i.e., genetic variation with little or no observable phenotypic effect but which can become phenotypically active under genetic or environmental perturbations. This genetic interaction between buffer genes and cryptic genetic variation thus provides "genetic buffering", where the activity of a buffer gene reduces the impact of some changes elsewhere in the genome (Masel, 2006; Jarosz and Lindquist, 2010; Rutherford and Lindquist, 1998). Genetic buffering allows organisms and populations to accumulate cryptic genetic variation over time without experiencing a significant fitness effect. Importantly, the accumulated cryptic variation might serve as raw material for evolution because certain environmental or genetic perturbations may reduce buffering activity and thus "release" the fitness effects associated with the genetic variants, with some variations possibly proving adaptive in the new conditions (Rutherford, 2000; Barkai and Shilo, 2007).

A key criticism of previous genetic buffering studies is that they have mostly focused on pre-existing (standing) genetic variation. It has been argued that the revelation of cryptic genetic variation upon inhibition of a putative buffer gene should not be taken as evidence for mutational robustness against all types of mutations (Geiler-Samerotte et al, 2016). Instead, such cryptic genetic variation could represent genetic variation that survived the filter of natural selection to exist in its cryptic form, favoring variation

[1]VIB Laboratory for Systems Biology, VIB-KU Leuven Center for Microbiology, Leuven 3001, Belgium. [2]Laboratory for Genetics and Genomics, Center of Microbial and Plant Genetics, Department M2S, KU Leuven, Leuven 3001, Belgium. [3]Laboratory of Molecular Cell Biology, Institute of Botany and Microbiology, KU Leuven, Leuven 3001, Belgium. [4]KU Leuven One Health Institute, 3000 Leuven, Belgium. [5]Present address: VIB-KU Leuven Center for Cancer Biology, 3000 Leuven, Belgium. [6]Present address: Laboratory of Experimental Hematology, Department of Oncology, KU Leuven, Leuven 3000, Belgium. [7]Present address: Laboratory of Molecular Biology of Leukemia, Department of Human Genetics, KU Leuven, Leuven 3000, Belgium. [8]Present address: Leuvens Kanker Instituut (LKI), KU Leuven - UZ Leuven, Leuven 3000, Belgium.
✉E-mail: patrick.vandijck@kuleuven.be; kevin.verstrepen@kuleuven.be

that is near-neutral, possibly because of interactions with specific genes. However, it is unclear whether similar interactions would exist for the majority of de novo mutations; mutations that are not "buffered" would likely be eliminated through negative selection, leaving an overrepresentation of the rare variants that are buffered. This could create the misleading impression that most genetic variation is buffered and that buffer genes broadly reduce the fitness effects of genetic variation (Hermisson and Wagner, 2004; Geiler-Samerotte et al, 2016). The very existence of buffer genes therefore remains somewhat contested, especially since the concept and definition of a buffer gene is not always clear and consistent across studies. We have therefore previously proposed a unifying definition for buffer genes as genes whose activity, in one way or another, contributes to mutational robustness by influencing the phenotypic outcome of an unusually high proportion of standing genetic variation and/or de novo mutations (Tawfeeq et al, 2024).

Apart from focusing on standing variation, the majority of buffering studies focus on a single gene, *HSP90*. The chaperone Hsp90 suppresses the effects of mutations in various organisms, including *Saccharomyces cerevisiae* (budding yeast), *Arabidopsis thaliana* (thale cress), *Drosophila melanogaster* (fruit fly), *Caenorhabditis elegans* (worm) and *Astyanax mexicanus* (cave fish) (Lempe et al, 2013; Burga et al, 2011; Rutherford, 2003; Jarosz, 2016; Rohner et al, 2013; Queitsch et al, 2002; Casanueva et al, 2012; Chen and Wagner, 2012; Yeyati et al, 2007; Condic et al, 2024). This buffering activity is believed to primarily rely on Hsp90's capacity to ensure correct folding of mutated proteins, thereby preventing loss-of-function and/or harmful intracellular protein aggregation (Rutherford and Lindquist, 1998; Lempe et al, 2013; Jarosz and Lindquist, 2010; Burga et al, 2011). In addition, Hsp90 indirectly interacts with mutations in *cis*-regulatory regions through its direct clients and therefore could modulate gene expression in response to stress stimuli, providing buffering via a different mechanism (Jakobson et al, 2023).

Besides buffering mutations, Hsp90 can also potentiate new mutations. Here, Hsp90's activity influences the phenotypic effect of mutations in such a way that they have a beneficial effect (rather than neutral or deleterious). The potentiating effect of Hsp90 has been most intensively studied for its role in fungal resistance to fluconazole (Cowen and Lindquist, 2005). Fluconazole inhibits the biosynthesis of ergosterol, a critical component of fungal cell membranes. Resistance to fluconazole can arise through various mechanisms, such as alterations to the drug's target site, upregulation of efflux pumps, changes in sterol biosynthesis, and modifications in membrane composition (Robbins et al, 2017). In a seminal study by Cowen and Lindquist, fluconazole-resistant strains were selected by exposure to high concentrations of fluconazole (a so-called rapid selection regime) (Cowen and Lindquist, 2005). When Hsp90 expression was reduced, these resistant strains again became fluconazole-sensitive. Interestingly, all resistant strains contained mutations in *ERG3*. Fluconazole inhibits Erg11, leading to toxic intermediates in ergosterol biosynthesis, but mutations in the C5 sterol desaturase Erg3 (as well as *erg3Δ*) prevent the accumulation of these toxic intermediates, allowing growth in the presence of fluconazole (Watson et al, 1989). Since different modes of selection lead to different routes of resistance, researchers also investigated fluconazole-resistant mutations that were selected under exposure to gradually increasing concentrations of fluconazole (Anderson et al, 2003). Strains subjected to this selection regime developed resistance through mutations in the transcription factor Pdr1, which upregulated

the expression of the multidrug transporter Pdr5. Notably, this resistance was independent of Hsp90 (Cowen and Lindquist, 2005).

Since current research on buffer genes has largely centered on one gene, Hsp90, and its capacity to affect the phenotypic effect of pre-existing genetic variation (Geiler-Samerotte et al, 2016; Richardson et al, 2013; Cowen and Lindquist, 2005; Burga et al, 2011; Tokuriki and Tawfik, 2009), we currently do not know if and how unique Hsp90 is in its capacity to buffer and potentiate genetic variation (Tawfeeq et al, 2024). The main reason for this gap in our understanding is that until recently, very few candidate buffer genes, apart from Hsp90, had been identified (Tawfeeq et al, 2024; Richardson et al, 2013; Lemus et al, 2023). A comprehensive screen for buffer genes requires studying the interaction between the activity of all genes in an organism and a very large number of mutations. Our group recently performed such a genome-wide screen, resulting in the identification of a set of genes that confer mutational robustness to de novo mutations (Frickel et al, 2024). These genes mainly belong to two functional categories/gene ontologies (GO): unfolded protein binding and chromatin modification.

To further investigate if and how other buffer genes apart from *HSP90* could influence the phenotypic effects of mutations, we here focus on one newly identified buffer gene, *GIM3*, which emerged as a top candidate in our genome-wide screen, showing some of the strongest buffering activity of the 4500 genes that were tested (Frickel et al, 2024). *GIM3* encodes a subunit of the prefoldin complex (Comyn et al, 2016), and its main physiological function is to assist in the proper folding of cytoskeletal proteins like actin and tubulin (Geissler et al, 1998; Millán-Zambrano et al, 2013; Comyn et al, 2016). Recent studies have uncovered additional roles of specific prefoldin subunits in cellular processes like proteasome assembly and mitochondrial function, extending their known functions beyond protein folding (Shahmoradi Ghahe et al, 2024; Tahmaz et al, 2023). In addition, Gim3 is also hypothesized to operate as a holdase, a specialized type of chaperone that stabilizes misfolded mutated proteins in an ATP-independent manner, preventing their aggregation and maintaining them in a soluble state suitable for refolding or degradation (Comyn et al, 2016; Tahmaz et al, 2022). Moreover, genome-wide genetic interaction screens have identified numerous genes that, when deleted alongside *GIM3*, result in synthetic lethality (Tong et al, 2004). These findings suggest that Gim3 plays a role in maintaining protein homeostasis, particularly in response to mutations.

Here, we assessed the role of the newly identified buffer gene *GIM3* in enabling new mutations to have phenotypic effects, thereby facilitating the acquisition of new traits in the model organism *S. cerevisiae*. We specifically focused on antifungal resistance and tolerance, traits of significant economic and biomedical importance (Lee et al, 2023). Given the limited number of clinically useful antifungal drugs and the emergence of resistance to all of them, understanding fungal drug resistance is crucial for developing more effective treatments. We selected fluconazole as the antifungal agent for our study because it is one of the most widely used antifungal drugs. Azole drugs are favored as a first line of treatment due to their accessibility, low toxicity to humans, and a broad spectrum of action against various fungal pathogens (Lee et al, 2023; Cowen and Lindquist, 2005). In addition, studies have suggested *GIM3* interacts with different fluconazole resistance-related genes (e.g., *ERG2*, *ERG3*, *ERG6*, *CNA1*)(Collins et al, 2007; Costanzo et al, 2016; Schuldiner et al, 2005; Hoppins et al, 2011; Gong et al, 2009).

In this study, we used EMS (Ethyl methanesulfonate) to chemically create new, random mutations in cells with different

levels of *GIM3* expression (knockout, wild-type, and overexpression) and exposed them to a high concentration of fluconazole to select mutants. We found that cells with wild-type or high levels of *GIM3* expression had more cells growing on fluconazole compared to *GIM3* knockout cells, suggesting that Gim3 is important for potentiating the phenotypic effect of mutations. Interestingly, most mutants (>80%) exhibited *GIM3*-dependent growth on fluconazole. Moreover, several strains that depended on Gim3 for growth on fluconazole contained mutations in the *ERG3* gene, suggesting that Gim3 might contribute to stabilizing these mutant proteins. Furthermore, we showed that Gim3-dependent fluconazole resistance also extends to strains that evolved resistance under constant moderate fluconazole stress achieved through adaptive laboratory evolution (ALE) (Dragosits and Mattanovich, 2013).

Apart from resistance, which is defined (in research labs) as having a higher minimum inhibitory concentration (MIC) for a drug than a control strain (Berman and Krysan, 2020); we also observed signs of fluconazole tolerance, particularly in the form of trailing growth beyond the defined MIC. Drug tolerance refers to the ability of a subpopulation of fungal cells to survive and slowly grow in the presence of antifungal concentrations that exceed the MIC, without a corresponding increase in MIC values (Rosenberg et al, 2018; Delarze and Sanglard, 2015). Note that the ability of cells to continue to divide, even slowly, in the presence of a drug (tolerance) may be an important contributor to the ability to acquire drug resistance (Berman and Krysan, 2020). While our study was not specifically designed to quantify tolerance using standardized protocols, our observations suggest that *GIM3* might be involved in both resistance and tolerance mechanisms. In addition, Gim3's role extends beyond fluconazole susceptibility, as it also influences cross-resistance to caspofungin, highlighting its broader role in antifungal resistance mechanisms.

Finally, we show that Gim3 also affects fluconazole susceptibility in the fungal pathogen *N. glabratus* (formerly named *Candida glabrata*), which diverged from *S. cerevisiae* approximately 150 million years ago (Katsipoulaki et al, 2024; Sugino and Innan, 2005; Van Ende et al, 2019; Ksiezopolska et al, 2024). *N. glabratus* is the second leading cause of candidemia, a serious bloodstream infection, and is classified as a high-priority pathogen (WHO, 2022). It is particularly known for its ability to overcome azole antifungal treatments (Whaley and Rogers, 2016; Galocha et al, 2022; Burki, 2023; Ksiezopolska et al, 2024; Tsai et al, 2006). Our results demonstrating a key role of Gim3 in mediating fluconazole susceptibility across multiple species may thus open novel routes toward combating drug-resistant fungal infections.

## Results

### Gim3 facilitates the immediate phenotypic consequences of new mutations

To investigate if and how *GIM3* affects the impact of de novo mutations on fluconazole resistance, we created a set of strains with varying levels of *GIM3* expression. Specifically, we deleted *GIM3* from the genome and then reintroduced it under one of two different promoters: (i) *GIM3* under its native promoter or (ii) *GIM3* under the strong constitutive *TDH3* promoter. In both cases, *GIM3* was introduced on a plasmid containing the *URA3* selection

marker, allowing for counterselection against the plasmid using FOA (5-fluoroorotic acid), resulting in the loss of the plasmid and thus also loss of *GIM3*. As a control, we also created a strain that retained *GIM3* in the genome and introduced the same plasmid but without *GIM3*. This control (wild-type, WT) allowed accounting for any side effects related to the plasmid selection or counterselection process rather than to *GIM3* itself (Fig. 1A).

To select for fluconazole-resistant mutants, we plated cells on medium containing high concentrations of fluconazole (128 µg/mL). However, we did not observe any colonies, indicating that the plated cells did not contain pre-existing resistant cells nor acquired fluconazole resistance during this rapid selection regime (Fig. 1B). Note that we used colony-forming units per mL culture medium (CFU/mL) to quantify growth on fluconazole-containing medium, since this allowed us to standardize across different dilutions and conditions tested. Importantly, we did not observe a significant difference in CFU/mL between strains with and without the plasmid when plated on medium lacking fluconazole (generalized linear model (GLM) with quasi-Poisson distribution and log link function; *P* values in Table EV1 and Fig. EV1A), suggesting that the absence of *GIM3* does not impact cell viability under unstressed conditions.

Next, we generated de novo genetic variation by exposing populations to EMS. To gain insight into the number and type of EMS-induced random mutations, we sequenced genomic DNA extracted from four different populations of cells: three derived from EMS-treated cultures and one from a non-EMS-treated culture. Analysis of the sequencing data revealed that EMS exposure introduces an average of 86–124 mutations per genome per strain, spread across the genome (Fig. EV2A–D).

We did not observe significant differences in CFU/mL between strains when plating the EMS-treated strains on medium lacking fluconazole (GLM, family = quasi-Poisson, link = log, *P* values in Table EV1, Fig. 1C). However, when the different EMS-treated populations were plated on medium with fluconazole, we found that strains lacking *GIM3* yielded approximately fourfold lower CFU/mL compared to strains expressing *GIM3* (generalized linear model with quasi-Poisson distribution and log link function, *P* values in Table EV1 and Fig. 1D). Importantly, MIC assays (Broth Dilution Assays (BDA) and Etests) show that non-EMS-treated strains lacking *GIM3* are not inherently more sensitive to fluconazole compared to strains expressing *GIM3* (Fig. EV3A,C).

Taken together, these results show that the growth of mutants on fluconazole in mutagenized populations depends on Gim3. In other words, Gim3 appears to potentiate the effect of de novo mutations in this case. Notably, although we did not observe significant differences in CFU/mL on medium lacking fluconazole (Fig. 1C), the colony size of EMS-treated cells lacking *GIM3* is on average significantly smaller than that of cells still containing *GIM3* (Fig. EV1B). Since colony size is indicative of fitness, this indicates that Gim3 can also potentiate mutations in the absence of fluconazole.

### Gim3 plays an important role in maintaining growth in fluconazole

The previous experiment showed that mutated populations expressing *GIM3* had higher CFU/mL when plated on fluconazole compared to cultures lacking *GIM3*. To further assess the

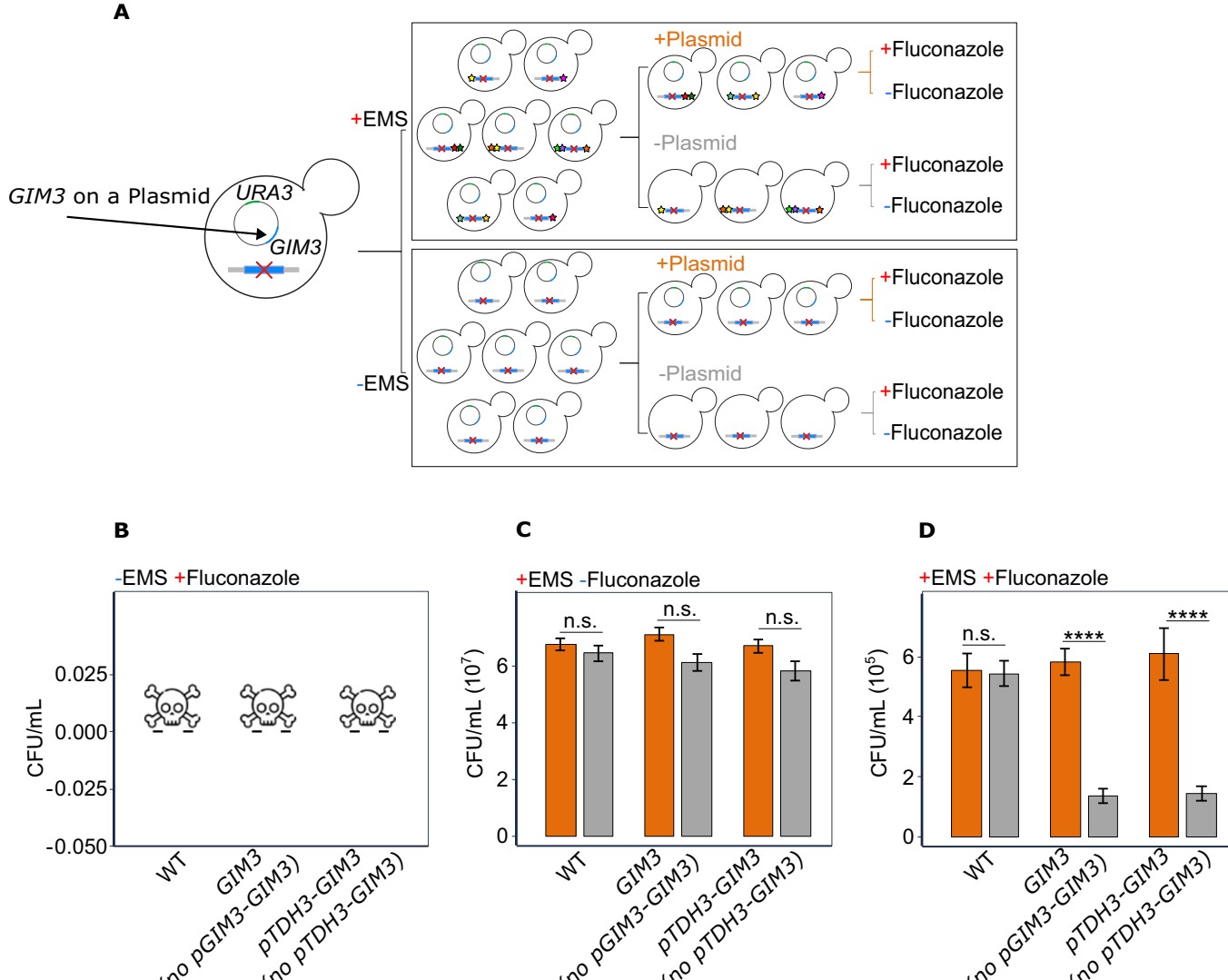

**Figure 1. Gim3 facilitates the immediate phenotypic consequences of de novo mutations on fluconazole resistance.**

(A) Experimental setup: a single yeast colony was cultivated in SC-uracil liquid media overnight. Subsequently, the culture was split into two populations; one culture received EMS treatment to generate de novo random mutations, and the second culture was treated with water. Both the EMS-treated and untreated cultures were next incubated in two types of media: SC-uracil, promoting plasmid retention, and SC + FOA, selecting for plasmid loss. The populations were subsequently plated on medium with or without fluconazole. The red cross in the yeast cells indicates the deletion of *GIM3* from the genome. The circle in the yeast cells represents the plasmid where *GIM3* was reintroduced, depicted in blue. The green color on the plasmid represents the selectable marker *URA3*, which was used to select for or against the plasmid, and hence *GIM3*. The colored stars denote various random mutations introduced by EMS. Figure created with BioRender.com. (B) Fluconazole-resistant colonies did not emerge without EMS treatment: no colonies were detected in the absence of EMS treatment at SC+fluconazole 128 µg/mL. The skull indicates that no colonies grew under these conditions. (C) No significant difference in CFU/mL in EMS-treated cultures under no stress: EMS-treated cells in the absence of fluconazole do not show a significant difference in CFU/mL in the presence or absence of *GIM3* (GLM, family = quasi-Poisson, two-sided test, link = log, the *P* values corresponding to the figure, from left to right, are 0.9685, 0.1119, and 0.1609). Error bars represent the standard error of the mean ( ± s.e.m.) for ten technical replicates. (D) *GIM3* increases CFU/mL in EMS-treated cultures under fluconazole stress: EMS-treated cultures show significant differences in CFU/mL when exposed to fluconazole, with a 4.2-fold higher CFU/mL for those that have *GIM3* compared to those that do not (GLM, family = quasi-Poisson, two-sided test, link = log, the *P* values corresponding to the figure, from left to right, are 0.9999, $6.22 \times 10^{-10}$ and $2.51 \times 10^{-10}$). Error bars represent the ±s.e.m. for $n = 10$ technical replicates. n.s. =not significant, *$P < 0.05$, ***$P < 0.001$, ****$P < 0.0001$. Table EV1 contains the *P* values for all comparisons. The orange bars indicate the presence of the plasmid, whereas the gray bars indicate the plasmid's absence. Exact genotypes of the strains used in this figure are provided in Table EV7. Source data are available online for this figure

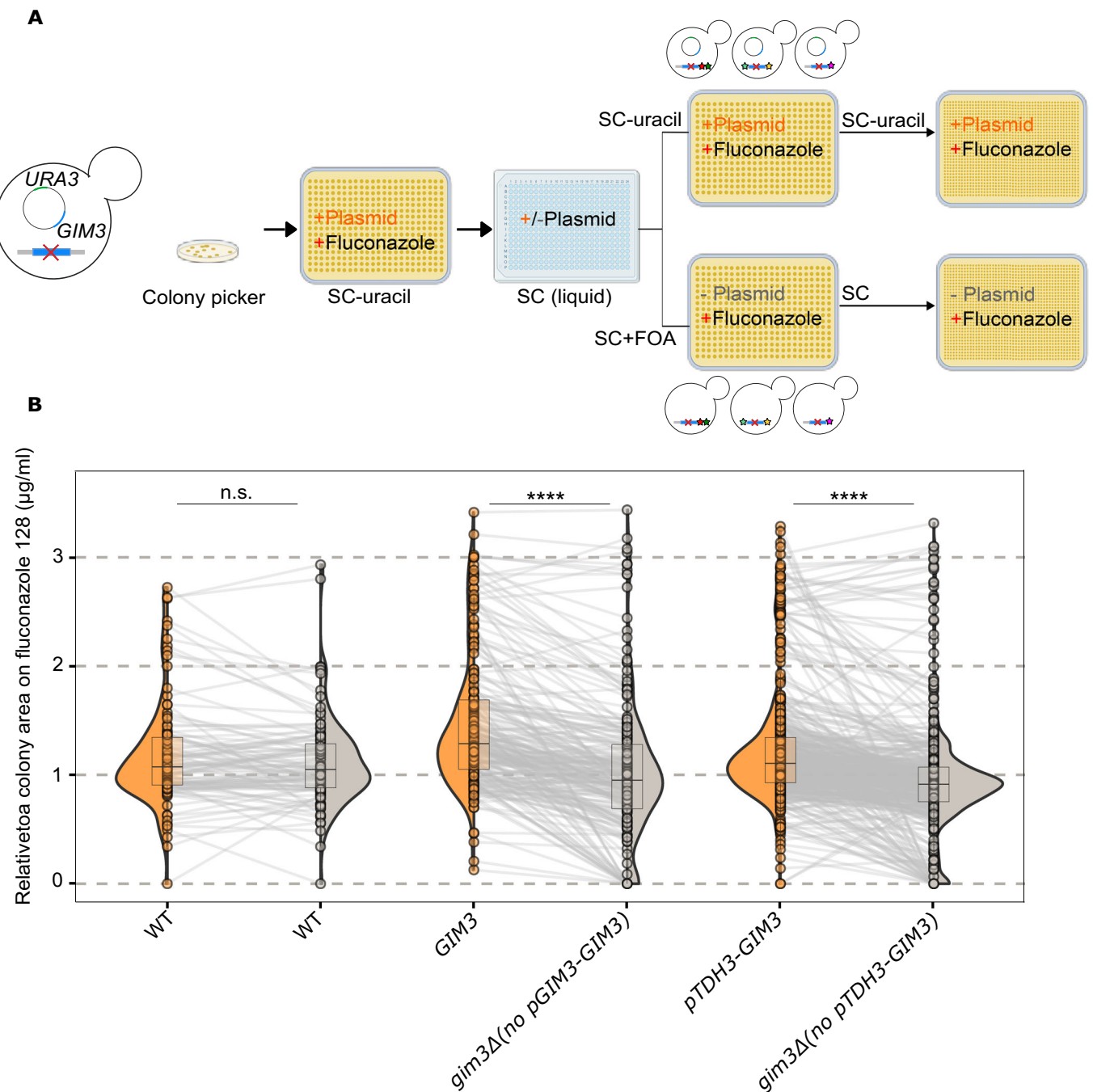

dependence of these mutants on *GIM3* activity, we tested the effect of removing *GIM3*. We selected all resistant mutants from the populations containing a plasmid (orange bars in Fig. 1D) and subsequently selected for and against the plasmid (Fig. 2A, details in "Methods" section). We evaluated the fitness of each mutant on medium with fluconazole by normalizing its colony area to that of a resistant control strain (normalized fitness).

As expected, the wild-type (WT) mutants did not show significant differences in normalized colony area with and without the plasmid (which, in this case, does not contain *GIM3*) when comparing the two conditions (Paired Wilcoxon signed-rank test, P

value = 0.43) (Fig. 2B). Second, mutants containing *GIM3* on a plasmid either under its native promoter or under the constitutively active *TDH3* promoter showed a significant decrease in normalized fitness on fluconazole after losing the plasmid (Paired Wilcoxon signed-rank test, P value = $5.1 \times 10^{-29}$ and $4.3 \times 10^{-38}$, respectively). Remarkably, losing the plasmid containing *GIM3* leads to reduced fitness in fluconazole in >80% of tested colonies (83.8%, 182 out of 217) and 87.5% (337 out of 429) for colonies with WT *GIM3* and *GIM3* overexpression, respectively.

To investigate whether the reduced growth observed on fluconazole-containing agar plates in strains lacking *GIM3* was

**Figure 2.  Gim3 plays an important role in maintaining growth in fluconazole.**

(A) Experimental setup to test the effect of *GIM3* deletion on growth under fluconazole: a total of 746 mutant colonies derived from three different populations (wild-type, *GIM3* expressed from a plasmid under its native promoter, and *GIM3* expressed from a plasmid under the *TDH3* promoter) were pinned onto a 384-format plate containing SC-uracil and fluconazole 128 µg/mL to maintain the plasmid and fluconazole resistance. Each isolated mutant was then inoculated into non-selective SC (liquid) media. These mutants were subsequently pinned onto two types of 384-format plates: one with SC-uracil and fluconazole 128 µg/mL to select for cells that retained the plasmid, and another with SC + FOA and fluconazole 128 µg/mL to select for cells that lost the plasmid. The 384-format plates were replicated in a 1536-format to obtain four technical replicates for each mutant. Colony areas were normalized using a resistant control strain (see "Methods" for details). The deletion of *GIM3* from the genome is represented by a red cross in the yeast cells, while *GIM3* on a selectable plasmid is shown as a blue circle. The green color on the plasmid represents the selectable marker *URA3*, which was used to select for or against the plasmid, and hence *GIM3*. Figure created with BioRender.com. (B) Deletion of *GIM3* reduces growth of mutants under fluconazole stress: losing the plasmids containing *GIM3* led to a significant decrease in fluconazole growth, while losing the control plasmid containing only *URA3* (and not *GIM3*) did not affect growth on fluconazole (Paired Wilcoxon signed-rank test, two-sided test, P values: 0.43, $5.1 \times 10^{-29}$, and $4.3 \times 10^{-38}$; P values are shown in the same order as they appear on the plot from left to right). n.s. = not significant, *$P < 0.05$, ***$P < 0.001$, ****$P < 0.0001$. The number of mutants analyzed for each background was $n = 100$ for WT, $n = 217$ for *GIM3*, and $n = 429$ for *pTDH3-GIM3*. The composite plot shows the colony area distribution relative to a plate control strain. This distribution is depicted using a half-violin plot, a dot plot for the median of the normalized colony areas of $n = 4$ technical replicates. Violin plots are overlaid by box plots showing the median (center line), interquartile range (25th–75th percentiles), and whiskers extending to 1.5 times the interquartile range. Source data are available online for this figure.

associated with changes in MIC, we performed Etest assays on 35 EMS-derived mutants that showed either a synthetic lethal phenotype or a ≥ 50% reduction in colony area at 128 µg/mL fluconazole following *GIM3* removal. Three strains showed a higher MIC in the presence of GIM3 (EMS18, EMS19, and EMS23), while most strains had comparable MIC values with or without *GIM3*, suggesting that for some, but not all backgrounds, *GIM3* influences the MIC (Table EV2). Interestingly, however, nearly all strains displayed pronounced trailing growth in the presence of *GIM3*, which was reduced following *GIM3* removal (Fig. EV3E). This pattern suggests that in most backgrounds, *GIM3* contributes to fluconazole tolerance. These results also indicate that the high-throughput assay used to determine fluconazole growth primarily identified tolerant rather than fully resistant variants.

In summary, these results show Gim3's role in supporting fluconazole resistance and tolerance, as the majority of mutants become more fluconazole-sensitive once *GIM3* is removed.

## Mechanisms of *GIM3*-dependent resistance and tolerance

To better understand the role of *GIM3* in mediating fluconazole resistance and tolerance, we focused on sequencing the *ERG3* gene, which is frequently associated with fluconazole resistance under the rapid selection regime used in our experiments (Cowen et al, 2006; Cowen and Lindquist, 2005; Anderson et al, 2003). We sequenced *ERG3* in 44 EMS-treated strains that showed either a synthetic lethal phenotype or increased sensitivity (more than a 50% decrease in colony area) to fluconazole when *GIM3* was lost (Fig. 2B). This analysis included 22 strains with *GIM3* under its native promoter and 22 strains with *GIM3* under the *TDH3* promoter (*GIM3* overexpression). The results reveal that 52.2% (23 out of 44) of these strains contain mutations within the *ERG3* gene (mutations are shown in Fig. 3A and Table EV2). Almost all identified *ERG3* mutations were non-synonymous mutations (missense and non-sense) (~87%, 20 out of 23), with the remaining mutations being frameshift mutations (Fig. 3B). Interestingly, our genome analysis of EMS-treated strains also showed on average significantly higher numbers of non-synonymous mutations (Fig. EV2C). Non-synonymous mutations can alter protein function and/or stability. Further analysis using the MutationExplorer online tool suggested that the non-synonymous mutations have varying degrees of

destabilizing effects on Erg3 (Fig. 3C) (Stein et al, 2019; Philipp et al, 2024).

To investigate how Gim3 could potentiate the identified Erg3 non-synonymous mutations, we experimentally assessed the stability of Erg3 proteins via a cycloheximide-based protein degradation assay (Christiano et al, 2014; Belle et al, 2006) (Fig. 4A). Specifically, we tested the stability of His-tagged Erg3$^{WT}$ and two Erg mutants with different predicted stabilities (Erg$^{G195D}$ and Erg3$^{T289P}$). The Erg3$^{T289P}$ mutation is predicted to have a more than fourfold stronger destabilizing effect on Erg3 compared to the Erg$^{G195D}$ (see Fig. 3C). All variants were analyzed in the presence and absence of *GIM3*. We find that while Erg3$^{WT}$ protein stability does not appear to depend on Gim3, the tested Erg3 mutants show reduced stability in the absence of Gim3. Interestingly, Erg3$^{T289P}$, the mutant with the lowest predicted stability, already has a lower protein half-life in the presence of Gim3 when compared to an Erg3$^{WT}$ protein (P value = 0.024, two-sided $t$ test), and this half-life is further reduced in the absence of Gim3 (P value = 0.016, two-sided $t$ test). For Erg3$^{G195D}$, we find a drastically reduced protein half-life in the absence of Gim3 (P value = 0.015, two-sided $t$ test) (Fig. 4A).

Next, we determined the fluconazole susceptibility of these Erg3 mutant strains. We found that an Erg3$^{T289P}$ mutant exhibits a high MIC$_{50}$ ( > 128 µg/mL; comparable to the MIC$_{50}$ observed for *erg3Δ*) in the presence and absence of *GIM3*. The Erg3$^{G195D}$ mutant showed an increased MIC$_{50}$ of 32 µg/mL in the presence of *GIM3* compared to 16 µg/mL in its absence (Fig. 4B). Notably, both Erg3 mutants still showed growth at fluconazole concentrations that completely inhibited the growth of the Erg3$^{WT}$ strain. Moreover, strains containing Erg3$^{T289P}$ or Erg3$^{G195D}$ also show enhanced fluconazole tolerance (Fig. 4C, Etest results after 48 h showing trailing growth), with reduced fluconazole tolerance of Erg3$^{G195D}$ in the absence of Gim3. Together, these results show that the tested Erg3 mutations affect fluconazole tolerance, and at least part of these phenotypes depend on Gim3.

To investigate how these Erg3 mutations and the presence of Gim3 affect the accumulation of toxic sterols, we next determined sterol profiles in the presence and absence of fluconazole (Fig. 4D,E). Under drug-free conditions, and in line with what was previously reported, the *erg3Δ* strain produces no ergosterol and accumulates ergosta-7,22-dienol, consistent with loss of Erg3 activity (Heese-Peck et al, 2002; Guan et al, 2009). Based on the sterol profile, Erg$^{T289P}$ has severely reduced Erg3

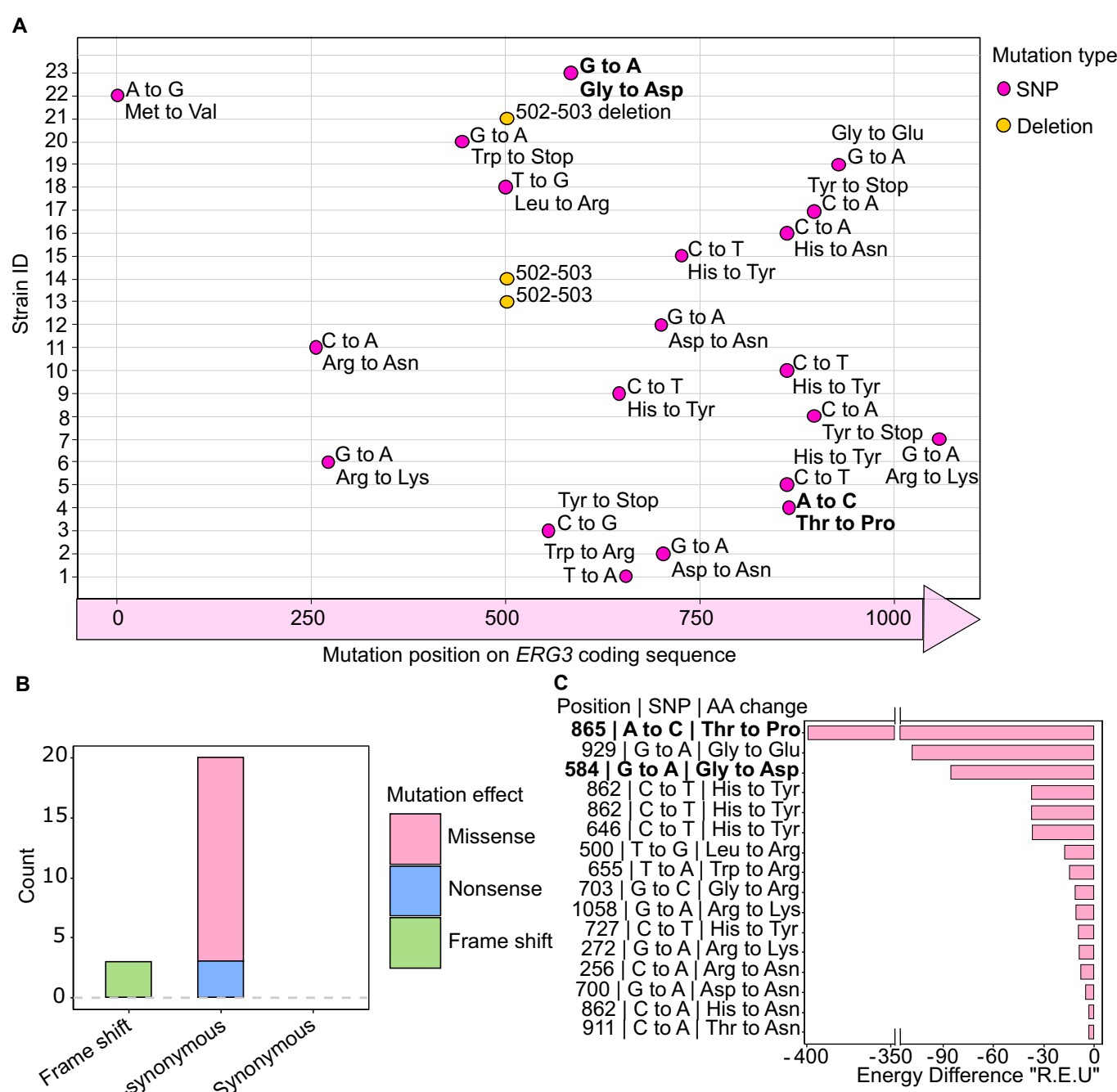

**Figure 3. Types of mutations found in *ERG3* of strains that became fluconazole-sensitive upon *GIM3* removal and their predicted effect on Erg3 stability.**

(**A**) Position and type of mutations in *ERG3* of fluconazole-resistant strains: this panel shows the position, type of mutation, nucleotide change, and resulting amino acid change in the *ERG3* gene of 23 strains that showed either a synthetic lethal phenotype or increased sensitivity (more than a 50% decrease in colony area) to fluconazole when GIM3 was removed. (**B**) Mutation types in the *ERG3* gene: a bar plot illustrating the types of mutations found (e.g., missense, nonsense, frameshift). (**C**) Predicted impact of mutations on Erg3 protein stability: a bar plot displaying the difference in relative energy units (R.E.U.) between the wild-type Erg3 protein and Erg3 proteins with specific non-synonymous mutations. More negative R.E.U. values indicate a greater destabilizing effect on the Erg3 structure. The bar plot shows 16 instead of 20 mutations because it excludes lineages carrying frameshift or premature stop codons. The MutationExplorer online tool was used to calculate the R.E.U. values (Philipp et al, 2024). Mutations highlighted in bold were selected for subsequent protein stability analyses. Strain details can be found in Table EV2, which corresponds to MT9-MT23 in Table EV7. Source data are available online for this figure.

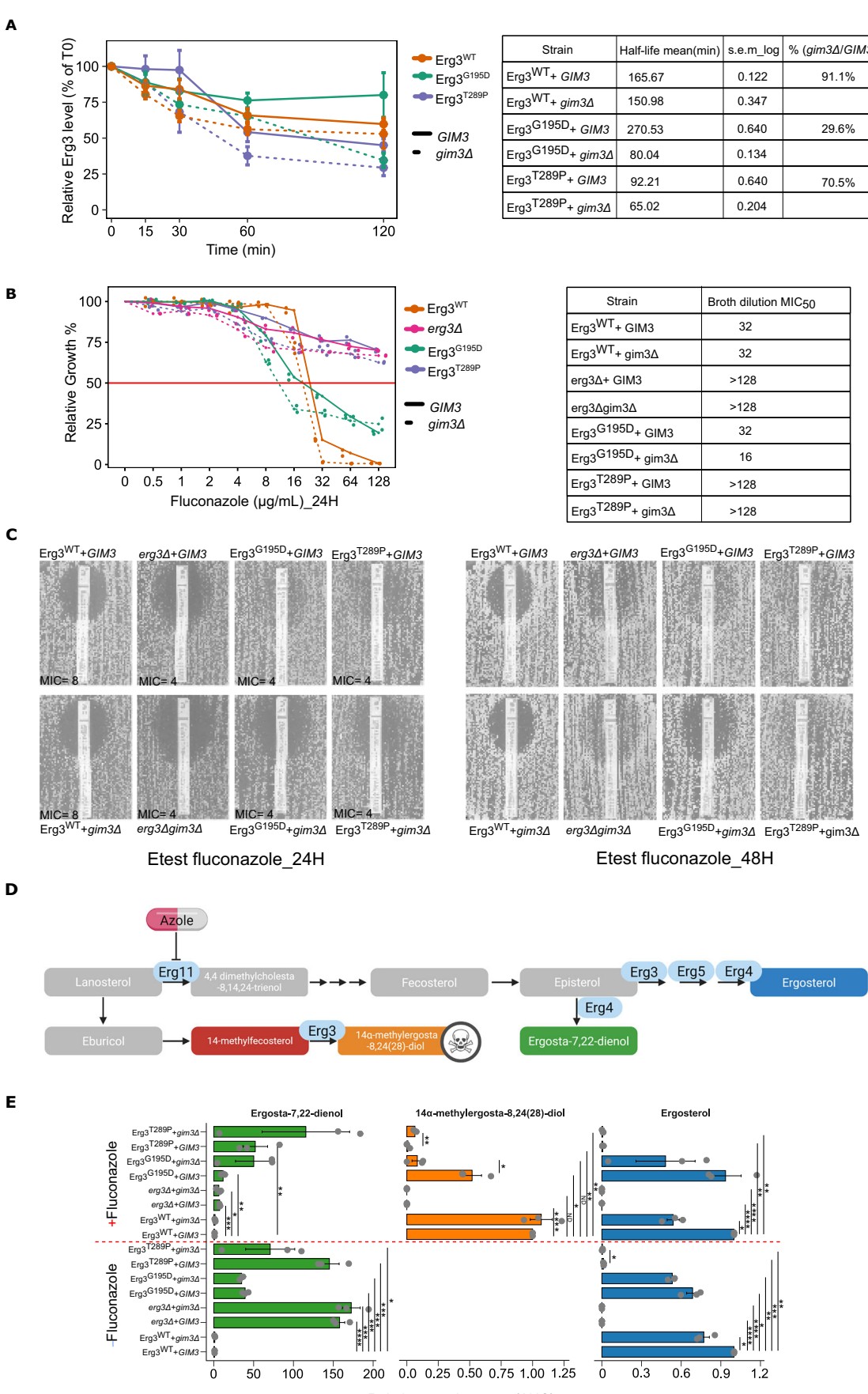

**Figure 4. Absence of Gim3 reduces the stability of Erg3 mutants and affects fluconazole susceptibility and sterol profiles.**

(A) Plot showing the percentage decrease in Erg3-His$_6$ protein levels in the presence and absence of Gim3 over time. Data represent mean ± s.e.m. from three independent biological replicates per strain and time point. Data is shown relative to the initial Erg3 protein levels at time 0 (T0) for each strain background. The table shows the per-strain protein half-life (t½, minutes) estimated from log-linear fits of %T0 vs time for each replicate, s.e.m-log indicates the standard error computed on log-transformed half-lives, and percent stability retained represents the relative half-life of each Erg3 variant in $gim3\Delta$ compared to $GIM3$. Two-sided $t$ test results comparing protein half-life values of each variant to the corresponding Erg3$^{WT}$ + $GIM3$ are provided in Table EV3. (B) Absence of Gim3 affects fluconazole susceptibility. Right: Fluconazole broth dilution assay of Erg3 variants in $GIM3$ and $gim3\Delta$ backgrounds ($n = 2$ biological replicates for each variant). Growth values were normalized to drug-free medium (Relative growth %) after 24 h exposure to increasing fluconazole concentrations, as shown for strains expressing Erg3$^{WT}$, Erg3$^{G195D}$, or Erg$^{T289P}$ either in a $GIM3$ background (solid lines) or in a $gim3\Delta$ background (dashed lines). The horizontal red line marks 50% growth relative to drug-free controls and corresponds to the MIC$_{50}$ threshold. Left: Table summarizing the MIC$_{50}$ values obtained from the broth-dilution assay for each strain. (C) Fluconazole Etest profiles for Erg3 variants ($n = 2$ biological replicates) in $GIM3$ and $gim3\Delta$ backgrounds. Representative Etest for fluconazole is shown after 24 h (left panel) and 48 h (right panel) incubation. MIC values (µg/mL) are indicated below each strip. (D) Simplified schematic of the late ergosterol biosynthesis pathway in *Saccharomyces cerevisiae*. Fluconazole inhibits Erg11, reducing ergosterol production and causing the accumulation of 14α-methyl sterol intermediates (toxic diols). When Erg3 activity is lost or reduced, these intermediates cannot be converted into the toxic diol, and alternative, less toxic products such as ergosta-7,22-dienol are formed. (E) Sterol profiles of Erg3$^{WT}$, Erg3 missense mutants (Erg3$^{G195D}$, Erg3$^{T289P}$), and an $erg3\Delta$ strain in $GIM3$ and $gim3\Delta$ backgrounds, grown with or without fluconazole. Bars represent the relative sterol content normalized to both the internal standard and Erg3$^{WT}$ under each condition. For the toxic intermediate 14α-methylergosta-8,24(28)-diol, loss of $GIM3$ significantly altered sterol levels in both Erg3$^{G195D}$ and Erg3$^{T289P}$ backgrounds ($P$ value = 0.013 and 0.006, respectively). In addition, levels of 14α-methylergosta-8,24(28)-diol were significantly reduced in the Erg3$^{G195D}$ + $GIM3$ ($P$ value = 0.038) and Erg3$^{T289P}$ + $GIM3$ ($P$ value = 0.0095) relative to Erg3$^{WT}$ + $GIM3$ under fluconazole treatment. For ergosterol, $GIM3$ deletion significantly affected ergosterol levels in the Erg3$^{WT}$ strain ($P$ value = 0.038) and in the Erg3$^{T289P}$ variant ($P$ value = 0.0259) under drug-free conditions, and also in the Erg3$^{WT}$ background in the presence of fluconazole ($P$ value = 0.018). Only statistically significant $P$ values are shown, corresponding to comparisons between the same variant in the presence and absence of $GIM3$, as well as comparisons of all variants to the Erg3$^{WT}$ reference. Significant values are indicated as n.s. = not significant, ND, not statistically testable (no variance across replicates), *$P < 0.05$, ***$P < 0.001$, ****$P < 0.0001$. All additional comparisons are reported in Table EV4. Exact strain genotypes are listed in Table EV7. Source data are available online for this figure.

activity, since Erg$^{T289P}$ strains produce no ergosterol and accumulate ergosta-7,22-dienol, mimicking the sterol profiles of an $erg3\Delta$. Strains expressing Erg$^{G195D}$ appear to retain partial Erg3 activity, producing some ergosterol while also accumulating ergosta-7,22-dienol (Fig. 4E).

Under fluconazole treatment, toxic 14α-methylergosta-8,24(28)-diol is formed in an Erg3$^{WT}$ strain. This toxic sterol does not accumulate in an $erg3\Delta$ strain, and levels of this sterol are also significantly lower in the Erg3 mutant strains than in the Erg3$^{WT}$ + $GIM3$. ($P$ value for Erg$^{G195D}$ + $GIM3$ = 0.038, $P$ value for Erg$^{T289P}$ + $GIM3$ = 0.0095, one-sample $t$ test). Interestingly, Gim3 appears to be required for the residual activity of Erg3$^{G195D}$, since almost no toxic diol is detected in the absence of Gim3.

Together, these sterol profiles are in line with what we observe in our protein stability assay (Fig. 4A), as well as our BDA and Etests (Fig. 4B,C), and indicate reduced Erg3 activity in the tested mutants, as well as a role for Gim3 in stabilizing mutant Erg3 proteins.

In summary, our findings suggest that $GIM3$-dependent fluconazole resistance and tolerance involve mutations in the $ERG3$ gene, while other resistant strains without $ERG3$ mutations also exhibited $GIM3$-dependent resistance (see also Table EV2). This indicates that in addition to $ERG3$ mutations, other complex genetic pathways could contribute to $GIM3$-dependent resistance and tolerance.

## Gim3 promotes the acquisition of fluconazole resistance upon long-term exposure to sublethal fluconazole concentrations

The previous results indicate that Gim3 promotes the occurrence and/or survival of fluconazole-resistant and tolerant mutants upon EMS-mutagenesis and subsequent "rapid" selection in media containing a high dose of fluconazole. Moreover, $GIM3$ appears crucial to maintain this growth on fluconazole. Interestingly, previous studies have shown that the mode of fluconazole selection (sudden exposure to high concentrations, as used in our experiments thus far, versus long-term exposure to lower, sublethal fluconazole concentrations as in e.g., ALE) plays a crucial role in

determining which resistance mechanisms evolve. We therefore investigated whether the role of $GIM3$ in mediating growth on fluconazole extends to modest, long-term exposure regimes where other, different resistance mechanisms might be more common. We set up a laboratory evolution experiment in which we evolved 76 parallel populations of non-resistant WT *S. cerevisiae* strains in the presence of 64 µg/mL of fluconazole. This concentration was specifically selected to represent a moderate selection pressure (Fig. 5A) (for more details, see "Methods").

After eight transfers (~35 generations), most of the populations showed a significant increase in fitness, as estimated by measuring the growth rate from the daily OD$_{600}$ measurements of the evolving populations. Figure 5B shows the significant difference in average growth rate between the populations at transfer 1 and transfer 8 (paired $t$ test, $P$ value $\leq 2.2 \times 10^{-16}$).

Next, to assess whether the observed adaptation depended on $GIM3$, a single fit evolved yeast clone was isolated from each evolved population (for more details, see "Methods"). $GIM3$ was deleted in these evolved clones, and growth before and after $GIM3$ deletion was evaluated at two fluconazole concentrations using colony area as a proxy for fitness. To select appropriate fluconazole concentrations for these assays, an unevolved wild-type (WT) strain was included on each plate as a control across a range of fluconazole concentrations. As shown in Fig. EV4, the colony area of the unevolved WT strain progressively decreased with increasing fluconazole concentration. Based on this, 128 µg/mL fluconazole was chosen as it strongly inhibited WT growth while still allowing measurable growth of evolved clones, thereby providing sufficient fitness differences, measured as colony area (Fig. 5C,D). At this concentration, 33/76 of evolved clones (43.4%, orange lines in Fig. 5D) exhibit a decrease in fitness after $GIM3$ deletion compared to their fitness before the deletion. These findings suggest that $GIM3$ influences fluconazole susceptibility independently of the mode of selection. Some evolved clones, 15/76 (19.7%, green lines in Fig. 5D) show an increase in fitness post-$GIM3$ deletion, which could indicate that $GIM3$ deletion could be adaptive in some cases under fluconazole stress. Moreover, 28/76 (36.8%, gray lines in

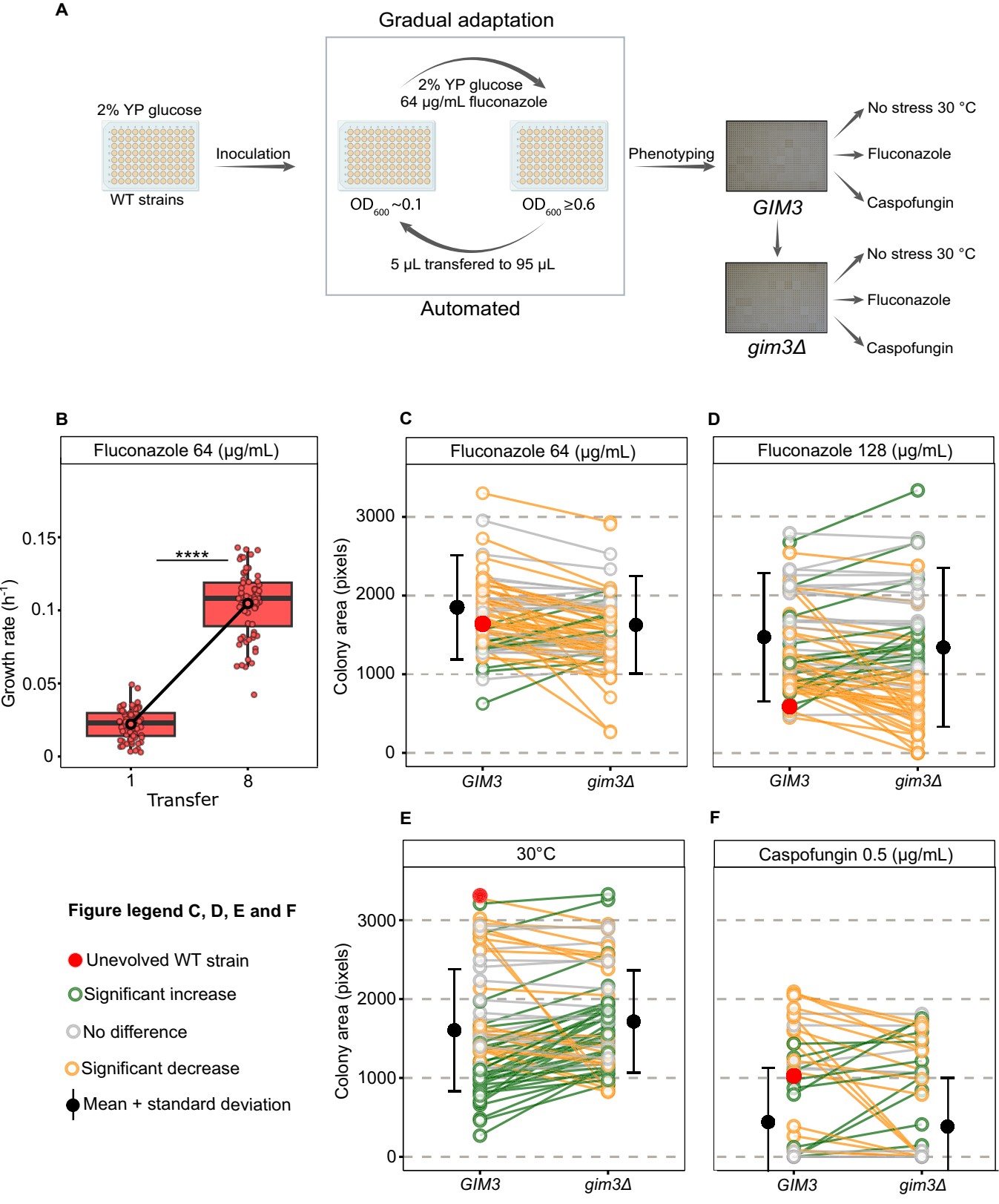

◄

**Figure 5. *GIM3* affects phenotype of fluconazole-resistant strains that have evolved under long-term exposure to lower, sublethal fluconazole concentrations.**

(A) Experimental design: schematic of the experiment outline and the phenotyping of $n = 76$ evolved populations. Figure created with BioRender.com. (B) Significant increase in growth rates after evolution under 64 μg/mL fluconazole: box plot illustrating the growth rates of populations in 64 μg/mL fluconazole at the beginning and end of the evolution experiment, showing significant differences in average growth rates before and after evolution (t test, P value ≤ 2.2 × 10⁻¹⁶, **** denotes P value < 0.0001). (C, D) Impact of *GIM3* deletion on fitness of evolved clones under fluconazole stress: line plot showing the colony area (proxy for fitness) of the evolved clones with and without *GIM3* on 64 and 128 μg/mL of fluconazole. (E) Effect of *GIM3* deletion on fitness under non-stress conditions: line plot showing the colony area of the evolved clones with and without *GIM3* at 30 °C on agar plates of 2% YP (w/v) glucose. (F) The role of *GIM3* in cross-resistance: line plot showing the colony area of the evolved clones with and without *GIM3* on 0.5 μg/mL of caspofungin; there are fewer lines here since many evolved strains did not grow at this concentration. The color of the lines in (C–F) represents changes in colony area after *GIM3* deletion under specific conditions: significant decrease (orange), no change (gray), and significant increase (green). These colors were assigned based on the results of a Wilcoxon ranked-sum test, comparing the colony area (pixels) between the 16 technical replicates of each evolved clone before and after *GIM3* deletion. The P values for each line are provided in Dataset EV3. The red dot represents the colony area of the control strain (unevolved WT strain). The black dot on each figure represents the mean of all evolved mutants before and after evolution, with the error bars indicating the standard deviations. Box plots show the median (center line), interquartile range (25th–75th percentiles), and whiskers extending to 1.5× the interquartile range. Points represent individual replicates, while the black-outlined dots indicate the mean growth rate, and the connecting black line links these means. Exact genotypes of the strains used in this figure are provided in Table EV7. Source data are available online for this figure.

Fig. 5D) of evolved clones show no significant change in fitness, suggesting that not all evolved growth on fluconazole is *GIM3*-dependent.

Interestingly, even without fluconazole stress, we observed changes in fitness upon deletion of *GIM3*, with some evolved clones showing a decrease and others an increase in fitness. Specifically, 23/76 (30.2%, orange lines in Fig. 5E) of the evolved clones exhibited decreased fitness, while 35/76 (46%, green lines in Fig. 5E) showed an increase and the rest of the evolved clones showed no change (gray lines in Fig. 5E). These results suggest that *GIM3* may interact with various mutations accumulated during the evolution experiment in different ways (positive and negative epistasis), consistent with its identification as a buffer gene. Notably, most of the evolved clones exhibited smaller colony areas (lower fitness) compared to the unevolved WT strain, indicating that mutations that improve growth in fluconazole may carry a fitness cost in unstressed environments (red dot in Fig. 5E).

Furthermore, fluconazole-resistant clones may also exhibit cross-resistance to drugs with different modes of action and targets (Lee et al, 2023). Consequently, we aimed to investigate whether *GIM3* contributes to resistance and/or tolerance to caspofungin, an antifungal agent from the echinocandin class. Caspofungin disrupts the integrity of the fungal cell wall by inhibiting β-D-glucan synthase, encoded by the *FKS1* gene. To select the appropriate concentration for evaluating whether the evolved clones exhibit cross-resistance, we tested growth at 4 different concentrations of caspofungin. We selected 0.5 μg/mL for further testing because it consistently resulted in a greater than 50% decrease in colony area of the unevolved WT strain in our plate pinning assay (Fig. EV4). This concentration imposed a relevant level of antifungal stress, allowing us to detect growth changes in evolved strains before and after *GIM3* removal. MIC assays (broth dilution assay (BDA) and Etests) showed that non-evolved strains lacking *GIM3* are not inherently more sensitive to caspofungin compared to strains expressing *GIM3* (Fig. EV3B,D). Our results show that 25% of the evolved clones (19/76) displayed larger colony size on caspofungin than the unevolved control WT strain (orange, gray, and green dots above the red dot in Fig. 5F). Of those 19 clones, 11/19 (57.9%) show decreased colony size on caspofungin after *GIM3* deletion (orange lines above the red dot in Fig. 5F). Notably, 7 of these 11 clones (63.6%) also showed a significantly smaller colony size on caspofungin-free medium (Dataset EV1). These results indicate

that for some clones, the observed reduced growth in caspofungin could (partially) reflect a general fitness defect.

In summary, our findings demonstrate that Gim3 potentiates the effects of de novo mutations, independent of the selection mode under which this resistance evolved. Moreover, the role of *GIM3* extends beyond fluconazole resistance and tolerance, as it could also confer cross-resistance and tolerance to caspofungin, highlighting its role in antifungal resistance mechanisms.

## *GIM3* influences fluconazole tolerance in *N. glabratus*

In *S. cerevisiae*, we found that fluconazole resistance/tolerance conferred by specific *ERG3* mutations often depended on the presence of *GIM3*, suggesting that *GIM3* may support resistance/tolerance either by stabilizing mutated Erg3 proteins or by enabling compensatory pathways.

Specifically, we tested the effect of three different drugs: fluconazole, caspofungin, and amphotericin B. The latter was tested since fluconazole-resistant strains of *Candida albicans* have been reported to display cross-resistance to amphotericin B (Kelly et al, 1997), and we wanted to test whether similar effects were seen in *N. glabratus*.

Across these three antifungal drugs, the Etest and broth microdilution assay (BDA) revealed assay-specific susceptibility patterns for the WT, *gim3Δ*, *erg3Δ*, and *erg3Δ gim3Δ* strains. In fluconazole, the BDA test showed a similar MIC₅₀ for *erg3Δ* compared to WT (MIC₅₀ = 4, for both strains, Fig. 6A). The Etest on the other hand, revealed a clear difference in tolerance, with *erg3Δ* showing a fourfold lower MIC compared to the wild-type (2 μg/mL versus 8 μg/mL) but developing pronounced trailing growth, a hallmark of azole tolerance (Fig. 6B). This trailing completely disappeared in the *erg3Δ gim3Δ* double mutant, indicating that *GIM3* is required for the *erg3Δ*-associated tolerance phenotype.

In amphotericin B, both assays showed that *GIM3* deletion increases drug susceptibility. Etests showed a fourfold MIC reduction for *gim3Δ* (0.125–0.032 μg/mL, Table EV6), and the BDA confirmed a decrease in MIC₅₀ values for this strain (*gim3Δ* = 0.25 μg/mL versus WT = 0.5 μg/mL, Fig. 6C). The *erg3Δ* strain did not exhibit higher MIC₅₀ than the WT, but it did show enhanced growth at higher amphotericin B concentration compared to the WT, *gim3Δ*, and *erg3Δ gim3Δ* strains (Fig. 6C, P values = 0.0021, 8.51 × 10⁻⁵, and 0.00083, respectively, two-way ANOVA test, Table EV5).

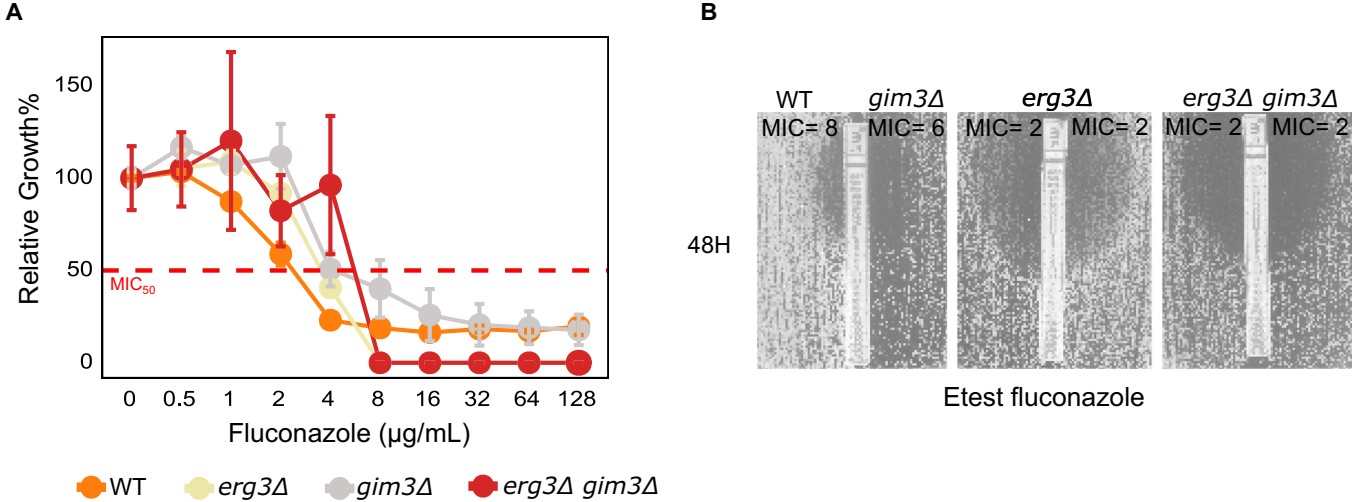

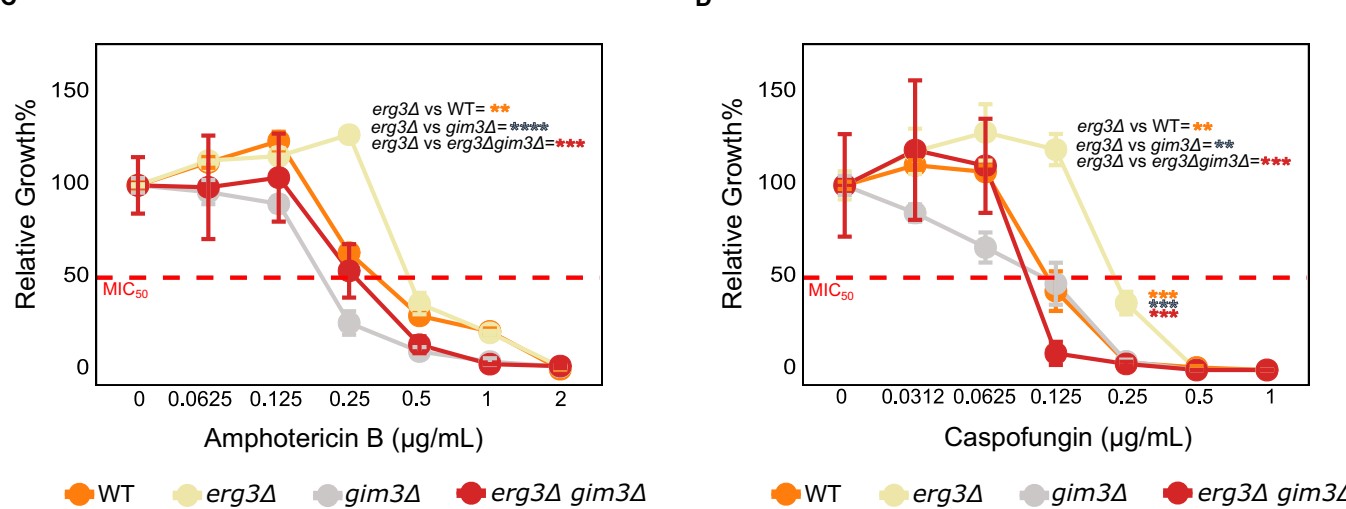

**Figure 6. Antifungal susceptibility profiles of *N. glabratus* WT, *gim3Δ*, *erg3Δ*, and *erg3Δ gim3Δ* strains.**

(A) Broth microdilution (BDA) fluconazole susceptibility curves. The BDA did not show significant growth differences between strains across fluconazole concentrations (Table EV5, two-way ANOVA), and the $MIC_{50}$ for both *erg3Δ* and WT was 4 µg/mL. Deletion of *GIM3*, either alone or in the *erg3Δ* background, did not affect fluconazole $MIC_{50}$ value. The red dashed line indicates the 50% growth threshold used to determine $MIC_{50}$. (B) Fluconazole Etest (48 h). The *erg3Δ* mutant showed a fourfold lower MIC (2 µg/mL) compared to WT (~8 µg/mL), but produced pronounced trailing growth, indicative of azole tolerance. This trailing phenotype was absent in the *erg3Δ gim3Δ* double mutant, despite an identical MIC (2 µg/mL). (C) Amphotericin B BDA assay. The *gim3Δ* mutant displayed increased susceptibility to amphotericin B, showing a lower $MIC_{50}$ than the WT (WT $MIC_{50}$ = 0.125 µg/mL, *gim3Δ* $MIC_{50}$ = 0.032 µg/mL). The *erg3Δ* strain did not exhibit a higher $MIC_{50}$ than the WT, but it did show enhanced growth at higher amphotericin B concentrations compared with the WT, *gim3Δ*, and *erg3Δ gim3Δ* strains ($P$ = 0.0021, $8.51 \times 10^{-5}$, and $8.38 \times 10^{-4}$, respectively, two-way ANOVA). (D) Caspofungin BDA assay. The *erg3Δ* strain exhibited higher $MIC_{50}$ values compared to WT and *gim3Δ* (0.25 µg/mL versus 0.125 µg/mL, 0.125 µg/mL, respectively). Deletion of *GIM3* had no significant effect on caspofungin susceptibility in either the single or double mutant, consistent with the Etest results (Table EV6). The figure further indicates significantly higher growth of *erg3Δ* strains at higher caspofungin concentrations, detailed statistical comparisons and $P$ values are provided in Table EV5. Table EV6 provides the complete Etest results for all three antifungal drugs, while Table EV5 contains the pairwise $P$ values from two-way ANOVA with Tukey's post hoc test comparing relative growth (%) across antifungal treatments among the WT, *gim3Δ*, *erg3Δ*, and *erg3Δ gim3Δ* strains. Statistical analyses shown in the figure were performed using three biological replicates. Data are shown as mean ± s.e.m. ns not significant; *$P$ < 0.05; **$P$ < 0.01; ***$P$ < 0.001; ****$P$ < 0.0001. Exact genotypes of the strains used in this figure are provided in Table EV7. Source data are available online for this figure.

Finally, for caspofungin, both methods consistently showed increased MICs for erg3Δ compared to WT and gim3Δ (0.25 μg/mL versus 0.125 μg/mL, 0.125 μg/mL, respectively, in both assays), whereas GIM3 deletion had no detectable effect in either the single or double mutant (Fig. 6D, Table EV6).

Together, these findings indicate that fluconazole tolerance, rather than classical resistance, is strongly influenced by GIM3, and this is readily detected by the Etest (via trailing growth) than by broth microdilution. Amphotericin B susceptibility is increased upon GIM3 deletion in both assay types. For caspofungin, ERG3 deletion consistently leads to elevated MICs, while GIM3 plays no major role.

Overall, our findings demonstrate that GIM3 has an important role in tolerance to fluconazole and susceptibility to amphotericin B in N. glabratus. This highlights the potential of targeting GIM3 as a strategy to combat drug resistance in clinical settings.

## Discussion

In this study, we explored the role of GIM3, one of a newly identified set of buffer genes, in modulating the phenotypic effects of de novo mutations. We examined how GIM3 not only buffers mutations but also potentiates the emergence of new traits under fluconazole stress. This approach extends the understanding of genetic buffering beyond the extensively studied Hsp90 and its role in unmasking cryptic genetic variation, revealing how buffer genes can interact with de novo mutations to influence the development of new traits.

Importantly, while both Gim3 and Hsp90 are molecular chaperones, our previous genome-wide screen (Frickel et al, 2024) showed that not all chaperones act as genetic buffers: out of 63 genes annotated as chaperones in the S. cerevisiae genome (Gong et al, 2009), only 7 displayed significant buffering activity. Hsp90 and Gim3 differ in their mechanisms: Hsp90 is an ATP-dependent chaperone that assists in folding difficult-to-fold proteins and refolding denatured ones, whereas Gim3 is a subunit of the ATP-independent prefoldin complex, which stabilizes nascent and misfolded proteins, such as actin and tubulin, and delivers them to downstream chaperones for folding. Gim3 has also been reported to maintain the solubility of unstable proteins, facilitating their degradation via the proteasome (Comyn et al, 2016). In addition, both chaperones are differentially expressed, suggesting that they may be active under distinct cellular conditions. This supports the idea that Hsp90 and Gim3 could buffer different sets of mutations or gene products depending on environmental or physiological context. Moreover, physical and genetic interactions between Gim3 and Hsp90 have been reported (Girstmair et al, 2019; Zhao et al, 2005), indicating that Gim3 may act within a broader chaperone-based buffering network, with potentially overlapping or complementary roles in maintaining proteostasis and mutational robustness.

### How does Gim3 enable growth on fluconazole?

Our results show that GIM3 expression influences the fitness effect of certain mutations that confer growth on fluconazole. Previous studies have shown that exposing S. cerevisiae to rapid selection regimes under high concentrations of fluconazole predominantly selects for ERG3 mutations, likely because these mutations impose a lower fitness cost than other resistance mechanisms, such as those affecting drug efflux (Cowen and Lindquist, 2005; Robbins and Cowen, 2021; Anderson et al, 2003). Rapid selection pressures demand quick adaptive responses, and ERG3 mutations may confer resistance more effectively than mechanisms involving efflux pump upregulation. To further explore this, we sequenced the ERG3 gene from strains exhibiting GIM3-dependent growth on fluconazole, all isolated from the rapid selection regime. We found that 52% of these strains had ERG3 mutations, including both stop codons and non-synonymous mutations, several of which are predicted to affect protein stability.

Upon further investigating 2 specific ERG3 mutations with different predicted impact on protein stability, our results indicated that Gim3 may act to stabilize the mutant Erg3 proteins (some of which could also have altered sterol profiles, as we find for the two tested ERG3 mutations), preventing their misfolding and thereby maintaining some degree of (altered) sterol biosynthesis that contributes to reduced fluconazole resistance/tolerance. Some fluconazole-resistant strains carrying non-synonymous ERG3 mutations have been described to maintain near-normal sterol profiles (Martel et al, 2010). These are known as ERG3 leaky mutants, as they retain sufficient function to reduce toxic intermediate build-up while still producing some ergosterol (Jackson et al, 2003). Gim3 interacts physically with Erg3 (Gong et al, 2009), and the stabilization of Erg3 mutants could explain their dependency on Gim3 for fluconazole resistance/tolerance.

Nonsense mutations in ERG3, introducing premature stop codons, are well-documented under fluconazole-induced stress across species such as S. cerevisiae, Candida albicans, and N. glabratus (Cowen and Lindquist, 2005; Anderson et al, 2003; Wang et al, 2022; Martel et al, 2010; Robbins and Cowen, 2021). These mutations, as well as the deletion of ERG3, inhibit toxic intermediate accumulation and alter sterol composition. To our knowledge, only one nonsense mutation identified in our study has been previously reported to contribute to fluconazole resistance, specifically following an evolution experiment at 40 °C in S. cerevisiae (Caspeta et al, 2014). This mutation occurred at the amino acid position 185, resulting in the substitution of Tyrosine (Tyr) with a stop codon (Tyr185*, C to G at position 555 of ERG3). This study found that this ERG3 mutant had a higher percentage of "bended" sterols (as opposed to "flat" sterols that are normally found in membranes), which could confer thermotolerance in the evolved S. cerevisiae strains (Caspeta et al, 2014, 2016). This suggests a possible overlap between thermotolerance and fluconazole resistance by changing membrane composition. Another non-synonymous mutation we identified has been previously reported in S. cerevisiae (see Table EV2). In previous studies, the mutation at nucleotide position 272 led to an arginine to isoleucine substitution (G to T) (Cowen and Lindquist, 2005). However, our study found a mutation at the same nucleotide position 272, resulting in an arginine to lysine substitution (G to A) at amino acid position 91.

For frameshift mutations and premature stop codons, it is more difficult to predict and understand how Gim3 could act mechanistically on these Erg3 mutants. Here, Gim3 could potentially act as previously described by Comyn et al (2016): Gim3 could promote degradation of these Erg3 mutants by maintaining their solubility and in this way facilitating proteasomal degradation (and thus mimicking an erg3Δ mutant). Absence of Gim3 would in this case

result in the accumulation of misfolded proteins, which could lead to membrane stress, indirectly influencing fluconazole resistance. Another option that we currently cannot rule out is that in the strains containing Erg3 mutants with frameshifts and premature stop codons, other (non-Erg3) Gim3-dependent mutations are responsible for the observed fluconazole resistance. Intriguingly, our results (see Table EV2) indicate that several mutants with an Erg3 protein with frameshift or nonsense mutations had a MIC lower than the WT. We hypothesize that in these strains, apart from the ERG3 mutations, other EMS-induced mutations affect fitness in fluconazole. Gim3 genetically interacts with Erg3 and other targets known to play a role in fluconazole resistance. These targets include Erg4 and Erg6 (both part of the ergosterol pathway), Cna1, the catalytic subunit of calcineurin, which interacts with Erg3 and is important for the induction of PDR1 and CDR1 (drug efflux mechanisms), Ndt80, a zinc cluster transcription factor that can bind to promoters of ERG genes and Fks1, a β-(1,3)-glucan-synthase, primarily associated with echinocandin resistance but also implicated in fluconazole resistance (Collins et al, 2007; Costanzo et al, 2016; Schuldiner et al, 2005; Hoppins et al, 2011; Gong et al, 2009; Vu et al, 2023) It remains to be investigated if (mutations in) some of these genes could underlie the observed Gim3-dependent fluconazole resistance in the EMS-treated strains without mutations in ERG3, for example.

Our findings provide a different perspective from that of Cowen and Lindquist (Cowen and Lindquist, 2005), who reported that all 12 fluconazole-resistant strains sequenced from a rapid selection regime were dependent on Hsp90 and had ERG3 mutations, whereas the three strains examined under a long-term exposure regime predominantly contained PDR1 mutations that did not depend on Hsp90 for fluconazole resistance. In our study, not all sequenced resistant mutants isolated from the rapid selection regime contained mutations in ERG3. This difference may be explained by differences in buffering activities between the two genes, but also by differences in experimental setup. We introduced random mutations via EMS and then selected resistant strains, whereas in previous studies, cells were immediately exposed to fluconazole, without deliberately introducing mutations before-hand. Our approach could have generated higher genetic variation that might allow for the occurrence of less common resistance mutations. This indicates that Gim3 might also potentiate other, non-ERG3 mutations that result in fluconazole resistance and tolerance. Further research is needed to identify these mutations and reveal the mechanisms by which this potentiation occurs. Our screening also included a larger number of mutants, 746 mutants under rapid selection and 76 evolved clones. This larger sample size may have captured a wider array of genetic backgrounds and adaptive strategies, offering a more nuanced view of GIM3's dual role in modulating fluconazole resistance (and tolerance). Our comprehensive screening approach complements the insights gained from earlier studies, enhancing our understanding of the diverse ways that buffering genes like GIM3 can influence resistance and tolerance.

We investigated whether the role of GIM3 in fluconazole susceptibility is limited to rapid selection or extends to other regimes where different mechanisms might be more common. We found that Gim3 was also required in several cases to maintain improved growth on fluconazole obtained during an adaptive

laboratory evolution experiment. The majority of evolved clones exhibited a decrease in fitness after GIM3 deletion, indicating that GIM3 is essential for maintaining fitness under fluconazole stress, regardless of the mode of selection. However, a subset of evolved clones showed an increase in fitness following GIM3 deletion, suggesting that in certain genetic contexts, the loss of GIM3 may be adaptive under fluconazole stress. These findings suggest a complex interaction between GIM3 and various mutations, influencing resistance through different pathways, and suggest that GIM3 can act both as a classic buffer, where it reduces the phenotypic effects of certain mutations, and as a potentiator, where it increases the (adaptive) effects of certain mutations (Cowen and Lindquist, 2005; Tawfeeq et al, 2024). This complex and paradoxical role of GIM3 is likely explained by its complex direct and indirect interactions with different fluconazole resistance-related genes and mutations.

GIM3 also appears to affect growth for different classes of antifungal drugs. Some clones evolved under moderate fluconazole stress also exhibited GIM3-dependent growth on caspofungin, a drug that targets the fungal cell wall by inhibiting the synthesis of β-(1,3)-glucan, a crucial component of the fungal cell wall (Lee et al, 2023). This cross-resistance is particularly intriguing, as it suggests a broader role for GIM3 beyond just facilitating resistance to drugs targeting the cell membrane (Carolus et al, 2025). The genetic interaction of GIM3 with FKS1, the gene encoding a catalytic subunit of the β-(1,3)-glucan synthase complex targeted by caspofungin, might explain this phenomenon. Mutations in FKS1 can alter the structure of the glucan synthase complex, reducing the binding affinity of caspofungin to its target and conferring resistance (Gong et al, 2009; Lee et al, 2023). However, the exact mechanisms remain unclear, highlighting the need for further research to understand GIM3's role in cross-resistance and its potential implications for therapeutic strategies.

Furthermore, our results indicate that GIM3 may also play a role in supporting fluconazole tolerance in N. glabratus. Deleting GIM3 in the clinically relevant species N. glabratus led to decreased fluconazole tolerance. The fact that removing GIM3 made an erg3Δ strain less tolerant suggests that GIM3 could support resistance by facilitating alternative compensatory mechanisms, possibly through other sterol biosynthetic components such as ERG6 (Schuldiner et al, 2005; Costanzo et al, 2016). Reports suggest that ERG6 is upregulated in response to ERG3 deletion, possibly acting as a compensatory mechanism. It is therefore plausible that GIM3 contributes to the stability, folding, or function of these alternative sterol pathway proteins. Loss of GIM3 may impair these compensatory routes, explaining the reduced fluconazole tolerance of erg3Δ gim3Δ double mutants.

Azole exposure is known to induce respiration-deficient ("petite") mutants in S. cerevisiae and N. glabratus (Zheng et al, 2025; Brun et al, 2004, 2005; Siscar-Lewin et al, 2021; Bouchara et al, 2000; Sanglard et al, 2001). Petite mutations in yeast are typically found in the mitochondrial genome or in genes related to mitochondrial function and result in respiratory deficiency. Importantly, these mutations also result in smaller colonies on medium containing glucose, since the respiratory defect leads to substantially reduced ATP yield when growing on glucose (Day, 2013; Baruffini et al, 2007; Hess et al, 2009). Mitochondrial dysfunction activates Pleiotropic Drug Resistance (PDR) regulators, increasing efflux and modulating azole response (Ferrari et al, 2009;

Tsai et al, 2006; Hallstrom and Moye-Rowley, 2000). In other words, fluconazole cannot only induce petite mutants, but these petite mutants can in turn be more fluconazole-tolerant. Although we did not perform a dedicated respiration assay (e.g., growth on non-fermentable carbon) in our study, all experiments were conducted on glucose-containing media, and we did not observe small-colony variants typically associated with petites.

Overall, our findings highlight the dual role of Gim3 in buffering and potentiating de novo mutations, contributing to both the evolution of resistance and cross-resistance in yeast. This demonstrates that Hsp90 is not unique and underscores the importance of exploring buffer genes beyond Hsp90 and their interactions with specific mutations. Understanding these interactions provides new insights into how buffer genes change the phenotypic effects of mutations, potentially influencing drug resistance, the impact of disease-causing mutations, and the broader evolutionary adaptability of organisms to new environments. Moreover, our study also suggests that buffer genes could be an interesting drug target that in some cases might reduce the development of tolerance and resistance to antibiotics.

# Methods

### Reagents and tools table

| Reagent/resource | Reference or source | Identifier or catalog number |
|---|---|---|
| Washing buffer | Thermo Fisher Scientific | Cat: #WB01 |
| ELISA assay buffer | Thermo Fisher Scientific | Cat: #DS98200 |
| 3,3',5,5'-Tetramethylbenzidine | Sigma-Aldrich | Cat: #T0565 |
| His SpinTrap column | Cytiva | Cat: #28401353 |
| NuPAGE™ 12% Bis-Tris gel | Thermo Fisher Scientific | Cat: #NP0349 |
| PageRuler™ Plus Prestained Protein Ladder, 10 to 250 kDa | Thermo Fisher Scientific | Cat: #11832124 |
| **Experimental models** | | |
| *S. cerevisiae*: BY4741, *MATa his3Δ1 leu2Δ0 met15Δ0 ura3Δ0* | | |
| *N. glabratus*: ATCC2001 | | https://www.atcc.org/products/2001 |
| pV1382 plasmid | | Vyas et al, 2018 |
| pLS10 plasmid | | Van Ende et al, 2021 |
| pYC44 plasmid | | Yáñez-Carrillo et al, 2015 |
| **Antibodies** | | |
| 6x-His Tag Monoclonal Antibody | Thermo Fischer Scientific | Cat: #MA1-21315 |
| Rabbit anti-mouse IgG | Thermo Fisher Scientific | Cat: #31194 |

| Reagent/resource | Reference or source | Identifier or catalog number |
|---|---|---|
| **Oligonucleotides and other sequence-based reagents** | | |
| crRNA Alt-R™ CRISPR-Cas9 tracrRNA | IDT | Cat: #1072533 |
| PCR primers | This study | Dataset EV2 |
| **Chemicals, enzymes, and other reagents** | | |
| Alt-R™ S.p. Cas9 Nuclease V3 | IDT | Cat: #1081058 |
| Dimethyl sulfoxide (DMSO) | Sigma-Aldrich | Cat: #D4540 |
| Hygromycin B | Sigma-Aldrich | Cat: #31282-04-9 |
| NEBuilder HiFi DNA Assembly Kit | New England Biolabs (NEB) | Cat: #E5520S |
| 5-Fluoroorotic acid (FOA) | Sigma-Aldrich | Cat: #207291-81-4 |
| Nourseothricin | Werner BioAgents | Cat: #S5.0 |
| Ecil Restriction Enzyme | New England Biolabs (NEB) | Cat: #10184089 |
| XhoI restriction enzyme | New England Biolabs (NEB) | Cat: #R0146 |
| BamHI restriction enzyme | New England Biolabs (NEB) | Cat: #R0136 |
| Fluconazole | Pfizer | Cat: #F8929 |
| Caspofungin | Sigma-Aldrich | Cat: #SML0425 |
| Ethyl methanesulfonate (EMS) | Sigma-Aldrich | Cat: #62-50-0 |
| Sodium thiosulfate ($Na_2S_2O_3$) | Sigma-Aldrich | Cat: #217263 |
| Sodium Azide | Sigma-Aldrich | Cat: #08591-1ML-F |
| Zymolyase | MP Biomedicals | Cat: #11473556 |
| Cycloheximide | Sigma-Aldrich | Cat: #C7698 |
| Tris-HCl (pH 8.0) | Formedium | Cat: #TRIS01 |
| $H_2SO_4$ | Sigma-Aldrich | Cat: #339741 |
| SDS | Roche | Cat: #11667289001 |
| Urea | Sigma-Aldrich | Cat: #U5128 |
| Bromophenol Blue | Sigma-Aldrich | Cat: #114391 |
| Silylating mixture II according to Horning | Sigma-Aldrich | Cat: #85432 |
| Cholestane | Sigma-Aldrich | Cat: #47124 |

| Reagent/resource | Reference or source | Identifier or catalog number |
|---|---|---|
| Protease inhibitors (1x cOmplete™ EDTA-free protease inhibitor cocktail) | Roche | Cat: #4693132001 |
| Glass beads (425–600 μm) | Sigma-Aldrich | Cat: #G8772 |
| Carbenicillin disodium salt | Labconsult bvba | Cat: #4800-94-6 |
| ETEST® Fluconazole (FL) | bioMérieux | Cat: #412349 |
| ETEST® Caspofungin | bioMérieux | Cat: #412268 |
| ETEST® Amphotericin-B | bioMérieux | Cat: #424317 |
| **Software** | | |
| Trimmomatic (version 0.38) | | http://www.usadellab.org/cms/?page=trimmomatic |
| BWA MEM (version 0.7.12-r1039) | | https://bio-bwa.sourceforge.net/ |
| Picard Tools (version 2.18.12) | | https://broadinstitute.github.io/picard/ |
| ImageJ (1.54j) | | https://imagej.net/ |
| OpenCFU (3.9.0) | | https://opencfu.sourceforge.net/ |
| R (version 4.3.1) | | https://www.r-project.org/ |
| RStudio (2023) | | https://posit.co/download/rstudio-desktop/ |
| SGAtools | | http://sgatools.ccbr.utoronto.ca/ |
| Inkscape (0.92.4) | | https://inkscape.org/release/inkscape-0.92.4/ |
| SnapGene (7.2.0) | | https://www.snapgene.com/ |
| MutationExplorer | | https://mutationexplorer.vda-group.de/mutation_explorer/ |
| Chromeleon v7 | | https://knowledge1.thermofisher.com/Software_and_Downloads/Chromatography_and_Mass_Spectrometry_Software/Chromeleon/Chromeleon_7_Software_Drivers_and_Release_Notes/Chromeleon_7.2.10_Software |
| **Other** | | |
| PerkinElmer Explorer G3 Workstation | PerkinElmer | https://www.biw.kuleuven.be/m2s/cmpg/hitman |
| Singer Rotor HDA Pinning Robot | Singer | |
| PIXL Colony Picker | Singer | |

## Construction of *S. cerevisiae* strains

All strains used in this study were derived from the same parental wild-type strain (BY4741, *MATa his3Δ1 leu2Δ0 met15Δ0 ura3Δ0*). A set of strains with varying levels of *GIM3* expression was created by deleting *GIM3* from the genome and then reintroducing it under two different promoters: (i) its native promoter and (ii) the constitutive *TDH3* promoter, resulting in high *GIM3* expression.

To create the first strains, *GIM3* was initially knocked out using the high-efficiency dimethyl sulfoxide (DMSO) transformation protocol and homologous recombination, replacing the open reading frame (ORF) with a hygromycin resistance marker (*HPH*) (Pan et al, 2004). The NEBuilder HiFi DNA Assembly kit (New England Biolabs, #E5520S) was then used to insert the *GIM3* gene (+ 200 bp upstream and downstream of *GIM3*) into the pV1382 plasmid (Vyas et al, 2018), which contains a *URA3* selectable marker. The presence of *URA3* on the plasmid allows for plasmid selection and counterselection (resulting in the loss of the plasmid and *GIM3*) using 5-fluoroorotic acid (FOA, Sigma-Aldrich, #207291-81-4). This resulted in a strain with the *GIM3* gene (with its native promoter) transferred from the genome to a selectable and counter-selectable plasmid. The primers used to make this strain are in the reagents and tools table.

The *GIM3* overexpression strain was constructed similarly to the one above. The only difference was that the native *GIM3* promoter within the plasmid was replaced by the *TDH3* promoter, one of the strongest constitutive promoters in *S. cerevisiae*, yielding high gene expression under various conditions (Mumberg et al, 1995; Bitter and Egan, 1984). As a control, we also created a strain that retained *GIM3* in the genome and contained the same plasmid (pV1382) as the other two strains but without *GIM3*. Moreover, we also integrated the hygromycin marker in the integration locus ARS416 of the control strain to make sure that all strains have the same genes. All transformations were confirmed by colony PCR and Sanger sequencing. All the primers are listed in the reagents and tools table.

*ERG3* mutations were integrated via CRISPR-Cas9 (Vyas et al, 2015), and a list of gRNA targeting *ERG3* sites can be found in the reagents and tools table. Yeast transformation was carried out using a DMSO-LiAc procedure (Pan et al, 2004). *ERG3* was knocked out by integrating a disruption cassette containing the nourseothricin resistance marker (NatMX). The deletion cassette was PCR-amplified from plasmid PV1382 (Addgene) (Pan et al, 2004). Transformation was performed using the high-efficiency DMSO-assisted yeast transformation protocol, and the correct integration was confirmed by PCR; primer details can be found in Dataset EV2.

## Construction of *N. glabratus* strains

*ERG3* and *GIM3* deletion cassettes (a nourseothricin resistance marker flanked by FRT sites and a 500 bp region flanking the targeted gene) were constructed in the pYC44 vector (Yáñez-Carrillo et al, 2015). The 500 bp promoter and terminator regions of *GIM3* and *ERG3* were amplified from genomic DNA using primers listed in the reagents and tools table. Both the promoter and terminator regions were inserted into the XhoI and BamHI-restricted pYC44 plasmid using Gibson Assembly. Before transformation, the vector containing the deletion cassette was digested with EciI.

Next, the *N. glabratus ERG3* and *GIM3* deletion strains were constructed in the reference strain ATCC2001 background (CBS 138). The wild-type strain was transformed by electroporation with the corresponding deletion cassette, and for *GIM3* deletion, with in vitro reconstituted RNA-protein complexes (RNPs) (Alt-R™ CRISPR-Cas9 tracrRNA, IDT, #1072533 and Alt-R™ S.p. Cas9 Nuclease V3, IDT, #1081058). The *GIM3*-specific RNA guide is listed in the reagents and tools table. Cells were plated on a YPD

agar medium supplemented with 200 µg/mL nourseothricin. Transformants were screened for the insertion of the deletion cassette by PCR using diagnostic primers listed in the reagents and tools table. Correct transformants were subsequently transformed by heat shock with the pLS10 plasmid (Van Ende et al, 2021) to induce expression of the flippase enzyme 300 µg/ml hygromycin selection (Sigma-Aldrich, #31282-04-9). Transformants were screened for removal of the nourseothricin cassette with the same diagnostic primers. To remove the pLS10 plasmid, cells were plated on non-selective YPD medium and verified by replating on YPD with 300 µg/mL hygromycin B (Sigma-Aldrich, #31282-04-9).

## Media

Media used in this study consisted of 1% (w/v) yeast extract (Neogen), 2% (w/v) peptone (Neogen), and 2% (w/v) glucose (YPD) (Sigma-Aldrich). Synthetic complete (SC) media consisted of 6.7 g/L yeast nitrogen base without amino acids (Difco), 0.77 g/L CSM-uracil (MP Biomedicals), 50 mg/L uracil (Sigma), and 2% (w/v) glucose (SC-Glu). YPD and SC media containing 200 mg/L hygromycin B (Sigma-Aldrich, #31282-04-9). Plates of these media were made with 2% agar (VWR). SC-uracil media consisted of yeast nitrogen base without amino acids 6.7 g/L, 0.77 g/L CSM-uracil, and 2% (w/v) glucose. SC + FOA consisted of yeast nitrogen base without amino acids 6.7 g/L, 0.77 g/L CSM-uracil, Uracil 50 mg/L (MP Biomedicals), 1 mg/L FOA (5-Fluoroorotic acid, Sigma-Aldrich, #207291-81-4), and 2% (w/v) glucose. Plates of these media were made with 2% agar. Luria-Bertani media (LB-carbenicillin) consisted of 10 g/L NaCl, 10 g/L tryptone, 5 g/L yeast extract, and 100 mg/L carbenicillin (Labconsult, #4800-94-6). The SC and SC-uracil agar plates used for phenotyping were prepared with fluconazole (Pfizer, #F8929). RPMI 1640-MOPS (pH 7, 2% glucose, 1% DMSO, Sigma-Aldrich, #R6504) medium was used for the antifungal susceptibility testing for *N. glabratus*.

## EMS mutagenesis

Mutagenesis was performed following a protocol adapted from a previous study (Mable and Otto, 2022). The culture was diluted using SC-uracil to obtain an $OD_{596}$ of 0.1 (~$10^7$ cells/mL) and then divided into two tubes containing 10 mL of SC-uracil media each. One of the tubes was treated with 100 µL ethyl methanesulfonate 10 µL/mL (EMS, Sigma-Aldrich, #62-50-0), in the other tube the EMS was replaced by water. This EMS concentration introduced 86–124 mutations per genome (Fig. EV2). Both tubes were incubated at 30 °C for 1 h and shaken at 250 rpm in a chest incubator. Then, the cells were washed twice with 5% sodium thiosulfate ($Na_2S_2O_3$, Sigma-Aldrich, #217263) and once with water to deactivate the EMS.

## Whole-genome sequencing for EMS-treated yeast clones

To confirm that EMS treatment effectively induced random mutagenesis and to gain a comprehensive understanding of the resulting mutations in *S. cerevisiae* BY4741, 15 EMS-treated single clones and 5 non-treated clones as controls were selected. These clones were first plated on SC media and then individually cultured in 3 mL of SC (liquid) media at 30 °C overnight, with continuous incubation on a rotating wheel. To prepare the pooled cultures, we

created four pools (3 EMS-treated pools, and 1 non-EMS-treated pool), each containing a mix of five clones. Cell numbers in each culture were accurately measured using an automated cell counter (Bio-Rad), and equal numbers of $10^7$ cells from each clone were combined.

The pooled cultures were then used for DNA extraction using a standard zymolyase-based protocol. DNA concentrations were measured using a Qubit 2.0 fluorometer, and the quality of the DNA was assessed with a NanoDrop 8000 spectrophotometer and confirmed by gel electrophoresis. The prepared samples were subsequently sent for paired-end sequencing on the DNBseq platform (BGI), with sequencing parameters set to an average read length of 150 bp and an average insert size of 350 bp. Each sample achieved a minimum sequencing coverage of 80×, providing robust data for subsequent analysis of mutations.

## Whole-genome sequencing data analysis for EMS-treated yeast clones

The raw sequencing reads were trimmed using Trimmomatic (version 0.38) to remove low-quality bases and adapter sequences (Bolger et al, 2014). Cleaned reads were then mapped to the *S. cerevisiae* reference genome (S288C_reference_sequence_R64-2-1_20150113.fasta, available at SGD Archive) using BWA MEM (Li and Durbin, 2009, version 0.7.12-r1039). Duplicate reads were marked and read group information was added using Picard (version 2.18.12). Variant calling followed GATK best practices (McKenna et al, 2010), using GATK HaplotypeCaller and GenotypeGVCFs (version 4.2.4.1), with the ploidy set to 5 to account for the mixed pool of five strains.

Variants were filtered (DP > 40, FS < 40, MQ > 40, MQRank-Sum > -10, ReadPosRankSum > -8, and QUAL > 50) to retain high-confidence variant calls, considering only single-nucleotide polymorphisms (SNPs). INDELs were excluded, and variants present in the ancestor strain were removed to focus on new mutations arising post-EMS treatment. The combined.vcf was annotated using NGSEP version 4.0.3 (https://doi.org/10.1093/nar/gkt1381), and bcftools (version 1.9) was used to collect unique variants per sample.

## Experimental set-up to investigate the impact of *GIM3* on de novo mutations under high fluconazole stress

The three constructed strains (see "Methods": "Strain construction") were each streaked from the glycerol stock onto SC-uracil agar plates and incubated at 30 °C for 72 h. A single colony from each plate was then inoculated into SC-uracil liquid media and incubated overnight at 30 °C in a rotary laboratory shaker. The overnight cultures were then diluted to an $OD_{596}$ of 0.1, divided, and treated with either EMS (Sigma-Aldrich, #62-50-0) or water. These treatments were incubated for 1 h at 30 °C in a chest incubator. Post-incubation, both mutagenized and non-mutagenized cells were centrifuged at 3000 rpm for 5 min and washed twice with 10 mL of sodium thiosulfate ($Na_2S_2O_3$) (Sigma-Aldrich, #217263), each time centrifuging at 3000 rpm for 5 min. After the $Na_2S_2O_3$ washes, the cells were washed with water, transferred to new tubes, and centrifuged again. The water was then removed, and 10 mL of SC (liquid) media was added and then incubated overnight at 30 °C and 200 rpm.

The cells were centrifuged at 3000 rpm for 5 min, washed with water, and subsequently divided into two cultures: one in SC-uracil (15 mL) to select for the plasmid and another in SC + FOA (15 mL) to lose the plasmid and hence *GIM3*. Both cultures were incubated at 30 °C and 200 rpm for 6 h. Following this, the cultures underwent an additional incubation step where they were diluted (1:10 for SC-uracil and 1:2 for SC + FOA) and incubated for another 6 h. This additional step was included because, in our pilot experiments, we observed that wild-type cells grew more slowly in FOA media compared to SC-uracil media. Therefore, to obtain a comparable number of cells before plating, this dilution step was performed.

Four different cultures were derived from each of the three initial starting cultures. Each of these four cultures was washed twice with SC (liquid) medium and then plated on both SC plates (without stress) and SC plates containing 128 µg/mL fluconazole (Pfizer, #F8929). For plating, 300 cells were used for the SC plates, and 1 million cells were used for the fluconazole plates to ensure countable colonies on each plate. Each of the four strain cultures was plated on 10 SC and SC + fluconazole agar plates, resulting in 10 technical replicates for each culture on SC media and SC + fluconazole. SC media was used to avoid differences in colony counts that may result when using SC-uracil vs SC + FOA. Plates without stress were incubated for 48 h, while plates with stress were incubated for 4 days at 30 °C. The colony count was then analyzed using ImageJ (version 1.54j), and the data were statistically assessed with a generalized linear model. This protocol enabled the assessment of *GIM3's* role in altering the immediate phenotypic consequences of new mutations under high fluconazole stress. Additionally, plates containing strains treated with EMS and plated without fluconazole were analyzed for colony area using OpenCFU (version 9.3.0) software.

## Picking colonies with increased fitness on fluconazole and selecting for and against the plasmid

Mutants with increased fitness on fluconazole, which had undergone the previously described setup (Fig. 1A) and grown under 128 µg/mL fluconazole stress with either the control plasmid, *GIM3* with the native promoter, or *GIM3* with an overexpression promoter, were selected from the SC+ fluconazole plates using the PIXL colony picker (Singer). These mutants were then pinned onto SC-uracil+fluconazole 128 µg/mL (Pfizer, #F8929) rectangular agar plates in a 384-pin format and incubated for 72 h at 30 °C. Next, colonies were inoculated into a 384-well plate, with each well containing 80 µL of SC (liquid) media, using the Singer rotor robot. The plates were incubated overnight to allow for a mix of cells with and without the plasmid in each well. Subsequently, the cells were pinned onto SC-uracil+fluconazole plates and SC + FOA+fluconazole plates to select for and against the plasmid. From the SC-uracil agar plates, strains were pinned into SC-uracil liquid in a 384-well plate to prepare glycerol stocks. A control WT strain, grown separately in SC, was added to specific wells at this step. From the FOA agar plates, strains were pinned into SC liquid (since the plasmid was already lost), and the same control strain was added to the corresponding wells here as well. Because the control was only added during the liquid culture step and not plated on either SC-uracil or FOA, it never underwent FOA selection. Finally, the cells were pinned in a 1536 format (four replicates per resistant

strain) onto SC+fluconazole agar plates and incubated at 30 °C for 48 h. This allowed us to measure the colony area of mutant cells with and without the plasmid, thereby comparing the effect of removing *GIM3* on mutant growth. The colony area (in pixels) of each resistant strain was determined using the online SGA tool, with the median of the four replicates used as a measurement of fitness. The colony area was then normalized to that of an evolved resistant control strain (MT1 in Table EV7 strain evolved on fluconazole stress using ALE) to account for variations across plates (Wagih et al, 2013).

Furthermore, we selected 44 strains exhibiting either a synthetic lethal phenotype or increased sensitivity (defined as a reduction of more than 50% in colony area) following the loss of *GIM3*. Samples were prepared according to the Sanger sequencing sampling protocol and subsequently submitted for Sanger sequencing. Sequencing results were analyzed using SnapGene software (version 6.1.1.).

## Protein extraction and cycloheximide chase assay

Stability of mutant Erg3 was assessed using a cycloheximide chase assay (Lan et al, 2021; Belle et al, 2006; Samant et al, 2018). Strains were inoculated in 5 mL SC (liquid) medium and grown overnight at 30 °C to saturation. Cultures were then diluted to an $OD_{600}$ of 0.2 in 100 mL SC (liquid) medium and allowed to grow to an $OD_{600}$ of 0.8, corresponding to approximately two doublings. Cycloheximide (Sigma-Aldrich, #C7698) was added to each culture to a final concentration of 200 µg/mL at the indicated timepoints (0, 15, 30, 60, and 120 min). At each timepoint, 12.5 mL of culture (corresponding to 10 $OD_{600}$ units) was harvested and rapidly transferred to 15 mL Falcon tubes containing 200 µL of 20× stop mix (200 mM sodium azide, Sigma-Aldrich, #08591-1ML-F). Tubes were immediately placed on ice to halt protein synthesis. Cells were pelleted by centrifugation at 3000 rpm for 3 min at 4 °C, the supernatant was removed, and cell pellets were resuspended in 100 µL lysis buffer containing 10 mM Tris-HCl (pH 8.0)(Formedium, #TRIS01), 1% SDS (Roche, #11667289001), 8 M urea (Sigma-Aldrich, #U5128), 0.01% bromophenol blue (Sigma-Aldrich, #114391), and protease inhibitors (1× cOmplete™ EDTA-free protease inhibitor cocktail, Roche, #4693132001). The suspension was transferred to a 1.5 mL microcentrifuge tube containing acid-washed glass beads (425–600 µm; Sigma-Aldrich, #G8772), vortexed for 5 min at 4 °C, and then heated to 65 °C for 10 min to denature proteins. Lysates were centrifuged at 13,000 rpm for 2 min to remove debris. Supernatants containing solubilized proteins were collected and frozen at -80 °C and then used for ELISA.

## Enzyme-linked immunosorbent assay (ELISA)

A colorimetric enzyme-linked immunosorbent assay (ELISA) was used for the measurement of the His-tagged Erg3p in the cell lysates. First, 100 µL of each cell lysate was added to a MaxiSorp Immulon plate (Thermo Fisher Scientific, #442404). The plate was then incubated at 37 °C in a plate shaker at 300 rpm for 60 min to ensure optimal binding. After incubation, each well was washed three times with 100 µL of washing buffer (Thermo Fisher Scientific, #WB01). Next, anti-His antibody (6x-His Tag Monoclonal Antibody, Thermo Fisher Scientific, #MA1-21315) was diluted (1:1000) in Assay Buffer (Thermo Fisher Scientific,

#DS98200) and 100 μL of the solution were added to each well. The plate was incubated again at 37 °C in a plate shaker at 300 rpm for 60 min. Following the incubation with the primary antibody, the wells were washed three times with 100 μL of washing buffer, and subsequently, 100 μL of Rabbit anti-mouse IgG (1:10,000 dilution in milk, Thermo Fisher Scientific, #31194) solution (1:10,000 dilution in Assay Buffer) was added to each well. The plate was then incubated at 37 °C in a plate shaker at 300 rpm for 60 min. The wells were washed three times again with 100 μL of washing buffer, and then 100 μL of TMB substrate (3,3',5,5' tetramethylbenzidine, Sigma-Aldrich, #T0565) was added to each well. The plate was incubated for 10 min at RT, away from light, until a blue color developed. The reaction was then quenched by adding 100 μL of 0.16 M $H_2SO_4$ (Sigma-Aldrich, #339741) to each well. Finally, absorbance was measured at 450 nm with the Multiskan™ FC Microplate Photometer (Thermo Fisher Scientific). The amount of His-tagged Erg3p was calculated based on a standard curve of known amounts of His-tagged proteins, purified through a His SpinTrap column (Cytiva, #28401353). The calculated His-tagged Erg3p values were then normalized to total protein levels, as quantified from Coomassie Blue-stained gels using ImageJ. Protein stability was assessed by plotting the relative values of the target protein at each time point relative to the initial time point (0 min).

## SDS PAGE and Coomassie protein staining

A total of 16 μL of cell lysate from each sample, as described above, was transferred to a new Eppendorf tube and subsequently boiled at 95 °C for 10 min. Proteins were then separated on a NuPAGE™ 12% Bis-Tris gel (Thermo Fisher Scientific, #NP0349), using the PageRuler™ Plus Prestained Protein Ladder, 10 to 250 kDa (Thermo Fisher Scientific, #11832124) as a reference. After electrophoresis, the gel was stained with InstantBlue Coomassie Protein Stain (Abcam, #ab119211) for 1 h, followed by a 5-min wash with ddH₂O before visualization using the FusionFX imaging system (Vilber).

## Antifungal susceptibility testing

The Minimal Inhibitory Concentration ($MIC_{50}$) of fluconazole was determined according to the M27-A3 protocol (National Committee for Clinical Laboratory Standards (NCCLS), 2002). A twofold dilution range of the drug was prepared in a total volume of 200 μL RPMI 1640 -MOPS (pH 7, 2% glucose, 1% DMSO, Sigma-Aldrich, #R6504) medium with approximately 200 cells in a round-bottom 96-well polystyrene microtiter plate (Greiner). The antifungal concentration ranged from 0 to 16 μg/mL for amphotericin B (Sigma-Aldrich, #A9528) and 0 to 8 μg/mL for caspofungin (Sigma-Aldrich, #SML0425) and 0, 64, 128, and 512 μg/mL for fluconazole (Pfizer, #F8929) in RPMI 1640 + MOPS (liquid), The MICs were determined after 48 h through optical density measurement at 600 (nm) with a Synergy H1 microplate reader (Biotek). The growth cut-off of all MIC values from BDA was 50% growth compared to the drug-free control.

## Etest for MIC determination

Etest® is one of the most widely used agar diffusion methods for antifungal susceptibility testing of both yeasts and molds (Bellanger et al, 2020). Strains of S. cerevisiae were retrieved from −80 °C stocks and initially streaked onto SC-uracil agar plates to select for plasmid-containing strains and incubated at 30 °C for 48 h. Single colonies were picked and inoculated in SC medium and incubated overnight at 30 °C. The overnight cultures were then streaked onto SC + FOA agar plates to counterselect for plasmid loss.

Single colonies from the SC + FOA plates were subsequently streaked onto SC agar plates and further screened for growth on SC-uracil, SC + FOA, and SC to confirm the successful loss of the plasmid. After confirming plasmid loss, strains were maintained on SC plates.

For the Etest assay, single colonies from the final SC plates (plasmid-lost strains) and SC-uracil plates (plasmid-retained strains) were picked and inoculated in their respective media (SC (liquid) for plasmid-lost strains and SC-uracil for plasmid-containing strains) and incubated overnight at 30 °C. Overnight cultures were washed three times with SC (liquid) medium, and the $OD_{600}$ was adjusted to 0.5 for S. cerevisiae.

A sterile cotton swab was soaked in the adjusted cell suspension for 10 s and used to evenly spread the inoculum over one half of a 2% YPD agar plate, with one half containing the plasmid strain and the other half without the plasmid. Plates were allowed to dry for 2–3 min before Etest strips for fluconazole (bioMérieux, #412349) and caspofungin (bioMérieux, #412268) were applied using sterile tweezers. Plates were incubated at 30 °C and imaged every 24 h for 72 h.

The minimal inhibitory concentration (MIC) was indicated by the boundary of the growth inhibition zone on the strip, following the manufacturer's guidelines. Trailing growth or tolerance was assessed by observing trailing growth within the zone of inhibition. For the EMS-treated strains, 35 of the 44 ERG3-sequenced strains were used for the MIC assay using the Etest.

The Etest protocol for N. glabratus followed the same general procedure as described for S. cerevisiae, with specific modifications in media, incubation temperature, and OD adjustment. Strains were retrieved from -80 °C stocks and initially streaked onto YPD agar plates, followed by overnight culture in 2% YPD at 37 °C. Overnight cultures were washed in PBS, and the $OD_{600}$ was adjusted to 0.2.

For the Etest assay, a sterile cotton swab was soaked in the adjusted cell suspension and spread evenly over the surface of the agar plate of RPMI 1640 + MOPS + 2% glucose + 1.5% agar (solid). Etest strips for fluconazole (bioMérieux, #412349), caspofungin (bioMérieux, #412268), and amphotericin B (bioMérieux, #424317) were applied using sterile tweezers, ensuring even contact with the agar surface. Plates were incubated at 37 °C and imaged every 24 h for 96 h using an Epson V600 flatbed scanner. The MIC was indicated by the boundary of the growth inhibition zone on the strip, following the manufacturer's guidelines. Trailing growth and tolerance were assessed by examining residual growth within the zone of inhibition. Two strains were tested per plate using two Etest strips to optimize space and resources.

## Sterol extraction and GC–MS analysis

The cells were grown in YPD with or without 16 μg/ml fluconazole (Pfizer, #F8929) for 24 h at 30 °C until the stationary phase. Cultures were harvested by centrifugation, washed once with 1× PBS, and cell pellets (20 mg wet weight) were resuspended in

300 µL saponification solution (4.46 mM KOH in 62.7% ethanol). Samples were vortexed, transferred to capped glass vials, and incubated at 80 °C for 1 h, then cooled to room temperature. Then, 100 µL MilliQ water and 400 µL hexane containing 1 µL of 5 mg/mL cholestane (internal standard) (Sigma-Aldrich, #47124) were added. Samples were vortexed for 3 min and left for 20 min to allow phase separation. The upper (hexane) layer (350 µL) was removed to a new glass tube. A second extraction was performed by adding 600 µL hexane, vortexing for 3 min, and allowing separation for 20 min. The upper layer (550 µL) was collected and combined with the first extract. The combined hexane fraction was dried by vacuum centrifugation at room temperature. Dried extracts were resuspended in 60 µL hexane and derivatized with 10 µL silylation reagent (Sigma-Aldrich, #85432) by brief vortexing and incubation at room temperature for ≥1 h. A 50 µL aliquot was transferred into a glass insert for GC-MS analysis.

Sterol analysis was performed on a Thermo Scientific Trace 1300 gas chromatograph coupled to an ISQ QD mass spectrometer, equipped with a Restek Rxi-5ms capillary column and a TriPlus RSH autosampler. Helium was used as the carrier gas at a flow rate of 1.4 mL per minute. Samples were injected in split mode (1:10) at 250 °C, following an initial 1 min hold at 50 °C. The oven temperature was then increased from 50 to 260 °C at a rate of 50 °C per minute, and subsequently from 260 to 325 °C at 2 °C per minute, with a final 3 min hold at 325 °C. The mass spectrometer was operated in electron ionization mode (70 eV), scanning from $m/z$ 50 to 600. The transfer line and ion source temperatures were set to 325 °C and 250 °C, respectively.

Sterols were identified based on their retention times relative to the internal standard (cholestane) and by comparing their mass spectral fragmentation patterns to the NIST/EPA/NIH mass spectral library (version 2) and published yeast sterol reference spectra (Müller et al, 2017), using Chromeleon v7 for data processing. Quantification was performed by integrating the base ion peaks relative to the internal standard. Each strain was analyzed in three biological replicates.

## Evolution experiment setup

The wild-type strain BY4741 was streaked on a 2% YP (w/v) agar plate and incubated for 72 h at 30 °C. Five independent colonies were each inoculated into five separate test tubes with 5 mL of 2% YP (w/v) glucose and incubated overnight at 30 °C. From these cultures, multiple replicates were used to inoculate 96-well plates (100 µL per well). The evolution experiment was done by using a custom-made Perkin-Elmer Explorer G3 workstation robotic system. The 96-well plates were centrifuged at 3000 rpm for 3 min, after which 100 µL of 2% YP (w/v) glucose was removed and replaced with 100 µL of 2% YP (w/v) glucose containing 64 µg/mL fluconazole. The optical density (OD$_{600}$) was adjusted to ~0.1 using this media, and the cultures were incubated at 30 °C. OD measurements were taken at least twice daily. Once the average OD of the 96-well plates (excluding blanks and border wells) reached ≥0.6, 5 µL from each well was transferred into 95 µL of 2% YP (w/v) glucose+ 64 µg/mL fluconazole (Pfizer, #F8929) and incubated at 30 °C. This bottlenecking process was repeated for 20 days, with eight transfers in total. Growth rates were calculated from the regular OD measurements. The experiment was stopped as the majority of strains showed a significant increase in growth

rate after 8 transfers. In total, 76 strains were evolved during the experiment.

## Sampling fit clones from the evolved populations

One fit-evolved yeast clone was isolated from each of the 76 evolved populations. Plates containing the evolved populations were thawed, and 5 µl of each was inoculated into 150 µl of YP with 2% (w/v) glucose for overnight growth. Then, 3 µl of the overnight culture was streaked onto plates containing the same growth medium used during the evolution experiment. Plates were incubated at 30 °C until single colonies were clearly visible. From each plate, one large colony was selected and used to generate the GIM3 deletion strains.

### GIM3 knock-out from the fit clones and phenotyping

To evaluate whether the fluconazole resistance of the evolved strains under a moderate selection regime depends on GIM3, we deleted GIM3 from each of the fit clones isolated as described above. GIM3 knockout clones were created using a high-efficiency dimethyl sulfoxide (DMSO) transformation protocol and homologous recombination, replacing the open reading frame (ORF) with a hygromycin resistance marker (gim3Δ::HPH) (Pan et al, 2004).

The fit clones, the unevolved control strain MT1 (Table EV7), and the transformed strains (GIM3-knockout clones) were grown in 2% YP (w/v) glucose liquid media in 96-well plates for 24 h at 30 °C, then pinned onto 2% YP (w/v) glucose agar plates using a Singer Rotor HDA pinning robot in a 384 format. These plates were incubated for 48 h at 30 °C and then further pinned onto agar plates in a 1536 format. From these, the strains were pinned onto plates containing 64, 128, and 256 µg/mL of fluconazole (Pfizer, #F8929), as well as 0.125, 0.25, 0.5, and 1 µg/mL of caspofungin (Sigma-Aldrich, #SML0425) to assess multidrug resistance and the influence of GIM3 deletion.

All analyses were based on plates incubated for 48 h at 30 °C. The colony area (in pixels) of each resistant strain was determined using the SGA tool, with the median of the 16 replicates used as a measurement of fitness (Wagih et al, 2013).

## Statistical analysis

Statistical analysis was performed using R (version 4.3.1) and RStudio (2023). To compare CFU/mL across different cultures, a generalized linear model with a quasi-Poisson distribution and log link was applied to account for overdispersion. Differences in growth between mutants before and after plasmid removal (including the control plasmid, a plasmid with GIM3 under its native promoter, and one with GIM3 under the TDH3 promoter) were assessed using a paired Wilcoxon signed-rank test.

Protein stability data were expressed as the percentage of the initial Erg3 protein level at time 0 (T$_0$) for each strain background. Protein half-lives (t½, minutes) were calculated for each biological replicate by fitting a log-linear regression model to %T$_0$ versus time. Two-sided $t$ test was performed to compare protein half-life values of each variant to Erg3$^{WT}$ + GIM3. For sterol quantification, two-sided $t$ tests was applied to compare each strain with the Erg3$^{WT}$ reference strain within each condition ( ±fluconazole). Additional

one-sample *t* tests were performed to assess differences in sterol profiles between the +*GIM3* and *gim3Δ* backgrounds for each Erg3 variant to Erg3$^{WT}$ + *GIM3*.

To evaluate changes in population growth rates before and after the evolution experiment, a *t* test was used. In addition, differences in the medians of colony area of evolved clones before and after *GIM3* deletion were assessed using the Wilcoxon signed-rank test. Wilcoxon rank-sum tests were also used to compare colony size distributions between EMS-treated strains that differed by the presence or absence of *GIM3* under non-stress conditions. For antifungal susceptibility assays (broth microdilution assay, BDA) for *N. glabratus*, MIC$_{50}$ values were calculated by determining the fluconazole concentration that reduced growth to 50% of the no-drug control. Relative values were two-way ANOVA for each broth dilution assay (BDA) experiment, with Tukey's HSD post hoc test applied for pairwise comparisons. All comparisons were based on three biological replicates.

Finally, for the EMS mutational spectrum analysis, *t* tests were employed to compare the average number of SNPs in coding versus non-coding regions, and the number of synonymous versus non-synonymous mutations. A significance threshold of 0.05 was applied for all tests, with the following significance notations: n.s. =not significant, $*P < 0.05$, $**P < 0.01$, $***P < 0.001$, $****P < 0.0001$.

## Data availability

The raw whole-genome sequencing data generated in this study have been deposited in the NCBI Sequence Read Archive (SRA) under BioProject accession number PRJNA1170047. The dataset includes raw whole-genome sequencing data from three EMS-treated populations and one control population of *Saccharomyces cerevisiae* (BY4741, derived from the S288C strain). All other source data is provided in the Source Data file accompanying this manuscript.

The source data of this paper are collected in the following database record: biostudies:S-SCDT-10_1038-S44319-026-00702-x.

## Peer review information

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

## Acknowledgements

Special thanks to Prof. Piet van den Berg, Dr. Lloyd Cool, and Ir. Michiel Schreurs for the valuable suggestions on statistical analysis. Thanks to

Dr. Quinten Deparis for analyzing the whole-genome sequencing data and to Sasha Yogiswara for assistance with various laboratory and hands-on support in the lab. Mohammed T. Tawfeeq and Jolien Vreys acknowledge the support from FWO via a PhD fellowship (Grant No. 11H1823N and 11L0423N, respectively). D. Konstantinidis was supported by an FWO postdoc fellowship (Grant No. 1283423 N) and a MSCA postdoctoral fellowship (ProteoYeast, Grant No. 101065618). The evolution experiment was carried out using the Perkin Elmer Explorer G3 workstation robot system (HiTMan), funded by an FWO large-scale research infrastructure grant (Grant No. I011820N). Work in prof. Kevin Verstrepen's team was also supported by KU Leuven through a KU Leuven C1 grant (C16/23/007).

## Author contributions

**Mohammed T Tawfeeq:** Conceptualization; Data curation; Formal analysis; Investigation; Visualization; Methodology; Writing—original draft; Writing—review and editing. **Dimitrios Konstantinidis:** Data curation; Formal analysis; Investigation; Methodology. **Ana Lucia Rocha Iraizos:** Data curation; Investigation; Visualization; Methodology; Writing—review and editing. **Wouter Van Genechten:** Investigation; Methodology; Writing—review and editing. **Jolien Vreys:** Investigation; Methodology; Writing—review and editing. **Lieselotte Vermeersch:** Data curation; Formal analysis; Investigation; Methodology; Writing—review and editing. **Karin Voordeckers:** Conceptualization; Methodology; Writing—review and editing. **Patrick Van Dijck:** Conceptualization; Supervision; Writing—review and editing. **Kevin J Verstrepen:** Conceptualization; Supervision; Funding acquisition; Writing—review and editing.

Source data underlying figure panels in this paper may have individual authorship assigned. Where available, figure panel/source data authorship is listed in the following database record: biostudies:S-SCDT-10_1038-S44319-026-00702-x.

## Disclosure and competing interests statement

The authors declare no competing interests.

# Expanded View Figures

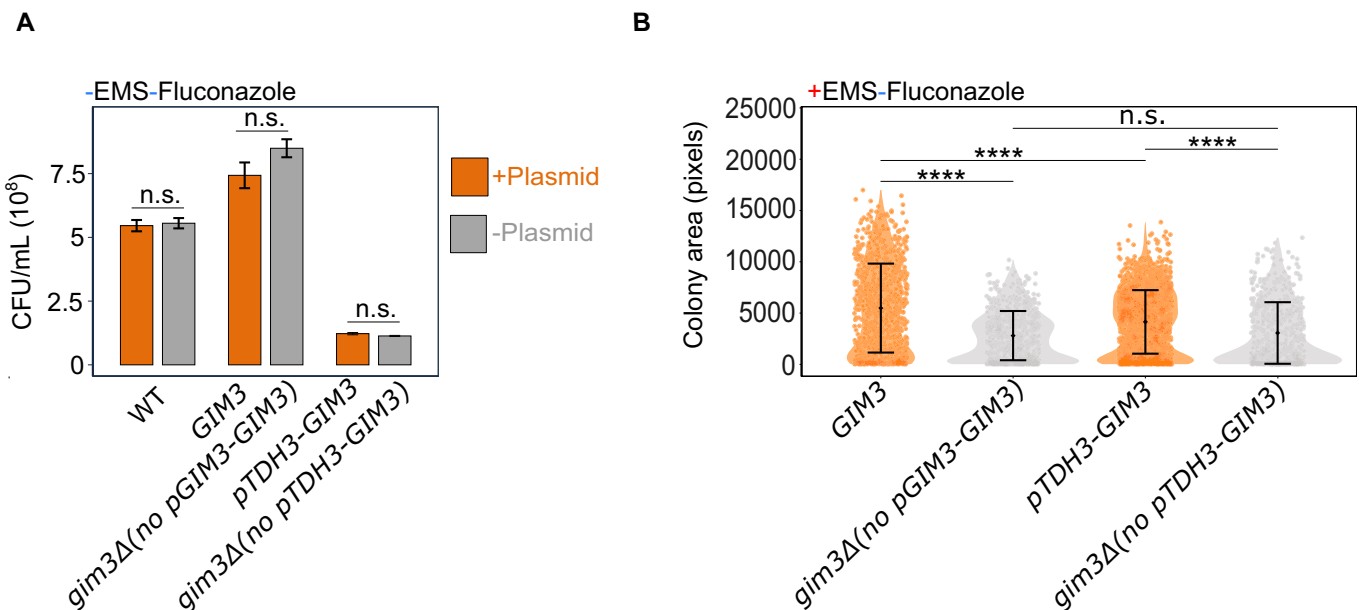

**Figure EV1.  Growth and colony size of yeast strains with and without GIM3 under non-stress conditions.**

(**A**) Growth comparison in non-stress conditions. Yeast strains (*S. cerevisiae* BY4741 background) with and without the *GIM3* gene deletion were grown in SC medium at 30 °C. Colony-forming units per milliliter (CFU/ml) were measured after incubation. Data represent the mean ± s.e.m. of $n = 10$ technical replicates. Statistical significance was assessed using a two-sided generalized linear model (GLM; family = quasi-Poisson, link = log). *P* values corresponding to the figure, from left to right, are 0.999, 0.120, and 0.994. (**B**) Colony size differences between EMS-treated yeast strains (*S. cerevisiae* BY4741 background) with and without the *GIM3* gene under non-stress conditions (SC medium at 30 °C). Error bars represent mean colony size ± s.e.m. of $n = 10$ technical replicates from a single experiment. Statistical significance was assessed using a two-sided pairwise Wilcoxon Rank-Sum Test. Exact *P* values *GIM3* vs. *Δgim3 (no pGIM3-GIM)* = $1.13 \times 10^{-28}$, *pTDH3_GIM3* vs. *Δgim3 (no pTDH3-GIM3)* = $5.04 \times 10^{-19}$, *GIM3* vs. *pTDH3_GIM3* = $3.20 \times 10^{-11}$, *Δgim3 (no pGIM3-GIM3) vs Δgim3 (no pTDH3-GIM3)* = 0.889. Statistical significance was considered at *P* value < 0.05. Exact genotypes of the strains used in this figure are provided in Table EV7. Significance thresholds: n.s. = not significant, *$P < 0.05$, ***$P < 0.001$, ****$P < 0.0001$. Source data are available online for this figure.

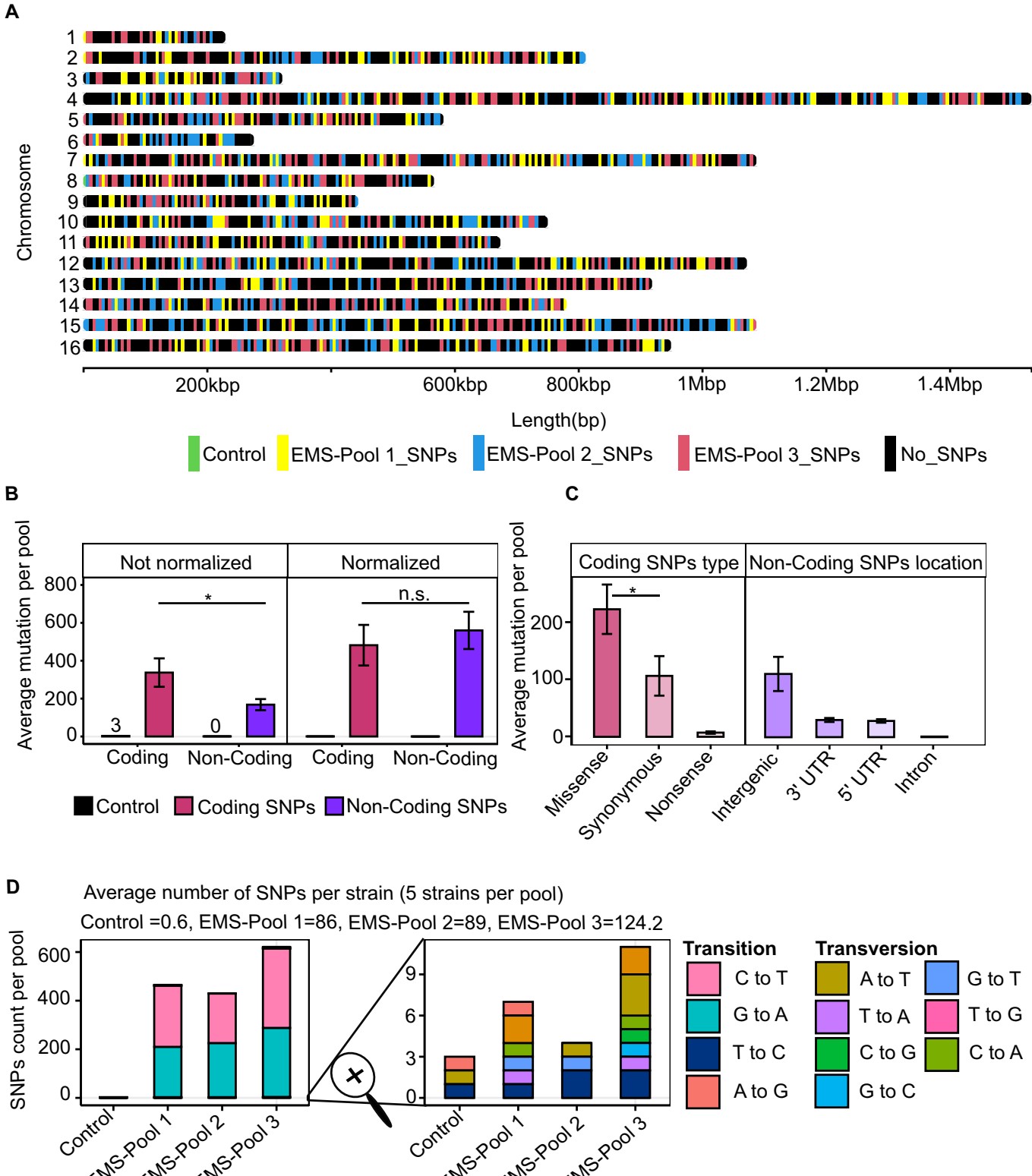

**Figure EV2. Comprehensive analysis of mutations induced by EMS random mutagenesis in *S. cerevisiae* BY4741.**

(A) Distribution of EMS-Induced Mutagenesis Across Chromosomes: Mutations are distributed across different chromosomes within the randomly mutagenized pool. Yellow, blue, and pink colors represent the mutations found in their respective pools, with each pool comprising 5 clones. Green indicates the control pool, where only 3 mutations were identified. Black represents the chromosomes themselves. All mutations shown are single-nucleotide polymorphisms (SNPs). (B) The average number of SNPs in coding vs. non-coding genome regions: Left: The average number of SNPs across all pools of EMS-treated clones, categorized by coding and non-coding regions of the genome. Right: The normalized average number of SNPs, adjusted for the coding (~70%) vs. non-coding (~30%) proportions of the BY4741 genome. A significant difference was observed in SNP counts between coding and non-coding regions before normalization (*t* test, *P* value = 0.045). However, after normalization to the genome proportions, no significant difference was found (*t* test, *P* value = 0.4), highlighting the randomness of EMS mutagenesis. Data represent the mean ± s.e.m. of *n* = 3 independent EMS-mutagenized pools. (C) Analysis of coding and non-coding mutations: Left: The average number of coding mutations across EMS-treated pools. Missense mutations are significantly higher than synonymous mutations (*t* test, *P* value = 0.021). Data represent the mean ± s.e.m. of *n* = 3 independent EMS-mutagenized pools. Right: The average number of non-coding SNPs across EMS-treated pools and their distribution across the genome. (D) Mutation types and frequency: The types of mutations observed in each pool, with a majority of C to T and G to A transitions. The left panel zooms in on other less common mutations observed at lower frequencies. On average, each strain within a pool exhibited between 86 and 124 mutations per genome. From the MT1 strain listed in Table EV7, we generated an untreated control pool and 3 independent EMS-mutagenized pool. Source data are available online for this figure.

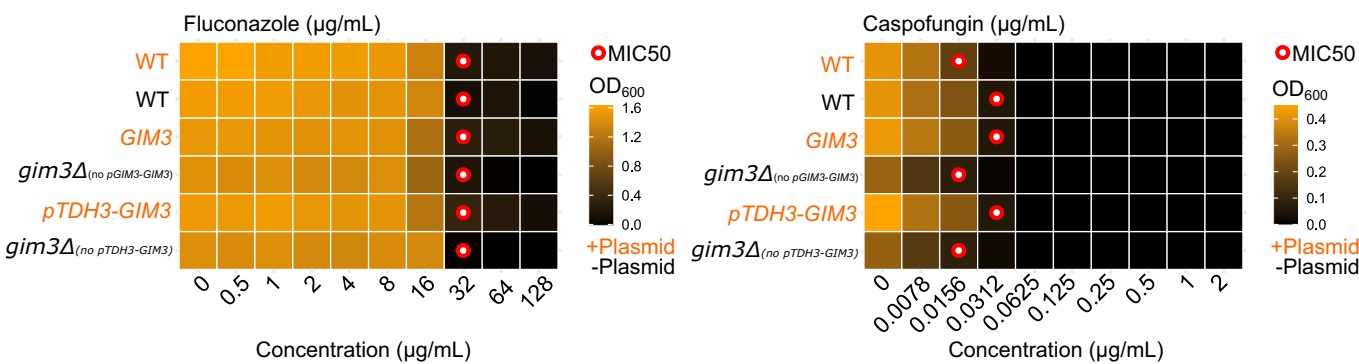

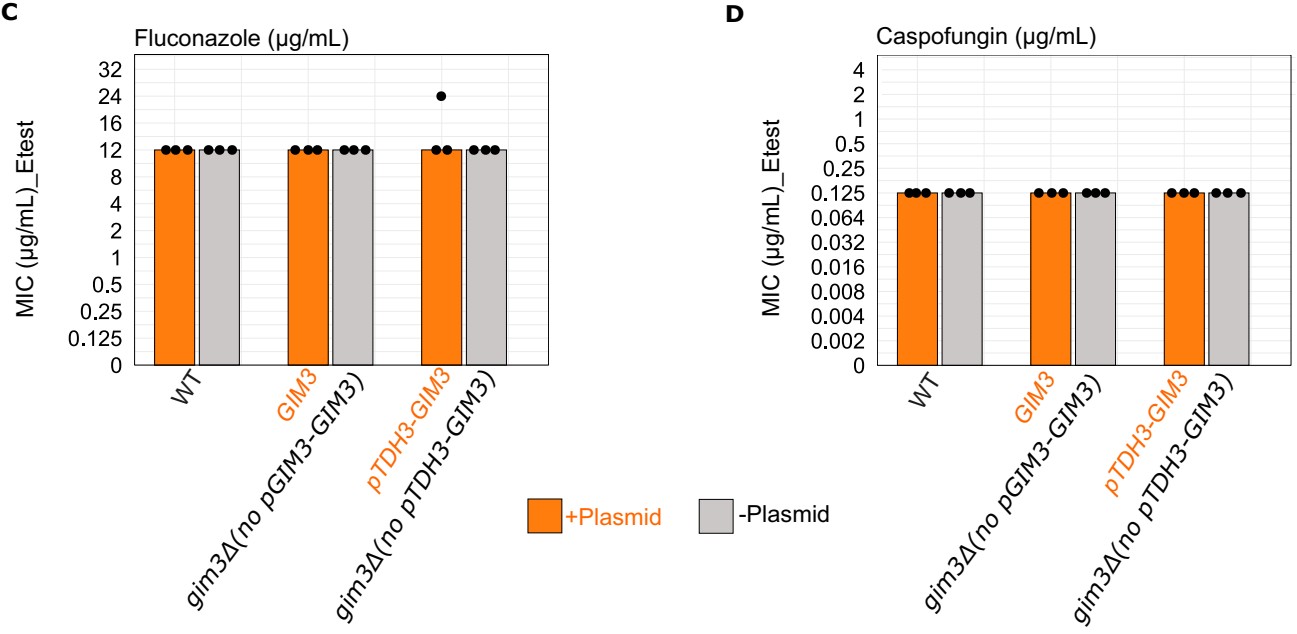

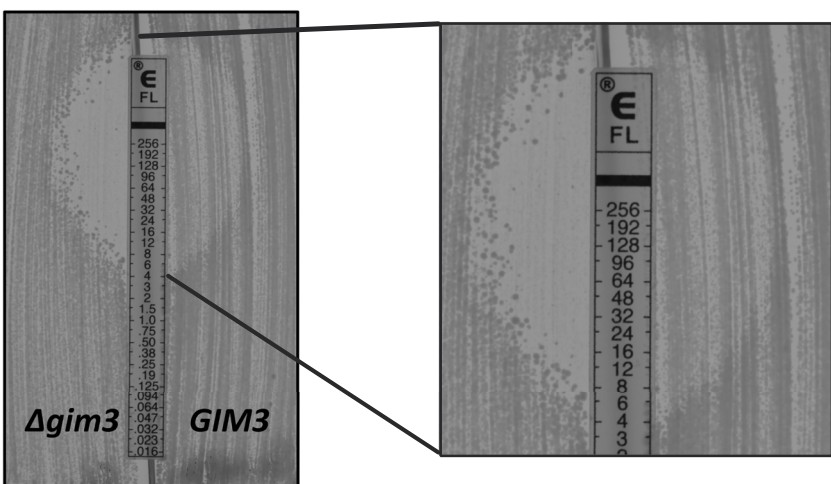

**Figure EV3. *GIM3* does not significantly alter fluconazole MIC$_{50}$ in *S. cerevisiae*.**

(A, B) Heatmaps showing OD$_{600}$ values from broth dilution assays (BDA) in the presence of increasing concentrations of fluconazole (**A**) and caspofungin (**B**) for wild-type (WT), *gim3Δ*, and *GIM3*-complemented strains (native promoter or *pTDH3-GIM3*). Orange indicates plasmid-containing strains (+ Plasmid), gray indicates plasmid-free strains (− Plasmid). Red circles mark MIC$_{50}$ values, defined as the lowest drug concentration that reduces OD$_{600}$ by ≥50% after 24 h at 30 °C. (**C, D**) MIC values determined by Etest for the same strains, shown for fluconazole (**C**) and caspofungin (**D**). Each bar represents the MIC of three biological replicates after 48 h incubation at 30 °C; black dots indicate individual replicates. Across both BDA and Etest assays, *GIM3* deletion does not significantly alter fluconazole MIC. In contrast, for caspofungin, a modest decrease in MIC$_{50}$ is observed in liquid BDA for *gim3Δ*, but not on solid media (Etest). (**E**) Etest MIC analysis of 35 EMS-derived strains identified as *GIM3*-dependent in Fig. 2B. Only 3 strains showed a higher MIC in the presence of *GIM3*, indicating *GIM3*-dependent resistance, while most strains had comparable MIC values with or without *GIM3*. Representative image is shown: the left panel shows a *gim3Δ* strain; the right panel depicts a strain carrying *GIM3*. Zone diameters (MIC) are similar, but residual growth ('trailing') within the zone is greater when *GIM3* is present. Complete MIC data are available in Table EV2. Exact genotypes of the strains used in this figure are provided in Table EV7. Source data are available online for this figure.

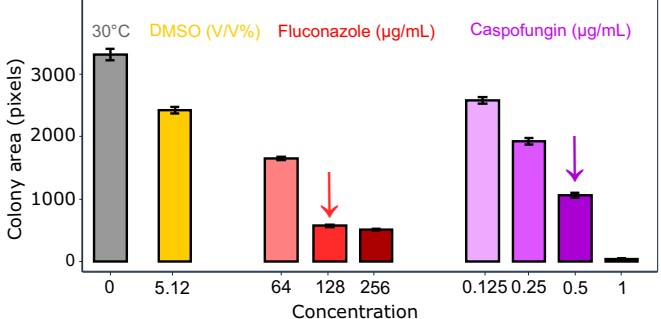

**Figure EV4. Bar plot showing colony area (pixels) of the unevolved *S. cerevisiae* WT strain (BY4741) under different stress conditions.**

Colony size was measured from plate images using the SGAtools online platform after 48 h of growth at 30 °C. Conditions include 0 µg/mL fluconazole (gray), 64, 128, and 256 µg/mL fluconazole (red gradient), 5.12% DMSO (yellow bar, used to dissolve 256 µg/mL fluconazole), and 0.125, 0.25, 0.5, and 1 µg/mL caspofungin (purple gradient). Bars represent the mean colony area from *n* = 16 technical replicates (colonies), black error bars indicate mean ± s.e.m. No statistical test was applied. Exact genotypes of the strains used in this figure are provided in Table EV7. Source data are available online for this figure.

