## [Peer Review File · EMBO Reports]

Gim3 buffers and potentiates de novo mutations that affect fluconazole susceptibility in yeast

Mohammed Tawfeeq, Dimitrios Konstantinidis, Ana Lucia Rocha Iraizos, Wouter Van Genechten, Jolien Vreys, Lieselotte Vermeersch, Karin Voordeckers, Kevin Verstrepen, and Patrick Van Dijck

Corresponding author(s): Kevin Verstrepen (kevin.verstrepen@kuleuven.be) , Patrick Van Dijck (patrick.vandijck@kuleuven.be)

Review Timeline:

Submission Date:	1st Oct 24
Editorial Decision:	18th Dec 24
Revision Received:	2nd Jun 25
Editorial Decision:	29th Jun 25
Revision Received:	24th Dec 25
Accepted:	20th Jan 26

Editor: Yehu Moran

Transaction Report:

Dear Prof. Verstrepen

Thank you for the submission of your manuscript to EMBO reports. We have now received the full set of referee reports as well as referee cross-comments that are all pasted below.

As you will see, the referees acknowledge that the findings are potentially interesting. However, they do raise comments and concerns that requires a significant revision before we can further consider your manuscript.

I would thus like to invite you to revise your manuscript with the understanding that the referee concerns must be fully addressed and their suggestions taken on board. Please address all referee concerns in a complete point-by-point response. Acceptance of the manuscript will depend on a positive outcome of a second round of review. It is EMBO Reports policy to allow a single round of major revision only and acceptance or rejection of the manuscript will therefore depend on the completeness of your responses included in the next, final version of the manuscript.

We realize that it is difficult to revise to a specific deadline. In the interest of protecting the conceptual advance provided by the work, we recommend a revision within 3 months (20th Mar 2025). Please discuss the revision progress ahead of this time with the editor if you require more time to complete the revisions.

- 1) A data availability section providing access to data deposited in public databases is missing. If you have not deposited any data, please add a sentence to the data availability section that explains that.
- 2) Your manuscript contains statistics and error bars based on $n=2$. Please use scatter blots in these cases. No statistics should be calculated if $n=2$.

<<https://www.embopress.org/page/journal/14693178/authorguide#expandedview>>

5) a complete author checklist, which you can download from our author guidelines

<<https://www.embopress.org/page/journal/14693178/authorguide>>. Please insert information in the checklist that is also reflected in the manuscript. The completed author checklist will also be part of the RPF.

6) Please note that all corresponding authors are required to supply an ORCID ID for their name upon submission of a revised manuscript (<<https://orcid.org/>>). Please find instructions on how to link your ORCID ID to your account in our manuscript tracking system in our Author guidelines

<<https://www.embopress.org/page/journal/14693178/authorguide#authorshipguidelines>>

10) Regarding data quantification (see Figure Legends:

<https://www.embopress.org/page/journal/14693178/authorguide#figureformat>)

12) All Materials and Methods need to be described in the main text using our 'Structured Methods' format, which is required for all research articles. According to this format, the Methods section includes a Reagents and Tools Table (listing key reagents, experimental models, software and relevant equipment and including their sources and relevant identifiers) followed by a Methods and Protocols section describing the methods using a step-by-step protocol format. The aim is to facilitate adoption of the methodologies across labs. More information on how to adhere to this format as well as a downloadable template (.docx) for the Reagents and Tools Table can be found in our author guidelines:

An example of a Method paper with Structured Methods can be found here: <https://www.embopress.org/doi/full/10.1038/s44320-024-00037-6#sec-4>

We would also welcome the submission of cover suggestions, or motifs to be used by our Graphics Illustrator in designing a

cover.

I look forward to seeing a revised form of your manuscript when it is ready.

Yours sincerely,

Yehu Moran
Academic Editor
EMBO Reports

Referee #1:

In this work the authors have tested the role of the chaperone Gim3 in modulating the phenotypic effects of fluconazole resistance mutations in the yeast *Saccharomyces cerevisiae*. They focused on Gim3, which is known to stabilize misfolded mutated proteins, because they had found in a previous study that it conferred mutational robustness to de novo mutations. Furthermore, Gim3 is known to interact with enzymes catalyzing ergosterol biosynthesis, which is inhibited by fluconazole, and mutations in *Erg3* are a known cause of resistance to this drug.

After chemical mutagenesis, a strain lacking Gim3 produced significantly fewer fluconazole-resistant mutants than control strains, and removing Gim3 from resistant strains resulted in reduced resistance in the majority of cases, demonstrating that Gim3 potentiates the effect of many resistance mutations. Sequencing of the *ERG3* gene showed that about half of the tested mutants contained mutations in this gene. Gim3 also affected the fitness of mutants that evolved fluconazole-resistance in the absence of chemical mutagenesis when passaged in the presence of a lower drug concentration, both positively and negatively, indicating that Gim3 can buffer as well as potentiate the effects of mutations. Some of the evolved strain also exhibited increased resistance to caspofungin, a drug with a different mode of action. Finally Gim3 was found to be required for the fluconazole resistance of *erg3Δ* mutants of the pathogenic yeast *Nakaseomyces glabratus* (*Candida glabrata*).

Some comments for the authors' consideration

- 1) Gim3 apparently did not buffer mutations that would prevent growth under nonselective conditions (no difference in CFUs of mutagenized strains with or without Gim3 was observed on drug-free medium, lines 194-196). Maybe the authors can briefly discuss this.
- 2) Gim3 might potentiate *Erg3* mutations by stabilizing the mutated proteins or by other mechanisms that compensate for altered *Erg3* activity. Apart from the in silico predictions (lines 295-300), the stability of tagged wild-type and mutated *Erg3* proteins in the presence and absence of Gim3 could be experimentally tested. The finding that Gim3 potentiated the fluconazole resistance caused by *ERG3* deletion in *N. glabratus* (lines 425-432) shows that mechanisms other than stabilizing mutated *Erg3* must be involved (Gim3 might actually destabilize the mutated *Erg3* proteins and promote their degradation [see line 123] to boost resistance). If Gim3 stabilizes mutated *Erg3* in *S. cerevisiae*, as suggested by the authors (lines 298-300, 492-494), the mechanism is not conserved between the two species.
- 3) Given the known role of *ERG3* mutations in fluconazole resistance and the established Gim3-*Erg3* interaction, it would be exciting to know which other mutations caused Gim3-dependent fluconazole resistance in the strains with wild-type *ERG3*. Did the authors consider whole-genome sequencing of individual clones (some of the 21 strains that did not contain *ERG3* mutations, lines 285-289) and testing candidate mutations? This would also be interesting for the evolved clones, some of which exhibited Gim3-dependent cross resistance to caspofungin (lines 391-394).
- 4) MIC values of fluconazole and caspofungin should be provided for all resistant strains and their *gim3Δ* derivatives, not only for the three starting strains (FIG EV1B). This will allow readers to compare the degree of resistance and its dependence on Gim3 in the strains described in the present study with the degree of resistance caused by known resistance mechanisms in medically important yeasts.
- 5) Lines 402 and 550: Should "77" not read "87" if 13 out of 100 populations (see line 343) got extinct? Or did 23 populations get

extinct?

6) Check the grammar of the sentence in lines 531-536.

Referee #2:

In this ms, the authors took a deep dive into the effect of GIM3, encoding for a protein chaperon, on a large number of de novo mutations related to fluconazole resistance in yeasts. They showed that for a large proportion of the mutations that leads to fluconazole resistance, the strength of the resistance phenotype depends on the presence of GIM3, demonstrating the effect of this chaperon as a phenotypic capacitor. These results nicely illustrate again the complexity underlying the genotype-phenotype in general, and offered a new potential target for antifungal treatment strategies that is of immediate interest. Specifically, the authors showed the effect of GIM3 is conserved in another yeast species, namely *Nakaseomyces glabratus*, a leading yeast pathogen in humans.

Overall, the ms is well written and the results are sound and clearly presented. Previous studies of these kind of phenotypic capacitors were mainly focused on Hsp90 and on standing variants. The current approach that the authors undertook directly test de novo mutations and nicely illustrated the extent to which such phenotypic capacitors can buffer the phenotypic impact of mutations linked to a specific trait immediately after selection. I believe this ms will be a nice addition to the literature that shows the non-linear and complex relationship between genotype and phenotype.

Major comments:

1. The authors focused on GIM3 because it was previously shown as one of the "buffer genes" that can modify the phenotypic outcome of novel mutations in yeast in a separate study by the same lab (<https://doi.org/10.1101/2024.09.24.614041>). In that study, it was shown that those buffer genes mainly belong to two functional groups: genes involved in protein folding or chromatin organization. While I understand the authors' motivation to go further with GIM3 due to the larger effect of this capacitor compared to the other candidates, I feel it is a bit of a pity because GIM3 is also a protein chaperon as the well characterized Hsp90, and it would be more or less expected that they would share the buffering capacity. Specifically, as the authors also mentioned, the relationship between Hsp90 and ERG3 mutations in fluconazole resistance has been previously reported, and it seems to be exactly like the effects the authors shown in this ms between GIM3 and the same mutation target ERG3, albeit the specific mutations are different. I think this potential redundancy should be discussed in the ms. Specifically, how unique is GIM3 buffering among all such chaperons? Do we expect that the potentiating effect to be different than Hsp90 and other buffer gene candidates in the same functional category?

2. The mechanism underlying phenotypic capacitors is always tricky to understand. In the present example, the authors suggested that GIM3 might stabilize Erg3 mutant proteins to maintain the resistance. However, it is counter intuitive to me why would mutants with frameshifts or premature stop codons will behave similarly compared to non-synonymous mutations that potentially destabilized the protein. In the same way, the data in *N. glabratus* showed deletion of ERG3 conferred resistance to fluconazole, and the resistance level is decreased with additional deletion of GIM3, and furthermore that particular interaction switched signs in higher concentrations of fluconazole (Figure 5). So if loss of function of ERG3 leads to resistance, then loss of GIM3 in that context should make the strain resistant but the result is the reverse. Can the authors clarify this point?

Referee #3:

The genetic linkage of the Gim3 and Erg3 is of interest and appears real but I am not convinced by the data that it is the way described through a chaperone. The gene deletion of *erg3* in *Candida glabrata* is without Erg3 protein so how can a chaperone have effect. The authors say it could be through something else and that should be revealed for EMBO Reports. I think including the frameshift and nonsense mutations that were not included would benefit the study as controls. Reference should occur to original publications so for fluconazole resistance and *erg3* as a cause this is from 1989 in BBRC for *Saccharomyces cerevisiae*. That this mechanism was found in *Candida albicans* was shown in isolates from AIDS patients in 1996 and 1997 (FEBS Lett). It was the first clinical mechanism of cross-resistance to different classes of drug that included amphotericin B so would have been interesting to include this drug. The method of measuring growth/ inhibition is usually by MIC testing and that would be easier to compare to the literature. There is also the known effect of fluconazole in causing petite mutations in both yeast species.

Dear Prof. Moran,

Thank you for sending us the reviewers' feedback. We are happy to read that the reviewers appreciated our work. Below, we address all questions and concerns raised by the reviewers (*in blue, in italics, in-between the reviewers' comments*) and highlight how we adapted our manuscript accordingly. As you can see, we have performed a considerable number of additional experiments suggested by the reviewers. In addition, we have reworked the text to address their concerns. Specifically, we have carried out the requested MIC assays, and in addition, we have also experimentally tested the stability of Erg3 proteins in the presence and absence of Gim3, using a cycloheximide-based protein degradation experiment, which is considered a standard in the field to look at protein stability. Our results reveal that, while a WT Erg3 protein is relatively stable over time in the absence of Gim3, mutant Erg3 displays drastically reduced stability in the absence of Gim3.

We have also addressed all editorial comments in our revised manuscript and reorganized our discussion to clarify some of the issues raised by the reviewers.

The revised manuscript is, in our opinion, much stronger, as it offers a deeper view of the exact role of *GIM3* as both a buffer and a potentiator. We wish to explicitly thank the reviewers for their constructive comments.

As to your question on the type of article, we prefer to submit our manuscript as 'Full Article'. We think this format suits our data best, allowing more than 5 figures as well as allowing us to clearly separate results from discussion.

Yours sincerely,

Kevin Verstrepen & Patrick Van Dijck, corresponding authors

COMMENTS to manuscript EMBOR-2024-60483V1 (*Gim3 buffers and potentiates de novo mutations that affect fluconazole resistance in yeast*)

Reviewer 1

In this work the authors have tested the role of the chaperone Gim3 in modulating the phenotypic effects of fluconazole resistance mutations in the yeast *Saccharomyces cerevisiae*. They focused on Gim3, which is known to stabilize misfolded mutated proteins, because they had found in a previous study that it conferred mutational robustness to de novo mutations. Furthermore, Gim3 is known to interact with enzymes catalyzing ergosterol biosynthesis, which is inhibited by fluconazole, and mutations in Erg3 are a known cause of resistance to this drug.

After chemical mutagenesis, a strain lacking Gim3 produced significantly fewer fluconazole-resistant mutants than control strains and removing Gim3 from resistant strains resulted in reduced resistance in the majority of cases, demonstrating that Gim3 potentiates the effect of many resistance mutations. Sequencing of the *ERG3* gene showed that about half of the tested mutants contained mutations in this gene. Gim3 also affected the fitness of mutants that evolved fluconazole-resistance in the absence of chemical mutagenesis when passaged in the presence of a lower drug concentration, both positively and negatively, indicating that Gim3 can buffer as well as potentiate the effects of mutations. Some of the evolved strain also exhibited increased resistance to caspofungin, a drug with a different mode of action. Finally, Gim3 was found to be required for the fluconazole resistance of *erg3Δ* mutants of the pathogenic yeast *Nakaseomyces glabratus* (*Candida glabrata*).

We thank the reviewer for his/her nice summary of our work.

Some comments for the authors' consideration

- 1) Gim3 apparently did not buffer mutations that would prevent growth under nonselective conditions (no difference in CFUs of mutagenized strains with or without Gim3 was observed on drug-free medium, lines 194-196). Maybe the authors can briefly discuss this.

This is a great comment! We indeed did not observe a difference in CFUs under nonselective conditions. Importantly, this does not necessarily imply that Gim3 is not buffering mutations in these conditions – we only expect a difference in CFUs if Gim3 specifically buffers mutations that would be lethal. In case Gim3 would buffer mutations that result in a (non-lethal) growth defect, we would still expect to see comparable number of CFUs in the presence and absence of Gim3; but varying colony sizes. Importantly, when we look at the colony size of EMS-treated cells in the absence of fluconazole, we see that cells without Gim3 show, on average, significantly reduced colony size, indicative of reduced fitness, compared to cells with Gim3.

B

Figure EV 1. (B) Colony size differences between EMS-treated yeast strains (*S. cerevisiae* BY4741 background) with and without the GIM3 gene under non-stress conditions (SC medium at 30 °C). Error bars represent mean colony size ± s.e.m. of 10 technical replicates from a single experiment. Statistical significance was assessed using a two-sided pairwise Wilcoxon Rank-Sum Test. Exact P-values GIM3 vs. $\Delta gim3 = 1.13 \times 10^{-28}$, pTDH3_GIM3 vs. $\Delta gim3 = 5.04 \times 10^{-19}$. Other comparisons are provided in **Table EV7**. Significance thresholds: n.s. = not significant, * $p < 0.05$, *** $p < 0.001$, **** $p < 0.0001$.

We have now added this data as **Supplementary Figure EV1B** and added the following text to our revised manuscript (lines 213-216):

“Notably, although we did not observe significant differences in CFU/mL on medium lacking fluconazole (**Fig. 1C**), the colony size of EMS-treated cells lacking GIM3 is on average significantly smaller than that of cells still containing GIM3 (**Fig. EV1B**). Since colony size is indicative of fitness, this indicates that Gim3 can also potentiate mutations in the absence of fluconazole.”

- 2) Gim3 might potentiate Erg3 mutations by stabilizing the mutated proteins or by other mechanisms that compensate for altered Erg3 activity. Apart from the in silico predictions (lines 295-300), the stability of tagged wild-type and mutated Erg3 proteins in the presence and absence of Gim3 could be experimentally tested.

Thank you for suggesting this experiment. We have now experimentally tested stability of Erg3 proteins in the presence and absence of Gim3, using a cycloheximide-based protein degradation assay, which is considered a standard in the field to look at protein stability. We have included the text below in our revised manuscript (lines 261-271):

“To investigate how Gim3 could potentiate the identified Erg3 non-synonymous mutations, we experimentally assessed stability of Erg3 proteins via a cycloheximide-based protein degradation assay (Fig. 4A). Specifically, we tested stability of a His-tagged Erg3^{T289P} in the presence and absence of Gim3, since this specific mutation was predicted to have the biggest destabilizing effect on Erg3 (see Fig. 3C, mutation A865C in ERG3). Our results reveal that, while a WT Erg3 protein in the absence of Gim3 is relatively stable over time, introducing the Erg3^{T289P} mutation drastically reduces Erg3 stability in the absence of Gim3 (Fig. 4B, one-way ANOVA, followed by Tukey’s HSD post-hoc test, p-value=0.029). Interestingly, this mutation is also located close to Erg3’s proposed active site (Jackson et al, 2003), so apart from altering Erg3’s stability, it might also result in altered activity. These results suggest that Gim3 might assist in stabilizing these mutant Erg3 proteins, potentially supporting a Gim3-dependent fluconazole resistance phenotype.”

Figure 4. Absence of Gim3 reduces Erg3^{T289P} stability. (A) Experimental setup for cycloheximide chase analysis to assess Erg3 stability over time. On the right is a schematic representation of the S.

cerevisiae strains constructed to assess the effect of GIM3 on ERG3 stability. Cells were incubated with cycloheximide (CHX) to stop translation, and samples were collected at the indicated timepoints after CHX addition. Protein extracts were prepared and Erg3-His levels were quantified using ELISA, and values were normalized based on total protein loading determined by Coomassie-stained gels. Figure created with BioRender.com. **(B)** Line plot showing the percentage decrease in Erg3-His relative to the initial concentration at time 0 for each strain background over time. Data represent mean \pm s.e.m. from three independent biological replicates per strain and time point. Statistical significance was assessed using one-way ANOVA at each time point, followed by Tukey's HSD post-hoc test, with correction for multiple comparisons. At the 120-minute timepoint, a significant decrease in Erg3-His stability in the presence of GIM3 was observed when compared to the non-mutant Erg3-His without GIM3 ($p=0.029$). While the decrease in Erg3^{T289P}-His stability in the absence of GIM3 compared to the stability of this protein in the presence of GIM3 was not statistically significant ($p=0.052$), the results also indicate reduced stability of Erg3^{T289P}-His. Colored asterisks, n.s. correspond to the strain used for comparison: orange asterisks indicate comparisons against the strain represented by the orange line, purple asterisks indicate comparisons against the strain represented by the purple line, and black asterisks represent comparisons between the strains represented by the orange and purple lines. n.s. =not significant, * $p < 0.05$, ** $p < 0.01$, *** $p < 0.001$, **** $p < 0.0001$.

We have now added these results to our revised manuscript as Figure 4.

The finding that Gim3 potentiated the fluconazole resistance caused by ERG3 deletion in *N. glabratus* (lines 425-432) shows that mechanisms other than stabilizing mutated Erg3 must be involved (Gim3 might actually destabilize the mutated Erg3 proteins and promote their degradation [see line 123] to boost resistance). If Gim3 stabilizes mutated Erg3 in *S. cerevisiae*, as suggested by the authors (lines 298-300, 492-494), the mechanism is not conserved between the two species.

*We agree with the reviewer that these results indicate that the mechanism by which Gim3 mediates fluconazole resistance might be different between the two species. Our results do indicate that GIM3 plays an important role in ERG3-associated resistance and tolerance to fluconazole in both *S. cerevisiae* and *N. glabratus*, although the exact molecular mechanisms appear to be different (at least partially).*

We have now adapted our manuscript accordingly to make this clearer. Examples include

In Results (lines 386-393):

*"In summary, the results demonstrate that GIM3 is necessary for the resistance conferred by ERG3 deletion in *N. glabratus*. The fact that removing GIM3 makes an *erg3Δ* strain less resistant, suggests that GIM3 could support resistance by facilitating alternative compensatory mechanisms, possibly through other sterol biosynthetic components such as ERG6 (Schuldiner et al, 2005; Costanzo et al, 2016). Reports suggest that ERG6 is upregulated in response to ERG3 deletion, possibly acting as a compensatory mechanism. It is therefore plausible that GIM3 contributes to the stability, folding, or function of these alternative sterol pathway proteins. Loss of GIM3 may impair these compensatory routes, explaining the drug sensitivity of *erg3Δ gim3Δ* double mutants."*

In Discussion (lines 519-529):

“Furthermore, our results indicate that GIM3 may also play a role in supporting fluconazole tolerance in N. glabratus. Deleting GIM3 in the clinically relevant species N. glabratus—specifically in erg3Δ fluconazole-resistant mutants—leads to a decreased resistance and effectively restores drug susceptibility. While GIM3 plays an important role in ERG3-associated resistance and tolerance to fluconazole in both S. cerevisiae and N. glabratus, the exact molecular mechanisms appear to be (at least partially) different in both species. In S. cerevisiae, GIM3-dependent fluconazole resistance appears to often involve mutations in the ERG3 gene, whereas in N. glabratus, deletion of GIM3 further reduces fluconazole resistance in an erg3Δ strain. This indicates that in N. glabratus, Gim3 has other targets, apart from Erg3, that are involved in fluconazole resistance. Given the limited availability of effective antifungal drugs and the rising incidence of multidrug-resistant N. glabratus infections, targeting GIM3 might offer a promising strategy to combat these pathogenic strains.”

- 3) Given the known role of ERG3 mutations in fluconazole resistance and the established Gim3-Erg3 interaction, it would be exciting to know which other mutations caused Gim3-dependent fluconazole resistance in the strains with wild-type ERG3. Did the authors consider whole-genome sequencing of individual clones (some of the 21 strains that did not contain ERG3 mutations, lines 285-289) and testing candidate mutations? This would also be interesting for the evolved clones, some of which exhibited Gim3-dependent cross resistance to caspofungin (lines 391-394).

We agree that this would indeed be exciting to know!

Whole-genome sequencing of EMS-treated populations revealed that EMS exposure introduces an average of ~86 to 124 mutations per genome, spread across the genome (Figure EV2). Unfortunately, this large number of mutations also implies that it is extremely difficult to pinpoint candidate causative mutations, even more so if we would not consider genes previously implicated in fluconazole resistance, as the reviewer is suggesting. Identifying candidate causative mutations would require a Quantative Trait Loci (QTL) analysis. This analysis involves crossing each strain with a fluconazole sensitive strain, analyzing segregants of this cross and creating different pools of segregants for whole-genome sequencing to be able to identify genetic regions associated with fluconazole resistance. Typically, genetic regions identified by QTL span multiple genes, requiring approaches such as allele replacement to identify the causative mutations. Such QTL approach typically takes 12-18 months to go from identifying potential candidate mutations to experimental validation of their causative nature. We feel this amount of work goes beyond what is reasonable for a revision.

Moreover, since the rapid selection regime we used is known to favor mutations in ERG3 (Cowen & Lindquist, 2005; Robbins & Cowen, 2021; Anderson et al, 2003), we think this provides sufficient motivation to focus on mutations in ERG3 and further explore their effects.

- 4) MIC values of fluconazole and caspofungin should be provided for all resistant strains and their gim3Δ derivatives, not only for the three starting strains (FIG EV1B). This will allow readers to compare the degree of resistance and its dependence on Gim3 in the strains described in the present study with the degree of resistance caused by known resistance mechanisms in medically important yeasts.

We thank the reviewer for this suggestion. We have now also performed MIC assays (MIC_{50} using BDA assays, as well as separate tests using E-strips) for fluconazole and caspofungin of the starting *S. cerevisiae* strains. These data are now included as **Figure EV3**.

We have also performed MIC assays (MIC_{50} using BDA assays, as well as separate tests using E-strips) for fluconazole on 35 EMS-treated strains that showed either a synthetic lethal phenotype or increased sensitivity to fluconazole (more than a 50% decrease in colony area) when *GIM3* was lost; and have included that data as **Figure EV3** and **Table EV2**. These data show that a few strains showed a higher MIC in the presence of *GIM3*, while in most cases, the MIC remains unchanged. However, we also find that *GIM3* deletion does affect tolerance (growth speed) for almost all mutant lineages, suggesting that apart from influencing the MIC (and thus resistance), *GIM3* seems to primarily act on tolerance.

Figure EV3. *GIM3* does not significantly alter fluconazole MIC_{50} in *S. cerevisiae*. (A, B) Heatmaps showing OD_{600} values from broth dilution assays (BDA) in the presence of increasing concentrations of fluconazole (A) and caspofungin (B) for wild-type (WT), *gim3\Delta*, and *GIM3*-complemented strains (native promoter or *pTDH3-GIM3*). Orange indicates plasmid-containing strains (+Plasmid), grey indicates plasmid-free strains (-Plasmid). Red circles mark MIC_{50} values, defined as the lowest drug concentration that reduces OD_{600} by $\geq 50\%$ after 24h at 30 °C. (C, D) MIC values determined by Etest for the same strains, shown for fluconazole (C) and caspofungin (D). Each bar represents the MIC of three biological replicates after 48h incubation at 30 °C; black dots indicate individual replicates.

Across both BDA and Etest assays, GIM3 deletion does not significantly alter fluconazole MIC. In contrast, for caspofungin, a modest decrease in MIC₅₀ is observed in liquid BDA for *gim3Δ*, but not on solid media (Etest). **(E)** Etest MIC analysis of 35 EMS-derived strains identified as GIM3-dependent in **Fig. 2B**. Only 5 strains showed a higher MIC in the presence of GIM3, indicating GIM3-dependent resistance, while most strains had comparable MIC values with or without GIM3. Representative images are shown: the left panel depicts a strain with increased MIC in the presence of GIM3; the right panel shows similar MIC values but reduced trailing growth after GIM3 removal, suggesting a role for GIM3 in fluconazole tolerance. Complete MIC data are available in **Table EV2**.

Table EV2. Fluconazole MICs by Etest for EMS-treated *S. cerevisiae* strains with and without GIM3

Strain	MIC_GIM3	MIC_Δgim3
EMS1	4	4
EMS2	8	8
EMS3	1.5	2
EMS4	3	2
EMS5	4	4
EMS6	4	4
EMS7	2	2
EMS8	16	16
EMS9	4	4
EMS10	4	4
EMS11	6	6
EMS12	8	8
EMS13	4	4
EMS14	4	4
EMS15	4	4
EMS16	6	8
EMS17	4	4
EMS18	6	8
EMS19	>256	2
EMS20	8	8
EMS21	16	4
EMS22	3	3
EMS23	6	6
EMS24	8	8
EMS25	8	8
EMS26	>256	4
EMS27	8	8
EMS28	3	2
EMS29	8	8
EMS30	8	8
EMS31	6	6
EMS32	48	3
EMS33	6	6
EMS34	4	4
EMS35	6	6

We did not perform caspofungin MIC assays for the 35 EMS-treated strains, since for these strains we did not previously demonstrate cross-resistance (this was for our evolved strains) and we feel this falls outside the scope of the current manuscript.

Following the request of Reviewer 3, we also performed MIC assays (MIC_{50} using BDA assays, as well as separate tests using E-strips) for the following:

- *N. glabratus* strains:
 - a. Strains Tested: WT, $\Delta erg3$, $\Delta gim3$, and $\Delta erg3\Delta gim3$
 - b. Antifungals: Fluconazole, Caspofungin, and Amphotericin B

These data are included as **Figure 6** (see also our response to Reviewer 3).

Figure 6. Deletion of *GIM3* affects antifungal resistance in the fungal pathogen *N. glabratus*. (A) Relative growth (%) of WT, *erg3Δ*, *gim3Δ*, and *erg3Δ gim3Δ* strains across increasing concentrations of fluconazole, measured using a broth dilution assay (BDA). All strains exhibit similar MIC_{50} values at 128 µg/mL; however, at 256 µg/mL, the *erg3Δ* strain maintains the highest MIC_{80} value, indicating increased resistance at higher drug concentrations. (B) Etest analysis at 37 °C (48h) shows a “halo” in regions of the medium with high fluconazole concentrations where cells are unable to grow. The images shown represent one biological replicate for WT, *gim3Δ* and two biological replicates for *erg3Δ*, and *erg3Δ gim3Δ*. Data from additional replicates, including MIC values, are available in **Table EV5**. While *erg3Δ* shows a lower MIC endpoint than WT and *gim3Δ*, it also displays more pronounced trailing growth, a phenotype reduced in the *erg3Δ gim3Δ* double mutant. (C) Relative growth (%) of WT, *erg3Δ*, *gim3Δ*, and *erg3Δ gim3Δ* strains across increasing concentrations of amphotericin B, measured using a BDA. At 0.25 µg/mL, *erg3Δ* shows significantly higher growth compared to the other backgrounds. (D) Relative growth (%) of WT, *erg3Δ*, *gim3Δ*, and *erg3Δ gim3Δ* strains across increasing concentrations of caspofungin, measured using a BDA. At 0.125 µg/mL, the *erg3Δ* strain

*shows the highest MIC₅₀ among all strains. All experiments were conducted with three biological replicates. Data are shown as mean ± S.E.M.. Red lines indicate the 50% relative growth threshold (MIC₅₀). Asterisks in the plots are shown only when the erg3Δ strain displays significantly higher growth than all other backgrounds, based on two-way ANOVA with Tukey's post hoc test. Full statistical test results for all pairwise comparisons are provided in Table EV6. ns = not significant; *p < 0.05; **p < 0.01; ***p < 0.001; ****p < 0.0001.*

5) Lines 402 and 550: Should "77" not read "87" if 13 out of 100 populations (see line 343) got extinct? Or did 23 populations get extinct?

Thank you for pointing this out. We have now corrected it to indicate that 23 populations went extinct.

6) Check the grammar of the sentence in lines 531-536.

We have now corrected this sentence.

Reviewer 2

In this ms, the authors took a deep dive into the effect of GIM3, encoding for a protein chaperon, on a large number of de novo mutations related to fluconazole resistance in yeasts. They showed that for a large proportion of the mutations that leads to fluconazole resistance, the strength of the resistance phenotype depends on the presence of GIM3, demonstrating the effect of this chaperon as a phenotypic capacitor. These results nicely illustrate again the complexity underlying the genotype-phenotype in general, and offered a new potential target for antifungal treatment strategies that is of immediate interest. Specifically, the authors showed the effect of GIM3 is conserved in another yeast species, namely *Nakaseomyces glabratus*, a leading yeast pathogen in humans.

Overall, the ms is well written and the results are sound and clearly presented. Previous studies of these kind of phenotypic capacitors were mainly focused on Hsp90 and on standing variants. The current approach that the authors undertook directly test de novo mutations and nicely illustrated the extent to which such phenotypic capacitors can buffer the phenotypic impact of mutations linked to a specific trait immediately after selection. I believe this ms will be a nice addition to the literature that shows the non-linear and complex relationship between genotype and phenotype.

We thank the reviewer for his/her positive comments.

Major comments:

1. The authors focused on GIM3 because it was previously shown as one of the "buffer genes" that can modify the phenotypic outcome of novel mutations in yeast in a separate study by the same lab (<https://doi.org/10.1101/2024.09.24.614041>). In that study, it was shown that those buffer genes mainly belong to two functional groups: genes involved in protein folding or chromatin organization. While I understand the authors' motivation to go further with GIM3 due to the larger effect of this capacitor compared to the other candidates, I feel it is a bit of a pity because GIM3 is also a protein chaperon as the well characterized Hsp90, and it would be more or less expected that they would share the buffering capacity. Specifically, as the authors also mentioned, the relationship between Hsp90 and ERG3 mutations in fluconazole resistance has been previously reported, and it seems to be exactly like the effects the authors shown in this ms between GIM3 and the same mutation target ERG3,

albeit the specific mutations are different. I think this potential redundancy should be discussed in the ms. Specifically, how unique is GIM3 buffering among all such chaperons? Do we expect that the potentiating effect to be different than Hsp90 and other buffer gene candidates in the same functional category?

These are great questions! We can understand that this reviewer is perhaps a bit disappointed that we focused on Gim3, since at first sight it might indeed look like we picked another protein chaperone to study buffering. However, our results show that Gim3 functions differently from the one other buffer gene (Hsp90) that has been previously characterized in detail.

Specifically:

- Our previous genome-wide screen (Frickel et al, 2024) revealed that not all chaperones are buffers: of the 63 genes annotated as chaperone in the Saccharomyces cerevisiae genome, only 7 were identified as potential buffering gene. Importantly, Gim3 showed the strongest effect in our screen, motivating us to further investigate Gim3.*
- Our results show that fluconazole resistance depends on Gim3 under both a rapid and gradual selection regime. This is not what is observed for Hsp90, which appears to be only required for fluconazole resistance acquired under a rapid selection regime (Cowen & Lindquist, 2005).*
- Hsp90 is a different type of chaperone than Gim3: Hsp90 is an ATP-dependent chaperone that assists in folding difficult-to-fold proteins and refolding denatured ones, whereas Gim3 is a subunit of the ATP-independent prefoldin complex, which stabilizes nascent and misfolded proteins, such as actin and tubulin, and delivers them to downstream chaperones for folding. Gim3 has also been reported to maintain the solubility of unstable proteins, facilitating their degradation via the proteasome (Comyn et al, 2016). Additionally, both chaperones are differentially expressed, suggesting that they may be active under distinct cellular conditions. This supports the idea that Hsp90 and Gim3 could buffer different sets of mutations or gene products depending on environmental or physiological context. Moreover, physical and genetic interactions between Gim3 and Hsp90 have been reported (Girstmair et al, 2019; Zhao et al, 2005), indicating that Gim3 may act within a broader chaperone-based buffering network, with potentially overlapping or complementary roles in maintaining proteostasis and mutational robustness.*

We have now clarified this in the discussion section of our revised manuscript (lines 404- 418). Specifically, the revised section is as follows:

“Importantly, while both Gim3 and Hsp90 are molecular chaperones, our previous genome-wide screen (Frickel et al, 2024) showed that not all chaperones act as genetic buffers: out of 63 genes annotated as chaperones in the S. cerevisiae genome (Gong et al, 2009), only 7 displayed significant buffering activity. Hsp90 and Gim3 differ in their mechanisms: Hsp90 is an ATP-dependent chaperone that assists in folding difficult-to-fold proteins and refolding denatured ones, whereas Gim3 is a subunit of the ATP-independent prefoldin complex, which stabilizes nascent and misfolded proteins, such as actin and tubulin, and delivers them to downstream chaperones for folding. Gim3 has also been reported to maintain the solubility of unstable proteins, facilitating their degradation via the proteasome (Comyn et al, 2016). Additionally, both chaperones are differentially expressed, suggesting that they may be active under distinct cellular conditions. This supports the idea that Hsp90 and Gim3 could buffer different sets of mutations or gene products depending on environmental or physiological context. Moreover, physical and genetic interactions between Gim3 and Hsp90 have been reported (Girstmair et al, 2019; Zhao et al, 2005), indicating that Gim3 may act

within a broader chaperone-based buffering network, with potentially overlapping or complementary roles in maintaining proteostasis and mutational robustness.”

Finally, the question on how unique Gim3 buffering is compared to Hsp90 buffering, buffering by other chaperones or even buffering by other buffer genes is a great question – but one that is unfortunately not straightforward to answer. In fact, we have received a similar question from reviewers evaluating our previous results (Frickel et al, 2024), and are currently performing experiments to address this question. Specifically, comparing Gim3 and Hsp90 with regards to their buffering capacity, requires evaluating which mutations (type of mutation, plus in what type of genes) these genes can buffer, with the same mutations tested for both genes. This requires us to generate a large library of mutations, and then assess the effect of removing Gim3 or Hsp90 on the phenotypic effects of these mutations. Together with our colleague prof. Sibylle Vonesch, we are currently using the CRISPR Cas9-based genome engineering method MAGESTIC to test the effect of candidate buffer genes on 10.000 mutations in parallel. We are as curious about these results as the reviewer! However, this really represents an enormous amount of work, which we feel goes beyond the scope of the current manuscript (and will hopefully be published in the future as part of another manuscript!)

2. The mechanism underlying phenotypic capacitors is always tricky to understand. In the present example, the authors suggested that GIM3 might stabilize Erg3 mutant proteins to maintain the resistance. However, it is counter intuitive to me why would mutants with frameshifts or premature stop codons will behave similarly compared to non-synonymous mutations that potentially destabilized the protein. In the same way, the data in *N. glabratus* showed deletion of ERG3 conferred resistance to fluconazole, and the resistance level is decreased with additional deletion of GIM3, and furthermore that particular interaction switched signs in higher concentrations of fluconazole (Figure 5). So if loss of function of ERG3 leads to resistance, then loss of GIM3 in that context should make the strain resistant but the result is the reverse. Can the authors clarify this point?

*We completely agree with the reviewer that the effect of Gim3 on Erg3 mutants with frameshift, premature stop codons and non-synonymous mutations are puzzling at first sight. Thanks to Reviewer 1 and Reviewer 2's comments, we realized we didn't properly explain the possible roles of Erg3 in fluconazole resistance. Fluconazole inhibits Erg11, leading to toxic intermediates in ergosterol biosynthesis, but mutations in Erg3 (as well as *erg3Δ*) prevent accumulation of these toxic intermediates, allowing growth in the presence of fluconazole*

*The majority of mutations in Erg3 are non-synonymous mutations (16 out of 23, ~70%), with several of these mutations predicted to have a destabilizing effect on Erg3. As outlined in our response to Reviewer 1, we have now also experimentally tested the effect of one of these mutations on Erg3 stability and could indeed confirm that stability of this Erg3 mutant is drastically reduced in the absence of Gim3. Hence, for non-synonymous ERG3 mutations, GIM3 may act to stabilize the mutant Erg3 proteins (some of which could also have altered sterol profiles), preventing their misfolding and thereby maintaining some degree of (altered) sterol biosynthesis that contributes to fluconazole resistance. For the remaining mutations (3 frameshift mutations and 4 premature stop codons), it is indeed more difficult to grasp how Gim3 could act mechanistically on these Erg3 mutants. Here possibly Gim3 could act as previously described by Comyn et al. (2016): Gim3 could promote degradation of these Erg3 mutants by maintaining their solubility and in this way facilitating proteasomal degradation (and thus mimicking an *erg3Δ* phenotype). Absence of Gim3 would in this*

case result in the accumulation of misfolded proteins, which could lead to membrane stress, indirectly influencing fluconazole resistance.

Another option that we currently cannot rule out is that in the strains containing *Erg3* mutants with frameshifts and premature stop codons, other (non-*Erg3*) *Gim3*-dependent mutations are responsible for the observed fluconazole resistance. We have now included this in the discussion of our revised manuscript. Specifically, the revised text reads (439-461):

“For non-synonymous Erg3 mutations, our results indicate that Gim3 may act to stabilize the mutant Erg3 proteins (some of which could also have altered sterol profiles), preventing their misfolding and thereby maintaining some degree of (altered) sterol biosynthesis that contributes to fluconazole resistance. Some fluconazole-resistant strains carrying non-synonymous ERG3 mutations have been described to maintain near-normal sterol profiles. These are known as ERG3 leaky mutants, as they retain sufficient function to reduce toxic intermediate build-up while still producing ergosterol. Such leaky mutants are typically seen in combination with other pump efflux mutations, which help to pump out the drug from the cell (Martel et al, 2010; Jackson et al, 2003). Gim3 interacts physically with Erg3 (Gong et al, 2009), and the stabilization of Erg3 mutants could explain their dependency on Gim3 for fluconazole resistance.

Nonsense mutations in ERG3, introducing premature stop codons, are well-documented under fluconazole-induced stress across species such as S. cerevisiae, Candida albicans, and N. glabratus (Cowen & Lindquist, 2005; Anderson et al, 2003; Wang et al, 2022; Martel et al, 2010; Robbins & Cowen, 2021). These mutations, as well as deletion of ERG3, inhibit toxic intermediate accumulation and alter sterol composition. For frameshift mutations and premature stop codons, it is more difficult to predict and understand how Gim3 could act mechanistically on these Erg3 mutants. Here, Gim3 could potentially act as previously described by Comyn et al. (2016): Gim3 could promote degradation of these Erg3 mutants by maintaining their solubility and in this way facilitating proteasomal degradation (and thus mimicking an erg3Δ mutant). Absence of Gim3 would in this case result in the accumulation of misfolded proteins, which could lead to membrane stress, indirectly influencing fluconazole resistance. Another option that we currently cannot rule out is that in the strains containing Erg3 mutants with frameshifts and premature stop codons, other (non-Erg3) Gim3-dependent mutations are responsible for the observed fluconazole resistance.”

In N. glabrata, we indeed see reduced fluconazole resistance in a Δerg3Δgim3 strain, which is difficult to explain if all Gim3 effects on fluconazole resistance depend on Erg3. These data indicate that Gim3 has other targets, apart from Erg3, that are involved in fluconazole resistance. For example, GIM3 has been shown to interact with other sterol biosynthetic enzymes, such as Erg6. Studies indicate that when ERG3 is deleted, ERG6 expression can be upregulated to compensate for the loss of Erg3 activity. In this context, it is plausible that GIM3 could modulate the stability, function or interaction dynamics of Erg6 or other sterol-related proteins. If GIM3 is deleted, such interactions may be disrupted, potentially affecting the compensatory sterol biosynthetic pathway and reducing resistance.

We acknowledge that our manuscript was unclear in presenting the ERG3 deletion data alongside the discussion of GIM3 interactions, potentially causing confusion. We have now revised this section to clearly distinguish between the observed effects of ERG3 deletion and the broader implications of GIM3 interactions with other genes involved in fluconazole resistance.

This revised discussion emphasizes that there may indeed be alternative pathways contributing to fluconazole resistance, given the network of interactions involving GIM3, which may function independently of or in addition to ERG3 disruptions.

Specifically, the revised text reads (lines 520-529):

*“Deleting GIM3 in the clinically relevant species *N. glabratus*—specifically in *erg3Δ* fluconazole-resistant mutants—leads to a decreased resistance and effectively restores drug susceptibility. While GIM3 plays an important role in ERG3-associated resistance and tolerance to fluconazole in both *S. cerevisiae* and *N. glabratus*, the exact molecular mechanisms appear to be (at least partially) different in both species. In *S. cerevisiae*, GIM3-dependent fluconazole resistance appears to often involve mutations in the ERG3 gene, whereas in *N. glabratus*, deletion of GIM3 further reduces fluconazole resistance in an *erg3Δ* strain. This indicates that in *N. glabratus*, Gim3 has other targets, apart from Erg3, that are involved in fluconazole resistance. Given the limited availability of effective antifungal drugs and the rising incidence of multidrug-resistant *N. glabratus* infections, targeting GIM3 might offer a promising strategy to combat these pathogenic strains.”*

Reviewer 3

The genetic linkage of the Gim3 and Erg3 is of interest and appears real but I am not convinced by the data that it is the way described through a chaperone. The gene deletion of *erg3* in *Candida glabrata* is without Erg3 protein so how can a chaperone have effect. The authors say it could be through something else and that should be revealed for EMBO Reports. I think including the frameshift and nonsense mutations that were not included would benefit the study as controls.

*We agree with Reviewer 3 (and the other reviewers) that we did not clearly explain and frame the *N. glabratus* results. Our data indeed indicates that Gim3 has other targets, apart from Erg3, that are involved in fluconazole resistance in *N. glabratus*. For example, GIM3 has been shown to interact with other sterol biosynthetic enzymes, such as Erg6 in *S. cerevisiae*. Studies indicate that when ERG3 is deleted, ERG6 expression can be upregulated to compensate for the loss of Erg3 activity. In this context, it is plausible that GIM3 could modulate the stability, function or interaction dynamics of Erg6 or other sterol-related proteins. If GIM3 is deleted, such interactions may be disrupted, potentially affecting the compensatory sterol biosynthetic pathway and reducing resistance.*

We acknowledge that our manuscript was unclear in presenting the ERG3 deletion data alongside the discussion of GIM3 interactions, potentially causing confusion. We have now revised this section to clearly distinguish between the observed effects of ERG3 deletion and the broader implications of GIM3 interactions with other genes involved in fluconazole resistance.

We have now included this in our revised manuscript. Specifically, the revised text reads (lines 386-393):

*“In summary, the results demonstrate that GIM3 is necessary for the resistance conferred by ERG3 deletion in *N. glabratus*. The fact that removing GIM3 makes an *erg3Δ* strain less resistant, suggests that GIM3 could support resistance by facilitating alternative compensatory mechanisms, possibly through other sterol biosynthetic components such as ERG6 (Schuldiner et al, 2005; Costanzo et al, 2016). Reports suggest that ERG6 is upregulated in response to ERG3 deletion, possibly acting as a compensatory mechanism. It is therefore plausible that GIM3 contributes to the stability, folding, or function of these alternative sterol pathway proteins. Loss of GIM3 may impair these compensatory routes, explaining the drug sensitivity of *erg3Δ gim3Δ* double mutants.”*

And (lines 520-529):

“Deleting GIM3 in the clinically relevant species N. glabratus—specifically in erg3Δ fluconazole-resistant mutants—leads to a decreased resistance and effectively restores drug susceptibility. While GIM3 plays an important role in ERG3-associated resistance and tolerance to fluconazole in both S. cerevisiae and N. glabratus, the exact molecular mechanisms appear to be (at least partially) different in both species. In S. cerevisiae, GIM3-dependent fluconazole resistance appears to often involve mutations in the ERG3 gene, whereas in N. glabratus, deletion of GIM3 further reduces fluconazole resistance in an erg3Δ strain. This indicates that in N. glabratus, Gim3 has other targets, apart from Erg3, that are involved in fluconazole resistance. Given the limited availability of effective antifungal drugs and the rising incidence of multidrug-resistant N. glabratus infections, targeting GIM3 might offer a promising strategy to combat these pathogenic strains.”

Reference should occur to original publications so for fluconazole resistance and erg3 as a cause this is from 1989 in BBRC for Saccharomyces cerevisiae. That this mechanism was found in Candida albicans was shown in isolates from AIDS patients in 1996 and 1997 (FEBS Lett). It was the first clinical mechanism of cross-resistance to different classes of drug that included amphotericin B so would have been interesting to include this drug.

We have now included these references in the text. (lines 101-104)

“Fluconazole inhibits Erg11, leading to toxic intermediates in ergosterol biosynthesis, but mutations in Erg3 (as well as erg3Δ) prevent the accumulation of these toxic intermediates, allowing growth in the presence of fluconazole (Watson et al, 1989).”

(lines 373-378)

“Since fluconazole-resistant strains of Candida albicans have been reported to display cross-resistance to amphotericin B (Kelly et al, 1997), we examined whether similar effects were seen in N. glabratus. Although erg3Δ strains did not show higher MIC₅₀ values, they did display enhanced survival at high amphotericin B concentrations, suggesting functional resistance not captured by standard MIC thresholds. Again, GIM3 deletion abolished this phenotype, lowering the MIC in the double mutant background (Fig. 6C, Table EV5).”

The method of measuring growth/ inhibition is usually by MIC testing and that would be easier to compare to the literature.

We have now included MIC assays (MIC₅₀ using BDA assays, as well as separate tests using E-strips) for the following:

- *N. glabratus strains:*
 - a. *Strains Tested: WT2001, Δerg3, Δgim3, and Δerg3Δgim3*
 - b. *Antifungals: Fluconazole, Caspofungin, and Amphotericin B*
- *S. cerevisiae strains:*
 - a. *Strains Tested: WT + plasmid, WT - plasmid, GIM3, Δgim3, pTDH3-GIM3, and pTDH3-GIM3 without plasmid*
 - b. *Antifungals: Fluconazole and Caspofungin*

- *EMS-Treated S. cerevisiae Mutants (MIC determined using E-strips):*
 - a. A subset of 35 EMS-treated mutant strains was selected based on the initial growth assays, where strains that lost the *GIM3* plasmid displayed significant inhibition or no growth under higher fluconazole concentrations.
 - b. For these selected strains, we performed MIC assays for fluconazole, to determine the specific impact of *GIM3* presence or absence on antifungal susceptibility.

The MIC data has been included as [Figure 6, Figure EV3 and Table EV2, Table EV5],

Figure 6. Deletion of *GIM3* affects antifungal resistance in the fungal pathogen *N. glabratus*. (A) Relative growth (%) of WT, *erg3Δ*, *gim3Δ*, and *erg3Δ gim3Δ* strains across increasing concentrations of fluconazole, measured using a broth dilution assay (BDA). All strains exhibit similar MIC₅₀ values at 128 µg/mL; however, at 256 µg/mL, the *erg3Δ* strain maintains the highest MIC₈₀ value, indicating increased resistance at higher drug concentrations. (B) Etest analysis at 37 °C (48h) shows a “halo” in regions of the medium with high fluconazole concentrations where cells are unable to grow. The images shown represent one biological replicate for WT, *gim3Δ* and two biological replicates for *erg3Δ*, and *erg3Δ gim3Δ*. Data from additional replicates, including MIC values, are available in Table EV5. While *erg3Δ* shows a lower MIC endpoint than WT and *gim3Δ*, it also displays more pronounced trailing growth, a phenotype reduced in the *erg3Δ gim3Δ* double mutant. (C) Relative growth (%) of WT, *erg3Δ*, *gim3Δ*, and *erg3Δ gim3Δ* strains across increasing concentrations of amphotericin B, measured using a BDA. At 0.25 µg/mL, *erg3Δ* shows significantly higher growth compared to the other backgrounds. (D) Relative growth (%) of WT, *erg3Δ*, *gim3Δ*, and *erg3Δ gim3Δ* strains across increasing concentrations of caspofungin, measured using a BDA. At 0.125 µg/mL, the *erg3Δ* strain shows the highest MIC₅₀ among all strains. All experiments were conducted with three biological replicates. Data are shown as mean ± S.E.M.. Red lines indicate the 50% relative growth threshold (MIC₅₀). Asterisks in the plots are shown only when the *erg3Δ* strain displays significantly higher

growth than all other backgrounds, based on two-way ANOVA with Tukey's post hoc test. Full statistical test results for all pairwise comparisons are provided in **Table EV6**. ns = not significant; * $p < 0.05$; ** $p < 0.01$; *** $p < 0.001$; **** $p < 0.0001$.

Figure EV3. GIM3 does not significantly alter fluconazole MIC₅₀ in *S. cerevisiae*. (A, B) Heatmaps showing OD₆₀₀ values from broth dilution assays (BDA) in the presence of increasing concentrations of fluconazole (A) and caspofungin (B) for wild-type (WT), *gim3Δ*, and GIM3-complemented strains (native promoter or pTDH3-GIM3). Orange indicates plasmid-containing strains (+Plasmid), grey indicates plasmid-free strains (-Plasmid). Red circles mark MIC₅₀ values, defined as the lowest drug concentration that reduces OD₆₀₀ by $\geq 50\%$ after 24h at 30 °C. (C, D) MIC values determined by Etest for the same strains, shown for fluconazole (C) and caspofungin (D). Each bar represents the MIC of three biological replicates after 48h incubation at 30 °C; black dots indicate individual replicates. Across both BDA and Etest assays, GIM3 deletion does not significantly alter fluconazole MIC. In contrast, for caspofungin, a modest decrease in MIC₅₀ is observed in liquid BDA for *gim3Δ*, but not on solid media (Etest). (E) Etest MIC analysis of 35 EMS-derived strains identified as GIM3-dependent in **Fig. 2B**. Only 5 strains showed a higher MIC in the presence of GIM3, indicating GIM3-dependent resistance, while most strains had comparable MIC values with or without GIM3. Representative

images are shown: the left panel depicts a strain with increased MIC in the presence of GIM3; the right panel shows similar MIC values but reduced trailing growth after GIM3 removal, suggesting a role for GIM3 in fluconazole tolerance. Complete MIC data are available in **Table EV2**.

Table EV2. Fluconazole MICs by Etest for EMS-treated *S. cerevisiae* strains with and without GIM3

Strain	MIC_GIM3	MIC_Δgim3
EMS1	4	4
EMS2	8	8
EMS3	1.5	2
EMS4	3	2
EMS5	4	4
EMS6	4	4
EMS7	2	2
EMS8	16	16
EMS9	4	4
EMS10	4	4
EMS11	6	6
EMS12	8	8
EMS13	4	4
EMS14	4	4
EMS15	4	4
EMS16	6	8
EMS17	4	4
EMS18	6	8
EMS19	>256	2
EMS20	8	8
EMS21	16	4
EMS22	3	3
EMS23	6	6
EMS24	8	8
EMS25	8	8
EMS26	>256	4
EMS27	8	8
EMS28	3	2
EMS29	8	8
EMS30	8	8
EMS31	6	6
EMS32	48	3
EMS33	6	6
EMS34	4	4
EMS35	6	6

Table EV5. MICs by E-test for *N. glabratus* strains (WT 2001, Δgim3, Δerg3, and Δgim3Δerg3).

Antifungal ($\mu\text{g/mL}$)	Replicate 1	Replicate 2	Replicate 3	Strains
Fluconazole	8	8	8	WT 2001
Fluconazole	6	6	6	Δgim3
Fluconazole	2	2	2	Δerg3
Fluconazole	2	2	2	$\Delta\text{gim3}\Delta\text{erg3}$
Amphotericin B	0,125	0,125	0,125	WT 2001
Amphotericin B	0,032	0,032	0,032	Δgim3
Amphotericin B	0,125	0,125	0,125	Δerg3
Amphotericin B	0,064	0,064	0,064	$\Delta\text{gim3}\Delta\text{erg3}$
Caspofungin	0,125	0,125	0,125	WT 2001
Caspofungin	0,125	0,125	0,125	Δgim3
Caspofungin	0,25	0,25	0,25	Δerg3
Caspofungin	0,25	0,25	0,25	$\Delta\text{gim3}\Delta\text{erg3}$

There is also the known effect of fluconazole in causing petite mutations in both yeast species.

We thank the reviewer for pointing this out. Petite mutations in yeast are typically found in the mitochondrial genome, or in genes related to mitochondrial function and result in respiratory deficiency. Importantly, these mutations also result in smaller colonies on medium containing glucose, since the respiratory defect leads to substantially reduced ATP yield when growing on glucose (Day, 2013; Baruffini et al, 2007; Hess et al, 2009). While we never specifically assessed petite formation, we performed all our experiments on glucose-containing medium, and never observed small colonies on our glucose-containing agar plates. This indicates that under our experimental conditions, no petites were formed.

References

- Anderson JB, Sirjusingh C, Parsons AB, Boone C, Wickens C, Cowen LE & Kohn LM (2003) Mode of Selection and Experimental Evolution of Antifungal Drug Resistance in *Saccharomyces cerevisiae*. *Genetics* 163: 1287–1298
- Baruffini E, Lodi T, Dallabona C & Foury F (2007) A Single Nucleotide Polymorphism in the DNA Polymerase Gamma Gene of *Saccharomyces cerevisiae* Laboratory Strains Is Responsible for Increased Mitochondrial DNA Mutability. *Genetics* 177: 1227–1231
- Comyn SA, Young BP, Loewen CJ & Mayor T (2016) Prefoldin Promotes Proteasomal Degradation of Cytosolic Proteins with Missense Mutations by Maintaining Substrate Solubility. *PLoS Genet* 12: e1006184
- Costanzo M, VanderSluis B, Koch EN, Baryshnikova A, Pons C, Tan G, Wang W, Usaj M, Hanchard J, Lee SD, et al (2016) A global genetic interaction network maps a wiring diagram of cellular function. *Science (1979)* 353: aaf1420
- Cowen LE & Lindquist S (2005) Cell biology: Hsp90 potentiates the rapid evolution of new traits: Drug resistance in diverse fungi. *Science (1979)* 309: 2185–2189
- Day M (2013) Yeast Petites and Small Colony Variants: For Everything There Is a Season. *Adv Appl Microbiol* 85: 1–41

- Frickel J, Tawfeeq MT, Baker E, Baco S, Rombout J, Jarosz DF, Vonesch SC, Verstrepen KJ, Leuven K & Geenslaan G (2024) Genes involved in protein folding and chromatin organization buffer genetic variation. *bioRxiv*: 2024.09.24.614041
- Girstmair H, Tippel F, Lopez A, Tych K, Stein F, Haberkant P, Schmid PWN, Helm D, Rief M, Sattler M, *et al* (2019) The Hsp90 isoforms from *S. cerevisiae* differ in structure, function and client range. *Nat Commun* 10: 1–15
- Gong Y, Kakahara Y, Krogan N, Greenblatt J, Emili A, Zhang Z & Houry WA (2009) An atlas of chaperone–protein interactions in *Saccharomyces cerevisiae*: implications to protein folding pathways in the cell. *Mol Syst Biol* 5: 275
- Hess DC, Myers C, Huttenhower C, Hibbs MA, Hayes AP, Paw J, Clore JJ, Mendoza RM, Luis BS, Nislow C, *et al* (2009) Computationally Driven, Quantitative Experiments Discover Genes Required for Mitochondrial Biogenesis. *PLoS Genet* 5: e1000407
- Jackson CJ, Lamb DC, Manning NJ, Kelly DE & Kelly SL (2003) Mutations in *Saccharomyces cerevisiae* sterol C5-desaturase conferring resistance to the CYP51 inhibitor fluconazole. *Biochem Biophys Res Commun* 309: 999–1004
- Kelly SL, Lamb DC, Kelly DE, Manning NJ, Loeffler J, Hebart H, Schumacher U & Einsele H (1997) Resistance to fluconazole and cross-resistance to amphotericin B in *Candida albicans* from AIDS patients caused by defective sterol $\Delta 5,6$ -desaturation. *FEBS Lett* 400: 80–82
- Martel CM, Parker JE, Bader O, Weig M, Gross U, Warrilow AGS, Rolley N, Kelly DE & Kelly SL (2010) Identification and Characterization of Four Azole-Resistant *erg3* Mutants of *Candida albicans*. *Antimicrob Agents Chemother* 54: 4527–4533
- Robbins N & Cowen LE (2021) Antifungal drug resistance: Deciphering the mechanisms governing multidrug resistance in the fungal pathogen *Candida glabrata*. *Current Biology* 31: R1520–R1523
- Schuldiner M, Collins SR, Thompson NJ, Denic V, Bhamidipati A, Punna T, Ihmels J, Andrews B, Boone C, Greenblatt JF, *et al* (2005) Exploration of the function and organization of the yeast early secretory pathway through an epistatic miniarray profile. *Cell* 123: 507–519
- Wang WY, Cai HQ, Qu SY, Lin WH, Liang CC, Liu H, Xie ZX & Yuan YJ (2022) Genomic Variation-Mediating Fluconazole Resistance in Yeast. *Biomolecules* 12: 845
- Watson PF, Rose ME, Ellis SW, England H & Kelly SL (1989) Defective sterol C5-6 desaturation and azole resistance: A new hypothesis for the mode of action of azole antifungals. *Biochem Biophys Res Commun* 164: 1170–1175
- Zhao R, Davey M, Hsu YC, Kaplanek P, Tong A, Parsons AB, Krogan N, Cagney G, Mai D, Greenblatt J, *et al* (2005) Navigating the Chaperone Network: An Integrative Map of Physical and Genetic Interactions Mediated by the Hsp90 Chaperone. *Cell* 120: 715–727

Dear Prof. Verstrepen

Thank you for the submission of your revised manuscript to EMBO Reports. We have now received the full set of referee reports that are all pasted below.

As you will see, one of the reviewers (Referee #1) remains concerned regarding some of the results.

I would thus like to invite you to revise your manuscript with the understanding that the referee concerns must be fully addressed and their suggestions taken on board. Please address the referee concerns in a complete point-by-point response. Acceptance of the manuscript will depend on a positive outcome of the next (and final) round of review. While it is EMBO Reports official policy to allow only a single round of major revision, we do sometimes make exceptions for papers like yours that we find exceptionally interesting and allow another round of major revision. Yet, please note that if Referee #1 will remain concerned/unconvinced in this round, we may be forced to reject your paper.

We realize that it is difficult to revise to a specific deadline. In the interest of protecting the conceptual advance provided by the work, we recommend a revision within 3 months (29th Sep 2025). Please discuss the revision progress ahead of this time with the editor if you require more time to complete the revisions.

Please also note the comments by editorial assistants' team that I include below. While those are mostly technical in their nature and you do not need to include them in your point-by-point response, they are still extremely important and require your attention during revision.

- 1) A data availability section providing access to data deposited in public databases is missing. If you have not deposited any data, please add a sentence to the data availability section that explains that.
- 2) Your manuscript contains statistics and error bars based on $n=2$. Please use scatter blots in these cases. No statistics should be calculated if $n=2$.

5) a complete author checklist, which you can download from our author guidelines

<<https://www.embopress.org/page/journal/14693178/authorguide>>. Please insert information in the checklist that is also reflected in the manuscript. The completed author checklist will also be part of the RPF.

6) Please note that all corresponding authors are required to supply an ORCID ID for their name upon submission of a revised manuscript (<<https://orcid.org/>>). Please find instructions on how to link your ORCID ID to your account in our manuscript tracking system in our Author guidelines

<<https://www.embopress.org/page/journal/14693178/authorguide#authorshipguidelines>>

10) Regarding data quantification (see Figure Legends:

<https://www.embopress.org/page/journal/14693178/authorguide#figureformat>)

- the name of the statistical test used to generate error bars and P values,

- the number (n) of independent experiments (please specify technical or biological replicates) underlying each data point,

- the nature of the bars and error bars (s.d., s.e.m.),

- If the data are obtained from n Program fragment delivered error ``Can't locate object method "less" via package "than" (perhaps you forgot to load "than"?) at //ejpvfs23/sites23b/embor_www/letters/embor_decision_revise_and_review.txt line 56.' 2, use scatter blots showing the individual data points.

12) All Materials and Methods need to be described in the main text using our 'Structured Methods' format, which is required for all research articles. According to this format, the Methods section includes a Reagents and Tools Table (listing key reagents, experimental models, software and relevant equipment and including their sources and relevant identifiers) followed by a Methods and Protocols section describing the methods using a step-by-step protocol format. The aim is to facilitate adoption of the methodologies across labs. More information on how to adhere to this format as well as a downloadable template (.docx) for the Reagents and Tools Table can be found in our author guidelines:

An example of a Method paper with Structured Methods can be found here: <https://www.embopress.org/doi/full/10.1038/s44320-024-00037-6#sec-4>

I look forward to seeing a revised form of your manuscript when it is ready.

Yours sincerely,

Yehu Moran
Editor
EMBO Reports

Specific comments by editorial assistant:

CHARACTER COUNT: 48,398; 6 figures; R&D not combined

Data Availability Statement: in, but needs to be placed before Acknowledgments

Conflict of interest statement: included, but it needs to be renamed to Disclosure and Competing Interests Statement

AC/CRedit: needs to be removed from the ms. It should be included only in the submission system form and not in the text itself.

FUNDING INFO: missing in our system, please complete - FWO grant I011820N

FIGURE CALLOUTS: Table EV8 is missing a callout in the manuscript text.

DATASET EV LEGENDS: 8 EV tables uploaded; Table EV4 and Table EV8 - are these datasets? (they look more complex) if yes, then these should be updated to Dataset EV1 and Dataset EV2; the table legends need to be removed from the manuscript.

SYNOPSIS IMAGE: missing, please provide.

SYNOPSIS TEXT: missing, please provide

Additional note:

- Methods and Protocols should be renamed Methods

Figure Legends - Comments

- Please note that the exact p values are not provided in the legends of figures 1D, 4B, 5B, 6A, C, D; EV1 B, EV2 B, C. Please provide.

- Please note that the box plots need to be defined in terms of minima, maxima, centre, bounds of box and whiskers, and percentile in the legends of figure 5B

- Please note that the box plots need to be defined in terms of minima, maxima, bounds of box and whiskers, and percentile in the legends of figure 2B.

- Please note that information related to n is missing in the legends of figures 5B, C, D, E, F. Please provide.

Referee #1:

In their revised manuscript the authors have partially addressed my previous criticisms, but some critiques remain.

1) In one of my previous comments I had suggested to experimentally test the stability of the mutated Erg3 proteins in the presence and absence of GIM3 and thereby verify the in silico predictions and the authors' conclusion that Gim3 potentiates fluconazole resistance by stabilizing the mutated proteins. The authors tested only one of the mutated proteins and the wild type Erg3 instead of a panel of proteins with different predicted stabilities and different dependencies on Gim3 for conferring fluconazole tolerance. The experiment shown in Fig. 4 also lacks an important control, the wild-type strain with wild-type Erg3, which would show the importance of Gim3 for the stability of wild-type Erg3. Furthermore, protein levels were determined at only two time points after translation inhibition, with highly variable values for biological replicates, instead of determining the half-life of the different proteins. The legend to the figure is highly confusing. It describes asterisks with different colors, but only one

purple asterisk is shown in Fig. 4B. On top of it, the legend erroneously states (lines 1171-1173) that the stability of (the mutated) Erg3 in the presence of GIM3 (orange line) was significantly decreased compared with that of wild-type Erg3 in the absence of GIM3 (purple line), although the difference was between the mutant Erg3 in the absence (grey line) and presence (purple line) of GIM3 (purple asterisk), as also stated in the text (lines 268-271).

2) As noted previously, given the known role of ERG3 mutations in fluconazole resistance and the established Gim3-Erg3 interaction, it would have been more exciting to know which other mutations caused Gim3-dependent fluconazole resistance in the strains with wild-type ERG3. In their rebuttal letter the authors argue that identifying responsible mutations would require an amount of work that goes beyond what is reasonable for a revision, which is understandable. However, in that case I would expect a deeper investigation of how Gim3 potentiates the effect of Erg3 mutations on fluconazole tolerance. It is thought that loss or reduced activity of Erg3 prevents the accumulation of toxic sterols that inhibit growth in the presence of the drug. If Gim3 stabilizes the mutated Erg3 proteins, as the authors conclude from the results with one mutated protein, one should a priori expect that this increases the levels of toxic sterols and actually reduces drug tolerance. Stabilization of the mutated Erg3 proteins might therefore be required to enable sufficient ergosterol synthesis to occur without the production of toxic sterols in the presence of fluconazole. To understand the effect of the various Erg3 mutations on fluconazole tolerance and its dependence on Gim3, the different mutations should be introduced into isogenic wild-type and *gim3Δ* backgrounds, since most of the evolved strains contain additional mutations that affect fluconazole susceptibility, as is evident from the new Table EV2. The stability of the proteins and the fluconazole tolerance of the strains should then be compared. Furthermore, sterol profiles of the strains in the presence and absence of fluconazole should be determined to see how the mutations and the presence of Gim3 affect the accumulation of toxic sterols.

3) Considering that ERG3 loss-of-function mutations are reported to result in fluconazole resistance in *S. cerevisiae*, it is surprising that the frameshift and nonsense mutations (note that nonsense mutations are not usually termed frameshift mutations, line 258-259), which would be expected to prevent production of a functional Erg3 protein, did not result in increased MICs compared to wild-type MIC (12 µg/ml, see Fig. EV3) and most of them even were associated with increased sensitivity (Table EV2). The new MIC data shown in Table EV2 and the phenotypes shown elsewhere demonstrate that Gim3 does not affect fluconazole resistance and instead enables residual growth in the presence of the drug, which is often referred to as trailing growth or tolerance. Since the authors emphasize the relevance of their results for clinical drug resistance, the term resistance should be used carefully. Was the trailing growth of the evolved strains and its loss after removal of GIM3 also observed in broth microdilution assays when growth was assayed after prolonged incubation (48 h instead of 24 h, which is often used to distinguish tolerance and resistance)?

Some editorial comments and suggestions:

- Clone 17 seems to be misnumbered as clone 14 in Fig. 3A.

- Tables EV2 and EV3 could be combined and the wild-type strain should be included for comparison of the MICs. The fluconazole MIC for the wild-type (12 µg/ml) can only be found in Fig. EV3. Including it in Table EV2 will highlight that most evolved fluconazole-"resistant" clones actually exhibit increased sensitivity and display only increased tolerance.

- Which are the 5 strains with a higher MIC in the presence of GIM3 versus its absence mentioned in lines 240-241? Table EV2 contains 4 strains with at least 4-fold higher MICs (EMS19, 21, 26, and 32) and two additional strains with a marginally higher MIC in the presence of GIM3 (3 µg/ml versus 2 µg/ml, EMS4 and 28).

4) In line 370 the authors state that "deletion of ERG3 in *N. glabratus* resulted in a higher MIC₈₀ value 256 µg/mL compared to WT", but the corresponding Fig. 6A uses MIC₅₀ as threshold and shows that this is the same for both strains (256 µg/mL). While different tests can give somewhat different MIC values, the drastic differences between broth microdilution (256 µg/mL for all strains) and Etest (8 µg/mL for the wild type and 2 µg/mL for *erg3Δ* mutants and *erg3Δ gim3Δ* double mutants) in this study are astonishing. Regardless, the results shown in Fig. 6A and B indicate that ERG3 deletion actually reduces fluconazole resistance in the *N. glabratus* strain used in this study (at least in the Etest) but increases tolerance (trailing growth at fluconazole concentrations above the MIC). Furthermore, MIC levels of *erg3Δ* mutants remain unchanged in the absence of Gim3, but Gim3 is required for the increased fluconazole tolerance of *erg3Δ* mutants. The authors should modify their conclusions accordingly and correctly describe the results. Examples:

- Line 375 "...*erg3Δ* did not show an increased MIC (Etest MIC = 2)...". In fact, the MIC was reduced by 4-fold compared to the wild type (2 µg/mL versus 8 µg/mL).

- Line 381 "Although *erg3Δ* strains did not show higher MIC₅₀ values, they did display enhanced survival at high amphotericin B concentrations". Fig. 6C shows growth, not survival.

- Line 387: "In contrast to the fluconazole and amphotericin B results, GIM3 deletion in the *erg3Δ* background did not lead to a measurable change in MIC by E-test". Table EV5 shows that fluconazole MICs also remained unchanged.

In my comments to the original manuscript I had addressed only some main points, but the following issues should also be clarified:

5) One thing that I failed to notice in the original manuscript: Why did the authors not introduce an empty vector into the *gim3Δ* mutants to complement their uracil auxotrophy, as in the strains in which GIM3 was reintroduced on a plasmid (see lines 182-186)? This would have been the appropriate control. Could uracil auxotrophy explain the now reported smaller colony size of EMS-treated *gim3Δ* mutants compared to EMS-treated cells containing GIM3 (lines 216-218)? The authors do not explicitly state that such a difference was not observed in the absence of EMS treatment.

6) The *gim3Δ* mutant was constructed only once from the wild-type strain BY4741 (lines 551-554). What is the difference between the two *gim3Δ* mutants in Fig. EV1 and why did one of them produce more colonies than the other in Fig. EV1A? The same applies to Fig. 1B-D and Fig. EV3. As I understood, these are not the *gim3Δ* mutants that were later derived from the plasmid-bearing strains by FOA treatment (lines 221-225). A table listing all strains with specific names, genotypes, and parents should be added, and the legends to the figures should explicitly state the names of the strains that were used in each particular experiment.

7) If I misunderstood the text and the two *gim3Δ* mutants in Fig. EV1 are the plasmid-cured derivatives of the strains containing GIM3 on a plasmid (expressed from its own or the TDH3 promoter), I still not understand the difference in CFUs in Fig. EV1A. The difference between the plasmid-bearing strains could be due to a negative effect of GIM3 overexpression (*pTDH3-GIM3*) versus expression from its own promoter (GIM3), but this difference should be lost after plasmid curing. And why is the difference between *pTDH3-GIM3* and GIM3 not seen after EMS treatment (Fig. 1C) if GIM3 overexpression is the cause of the reduced CFU of the former in Fig. EV1A?

8) Other descriptions in the figures are also confusing. SC-uracil in Fig. 2A indicates synthetic complete medium without uracil. Accordingly, SC-liquid would mean SC medium without liquid (!), and SC-FOA would mean SC medium without FOA, but the latter in fact contains FOA. The minus sign should not have different meanings.

9) Lines 1134-1135: What is the identity of the control strain? How can it grow on medium without uracil (selects for URA3) and also on medium with FOA (selects for *ura3*)?

10) The interpretation of the results shown in Fig. EV3E was not obvious to me. In both the left and right panels the growth inhibition zone appears to be the same for the strains with and without GIM3, and in both cases the strain with GIM3 exhibits more residual growth within the inhibition zone. The identity of the strains should be given for comparison with the data in Table EV2.

11) In lines 315-316 the authors state that they deleted GIM3 from the 100 evolved clones, but in the legend to Fig. 5 (line 1183) and in the methods (lines 822-823 and 825) they state that only 77 evolved clones were tested because 23 went extinct.

12) Lines 349-353: 25% of evolved clones (i.e. 19 of the 77 clones?) showed increased caspofungin resistance, of which 57.8% (i.e. 11 of those 19 clones?) exhibited decreased resistance upon GIM3 deletion. How many of these showed decreased fitness also in the absence of selection (30.2%, i.e. 23 of the 77 clones (?), see lines 329-331)? In such clones the apparent importance of *Gim3* for caspofungin resistance (tolerance?) might not be true. The same applies to the 43.4% of evolved clones that lost their fluconazole tolerance upon GIM3 deletion (lines 322-325). I did not understand how many of the 77 evolved clones can yield the stated percentages of 57.8%, 30.2%, and 43.4%.

Referee #2:

The authors addressed all my concerns and have significantly strengthened the ms. The ms is suitable to be published in EMBO reports in its current format in my opinion

Referee #3:

The authors have addressed the comments of the referees. They fail to mention the induction of petite mutants under azole treatment although they saw no small colonies. I think it should be cited to bring this to the attention of others for both species. The additional sensitivity testing is a positive addition. It is possible a mixture exist in their colonies under treatment. They also cite Martel et al 2010 and Jackson et al 2003 claiming drug efflux changes seen in conjunction with *erg3* mutations but this is incorrect.

Dear Dr. Moran,

We have now addressed all remaining reviewers' comments in our revised manuscript. As you can see, we -again- performed a considerable number of additional experiments to address Reviewer 1's follow-up questions. Specifically, to further investigate how Gim3 could potentiate the effect of the identified mutations in Erg3, we now assayed two Erg3 mutants with different predicted stabilities (Erg3^{G195D} and Erg3^{T289P}, with the T289P mutation predicted to have a more than 4 fold higher destabilizing effect on Erg3 structure than the G195D mutation) in addition to Erg3^{WT}, each with (*GIM3*) and without (*gim3Δ*) *GIM3*, and determined protein levels at 5 different timepoints. Additionally, we also assayed fluconazole susceptibility by BDA and Etest, and determined sterol profiles of these strains in the absence and presence of fluconazole. Together, these results provide further evidence for reduced Erg3 activity in the tested mutants, as well as a role for Gim3 in stabilizing the mutant Erg3 proteins.

Apart from these results, we have now also more clearly differentiated between tolerance and resistance in the different sections of our manuscript.

We have also adjusted the manuscript to address all editorial comments and additional notes as well. We opt for publishing our study as a full article, since we think this format suits our data best.

Below, we address all questions and concerns raised by the reviewers (*in blue, in italics, in-between the reviewers' comments*) and highlight how we adapted our manuscript accordingly.

Yours sincerely,
Kevin Verstrepen & Patrick Van Dijck, corresponding authors

COMMENTS to manuscript EMBOR-2024-60483V2 (Gim3 buffers and potentiates *de novo* mutations that affect fluconazole resistance in yeast)

Reviewer 1

In their revised manuscript the authors have partially addressed my previous criticisms, but some critiques remain.

1) In one of my previous comments I had suggested to experimentally test the stability of the mutated Erg3 proteins in the presence and absence of GIM3 and thereby verify the *in silico* predictions and the authors' conclusion that Gim3 potentiates fluconazole resistance by stabilizing the mutated proteins. The authors tested only one of the mutated proteins and the wild type Erg3 instead of a panel of proteins with different predicted stabilities and different dependencies on Gim3 for conferring fluconazole tolerance. The experiment shown in Fig. 4 also lacks an important control, the wild-type strain with wild-type Erg3, which would show the importance of Gim3 for the stability of wild-type Erg3. Furthermore, protein levels were determined at only two time points after translation inhibition, with highly variable values for biological replicates, instead of determining the half-life of the different proteins. The legend to the figure is highly confusing. It describes asterisks with different colors, but only one purple asterisk is shown in Fig. 4B. On top of it, the legend erroneously states (lines 1171-1173) that the

stability of (the mutated) Erg3 in the presence of GIM3 (orange line) was significantly decreased compared with that of wild-type Erg3 in the absence of GIM3 (purple line), although the difference was between the mutant Erg3 in the absence (grey line) and presence (purple line) of GIM3 (purple asterisk), as also stated in the text (lines 268-271).

We have now assayed two Erg3 mutants with different predicted stabilities (Erg3^{G195D} and Erg3^{T289P}, with the T289P mutation predicted to have a more than 4-fold higher destabilizing effect on Erg3 structure compared to the G195D mutation) in addition to Erg3^{WT}, each with (GIM3) and without (gim3Δ) GIM3, and determined protein levels at 5 different timepoints. Specifically, we sampled at 0, 15, 30, 60, and 120 min in a cycloheximide chase experiment and analyzed n = 3 independent biological replicates per strain at each time point. We estimated half-life (t_{1/2}) from log-linear fits of ln(%T0) vs time, as is typically done to determine protein half-life (Christiano et al, 2014; Belle et al, 2006).

We find that while Erg3^{WT} protein stability does not appear to depend on Gim3, the tested Erg3 mutants show reduced stability in the absence of Gim3. Interestingly, Erg3^{T289P}, the mutant with the lowest predicted stability, already has a lower protein half-life in the presence of Gim3 when compared to an Erg3^{WT} protein (p-value= 0.024, two-sided t-test), and this half-life is further reduced in the absence of Gim3 (p-value= 0.016, two-sided t-test). For Erg3^{G195D}, we find a drastically reduced protein half-life in the absence of Gim3 (p-value= 0.015, two-sided t-test). Together, these results indicate that Gim3 stabilizes these mutant Erg3 proteins, likely supporting a Gim3-dependent fluconazole susceptibility phenotype.

We agree that the way we indicated statistically significant differences in the previous version of the manuscript was confusing. We have now mentioned the results of the statistical test in the figure legend, but did not indicate it in the figure explicitly to avoid confusion.

The revised text now reads as follows (lines 265-276)

*“To investigate how Gim3 could potentiate the identified Erg3 non-synonymous mutations, we experimentally assessed the stability of Erg3 proteins via a cycloheximide-based protein degradation assay (Christiano et al, 2014; Belle et al, 2006) (**Fig. 4A**). Specifically, we tested stability of His-tagged Erg3^{WT} and two Erg mutants with different predicted stabilities (Erg^{G195D} and Erg3^{T289P}). The Erg3^{T289P} mutation is predicted to have a more than 4-fold stronger destabilizing effect on Erg3 compared to the Erg^{G195D} (see **Fig. 3C**). All variants were analyzed in the presence and absence of GIM3. We find that while Erg3^{WT} protein stability does not appear to depend on Gim3, the tested Erg3 mutants show reduced stability in the absence of Gim3. Interestingly, Erg3^{T289P}, the mutant with the lowest predicted stability, already has a lower protein half-life in the presence of Gim3 when compared to an Erg3^{WT} protein (p-value= 0.024, two-sided t-test), and this half-life is further reduced in the absence of Gim3 (p-value= 0.016, two-sided t-test). For Erg3^{G195D}, we find a drastically reduced protein half-life in the absence of Gim3 (p-value= 0.015, two-sided t-test) (**Fig. 4A**).”*

*The data is represented as panel A of revised **Figure 4**.*

Figure 4. Absence of Gim3 reduces the stability of Erg3 mutants and affects fluconazole susceptibility and sterol profiles. (A) Plot showing the percentage decrease in Erg3-His₆ protein levels in the presence and absence of Gim3 over time. Data represent mean \pm s.e.m. from three independent biological replicates per strain and time point. Data is shown relative to the initial Erg3 protein levels at time 0 (T0) for each strain background. The table shows the per-strain protein half-life ($t_{1/2}$, minutes) estimated from log-linear fits of %T0 vs time for each replicate, s.e.m-log indicates the standard error computed on log-transformed half-lives, and percent stability retained represents the relative half-life of each Erg3 variant in *gim3Δ* compared to *GIM3*. Two-sided t-test results comparing protein half-life values of each variant to the corresponding Erg3^{WT}+*GIM3* are provided in **Table EV3**.

2) As noted previously, given the known role of ERG3 mutations in fluconazole resistance and the established Gim3-Erg3 interaction, it would have been more exciting to know which other mutations caused Gim3-dependent fluconazole resistance in the strains with wild-type ERG3. In their rebuttal letter the authors argue that identifying responsible mutations would require an amount of work that goes beyond what is reasonable for a revision, which is understandable. However, in that case I would expect a deeper investigation of how Gim3 potentiates the effect of Erg3 mutations on fluconazole tolerance. It is thought that loss or reduced activity of Erg3 prevents the accumulation of toxic sterols that inhibit growth in the presence of the drug. If Gim3 stabilizes the mutated Erg3 proteins, as the authors conclude from the results with one mutated protein, one should a priori expect that this increases the levels of toxic sterols and actually reduces drug tolerance. Stabilization of the mutated Erg3 proteins might therefore be required to enable sufficient ergosterol synthesis to occur without the production of toxic sterols in the presence of fluconazole. To understand the effect of the various Erg3 mutations on fluconazole tolerance and its dependence on Gim3, the different mutations should be introduced into isogenic wild-type and *gim3Δ* backgrounds, since most of the evolved strains contain additional mutations that affect fluconazole susceptibility, as is evident from the new Table EV2. The stability of the proteins and the fluconazole tolerance of the strains should then be compared. Furthermore, sterol profiles of the strains in the presence and absence of fluconazole should be determined to see how the mutations and the presence of Gim3 affect the accumulation of toxic sterols.

*We have now determined fluconazole susceptibility as well as sterol profiles of WT and *gim3Δ* strains expressing Erg3^{WT}, Erg3^{G195D} and Erg3^{T289P}.*

*Results for fluconazole susceptibility are shown in revised **Figure 4B**. We find that an Erg3^{T289P} mutant exhibits a high MIC₅₀ (>128 μ g/mL; comparable to the MIC₅₀ observed for *erg3Δ*) in the presence and absence of *GIM3*. The Erg3^{G195D} mutant showed an increased MIC₅₀ of 32 μ g/mL in the presence of *GIM3* compared to 16 μ g/mL in its absence (see **Figure 4B**). Notably, both Erg3*

mutants still showed growth at fluconazole concentrations that completely inhibited the growth of the *Erg3* WT strain. Moreover, strains containing *Erg3*^{T289P} or *Erg3*^{G195D} also show enhanced fluconazole tolerance (**Figure 4C**, Etest results after 48h showing trailing growth), with reduced fluconazole tolerance of *Erg3*^{G195D} in the absence of *Gim3*. Together, these results show that the tested *Erg3* mutations affect fluconazole tolerance, and at least part of these phenotypes depend on *Gim3*.

The text now reads as follows (lines 277-285)

“Next, we determined fluconazole susceptibility of these *Erg3* mutant strains. We found that an *Erg3*^{T289P} mutant exhibits a high MIC₅₀ (>128 µg/mL; comparable to the MIC₅₀ observed for *erg3Δ*) in the presence and absence of *GIM3*. The *Erg3*^{G195D} mutant showed an increased MIC₅₀ of 32 µg/mL in the presence of *GIM3* compared to 16 µg/mL in its absence (**Fig. 4B**). Notably, both *Erg3* mutants still showed growth at fluconazole concentrations that completely inhibited the growth of the *Erg3*^{WT} strain. Moreover, strains containing *Erg3*^{T289P} or *Erg3*^{G195D} also show enhanced fluconazole tolerance (**Fig. 4C**, Etest results after 48h showing trailing growth), with reduced fluconazole tolerance of *Erg3*^{G195D} in the absence of *Gim3*. Together, these results show that the tested *Erg3* mutations affect fluconazole tolerance, and at least part of these phenotypes depend on *Gim3*.”

(B) Absence of *Gim3* affects fluconazole susceptibility. Right: Fluconazole broth dilution assay of *Erg3* variants in *GIM3*⁺ and *gim3Δ* backgrounds (n= 2 biological replicates for each variant). Growth values were normalized to drug-free medium (Relative growth %) after 24h exposure to increasing fluconazole concentrations, as shown for strains expressing *Erg3*^{WT}, *Erg3*^{G195D}, or *Erg3*^{T289P} either in a *GIM3* background (solid lines) or in a *gim3Δ* background (dashed lines). The horizontal red line marks 50% growth relative to drug-free controls and corresponds to the MIC₅₀ threshold. Left: Table summarizing

the MIC₅₀ values obtained from the broth-dilution assay for each strain. **(C) Fluconazole Etest profiles for Erg3 variants (n= 2 biological replicates) in GIM3 and gim3Δ backgrounds.** Representative Etest for fluconazole is shown after 24h (left panel) and 48h (right panel) incubation. MIC values (μg/mL) are indicated below each strip.

The text now reads as follows (lines 286-301)

“To investigate how these Erg3 mutations and the presence of Gim3 affect the accumulation of toxic sterols, we next determined sterol profiles in the presence and absence of fluconazole (Fig. 4D & 4E). Under drug-free conditions, and in line with what was previously reported, the erg3Δ strain produces no ergosterol and accumulates ergosta-7,22-dienol, consistent with loss of Erg3 activity (Heese-Peck et al, 2002; Guan et al, 2009). Based on the sterol profile, Erg^{T289P} has severely reduced Erg3 activity, since Erg^{T289P} strains produce no ergosterol and accumulate ergosta-7,22-dienol, mimicking the sterol profiles of an erg3Δ. Strains expressing Erg^{G195D} appear to retain partial Erg3 activity, producing some ergosterol while also accumulating ergosta-7,22-dienol (Fig. 4E).

Under fluconazole treatment, toxic 14α-methylergosta-8,24(28)-diol is formed in an Erg3^{WT} strain. This toxic sterol does not accumulate in an erg3Δ strain, and levels of this sterol are also significantly lower in the Erg3 mutant strains than in the Erg3^{WT}+GIM3. (p-value for Erg^{G195D}+GIM3 = 0.038, p-value for Erg^{T289P}+GIM3 = 0,0095, one-sample t-test). Interestingly, Gim3 appears to be required for the residual activity of Erg3^{G195D}, since almost no toxic diol is detected in the absence of Gim3.

Together, these sterol profiles are in line with what we observe in our protein stability assay (Fig. 4A), as well as our BDA and Etests (Fig. 4B & 4C), and indicate reduced Erg3 activity in the tested mutants, as well as a role for Gim3 in stabilizing mutant Erg3 proteins.”

D

E

(D) Simplified schematic of the late ergosterol biosynthesis pathway in *Saccharomyces cerevisiae*. Fluconazole inhibits Erg11, reducing ergosterol production and causing the accumulation of 14 α -methyl sterol intermediates (toxic diols). When Erg3 activity is lost or reduced, these intermediates cannot be converted into the toxic diol and alternative, less toxic products such as ergosta-7,22-dienol are formed. **(E) Sterol profiles of Erg3^{WT}, Erg3 missense mutants (Erg3^{G195D}, Erg3^{T289P}), and an *erg3 Δ* strain in *GIM3* and *gim3 Δ* backgrounds, grown with or without fluconazole.** Bars represent the relative sterol content normalized to both the internal standard and Erg3^{WT} under each condition. For the toxic intermediate 14 α -methylergosta-8,24(28)-diol, loss of *GIM3* significantly altered sterol levels in both Erg3^{G195D} and Erg3^{T289P} backgrounds (p-value = 0.013 and 0.006, respectively). Additionally, levels of 14 α -methylergosta-8,24(28)-diol were significantly reduced in the Erg3^{G195D}+*GIM3* (p-value = 0.038) and Erg3^{T289P}+*GIM3* (p-value = 0.0095) relative to Erg3^{WT}+*GIM3* under fluconazole treatment. For ergosterol, *GIM3* deletion significantly affected ergosterol levels in the Erg3^{WT} strain (p-value = 0.038) and in the Erg3^{T289P} variant (p-value = 0.0259) under drug-free conditions, and also in the Erg3^{WT} background in the presence of fluconazole (p-value = 0.018). Only statistically significant p-values are shown, corresponding to comparisons between the same variant in the presence and absence of *GIM3*, as well as comparisons of all variants to the Erg3^{WT} reference. Significant values are indicated as n.s. = not significant, ND, not statistically testable (no variance across replicates), *p < 0.05, ***p < 0.001, ****p < 0.0001. All additional comparisons are reported in **Table EV4**. Exact strain genotypes are listed in **Table EV7**.

3) Considering that ERG3 loss-of-function mutations are reported to result in fluconazole resistance in *S. cerevisiae*, it is surprising that the frameshift and nonsense mutations (note that nonsense mutations are not usually termed frameshift mutations, line 258-259), which would be expected to prevent production of a functional Erg3 protein, did not result in increased MICs compared to wild-type MIC (12 μ g/ml, see Fig. EV3) and most of them even were associated with increased sensitivity (Table EV2). The new MIC data shown in Table EV2 and the phenotypes shown elsewhere demonstrate that Gim3 does not affect fluconazole resistance and instead enables residual growth in the presence of the drug, which is often referred to as trailing growth or tolerance. Since the authors emphasize the relevance of their results for clinical drug resistance, the term resistance should be used carefully. Was the trailing growth of the evolved strains and its loss after removal of *GIM3* also observed in broth microdilution assays when growth was assayed after prolonged incubation (48 h instead of 24 h, which is often used to distinguish tolerance and resistance)?

We have clarified the terminology (resistance and tolerance) throughout the text and now refer to the phenotypes of ERG3 mutants as “tolerance” rather than “resistance” where appropriate. We also corrected the wording in lines 258–259 to distinguish frameshift from nonsense mutations.

Lines 256-260:

“Almost all identified ERG3 mutations were non-synonymous mutations (missense and nonsense) (~87%, 20 out of 23), with the remaining mutations being frameshift mutations (Fig. 3B).”

Note that we think the reviewer is here asking about the tolerance and trailing growth of the EMS-treated strains instead of the evolved strains, since this is the data shown in Table EV2. Table EV2 indeed shows that only a few of the tested EMS-treated strains display increased MIC based on the Etest (resistance), while most actually appear to exhibit trailing growth or tolerance (based on Etest). We believe that the high-throughput assay used in Figure 2, which used colony area

measurements to identify strains affected in growth on fluconazole, primarily detected tolerant rather than fully resistant variants, and that this is also reflected in **Table EV2**. We have now clarified this in the revised manuscript (lines 236-246).

“To investigate whether the reduced growth observed on fluconazole-containing agar plates in strains lacking GIM3 was associated with changes in MIC, we performed Etest assays on 35 EMS-derived mutants that showed either a synthetic lethal phenotype or a $\geq 50\%$ reduction in colony area at 128 $\mu\text{g}/\text{mL}$ fluconazole following GIM3 removal. **3 strains showed a higher MIC in the presence of GIM3 (EMS18, EMS19 and EMS23)**, while most strains had comparable MIC values with or without GIM3, suggesting that for some, but not all backgrounds, GIM3 influences the MIC (**Table EV2**). Interestingly, however, nearly all strains displayed pronounced trailing growth in the presence of GIM3, which was reduced following GIM3 removal (**Fig. EV3E**). This pattern suggests that in most backgrounds, GIM3 contributes to fluconazole tolerance. These results also indicate that the high-throughput assay used to determine fluconazole growth primarily identified tolerant rather than fully resistant variants.”

We also agree that at first sight it is surprising that cells containing an Erg3 protein with frameshift or nonsense mutations would have a MIC lower than WT (since these cells are expected to behave like an *erg3 Δ* strain). Since EMS exposure introduces an average of ~ 86 to 124 mutations per genome per strain (**Figure EV2**), we hypothesize that these mutations, apart from the mutations in *ERG3*, also influence fluconazole resistance in some of the strains.

Some editorial comments and suggestions:

- Clone 17 seems to be misnumbered as clone 14 in Fig. 3A.

*Thank you for noticing this. **Figure 3A** has been corrected.*

- Tables EV2 and EV3 could be combined and the wild-type strain should be included for comparison of the MICs. The fluconazole MIC for the wild-type (12 $\mu\text{g}/\text{ml}$) can only be found in Fig. EV3. Including it in Table EV2 will highlight that most evolved fluconazole-"resistant" clones actually exhibit increased sensitivity and display only increased tolerance.

*We have now included MIC for the WT strain in **Table EV2**. We have also indicated in Table EV2 which Erg3 mutations the strains contain, and combined **Table EV2** and **EV3**.*

- Which are the 5 strains with a higher MIC in the presence of GIM3 versus its absence mentioned in lines 240-241? Table EV2 contains 4 strains with at least 4-fold higher MICs (EMS19, 21, 26, and 32) and two additional strains with a marginally higher MIC in the presence of GIM3 (3 $\mu\text{g}/\text{ml}$ versus 2 $\mu\text{g}/\text{ml}$, EMS4 and 28).

We have now clearly indicated this in the revised manuscript. Specifically, lines 239-241.

“**3 strains showed a higher MIC in the presence of GIM3 (EMS18, EMS19 and EMS23)**, while most strains had comparable MIC values with or without GIM3, suggesting that for some, but not all backgrounds, GIM3 influences the MIC (**Table EV2**).”

*In the revised manuscript, we clarify that **three, rather than five, strains show a higher MIC in the presence of GIM3**. In the original version, the numbering of strains in **Figure 3A** did not*

correspond to Table EV2, which may have contributed to confusion. This has now been corrected by including the ERG3 mutation information directly in **Table EV2**.

We have also re-examined the MIC values according to the reviewer's feedback, and now define MIC strictly based on the **visible breakpoint** in the Etest (rather than tolerance beyond this point), which aligns the interpretation with standard clinical MIC definitions.

Based on this revised analysis, three strains exhibit a higher MIC in the presence of GIM3: EMS18 (*Erg3^{L167R}*), EMS19 (*Erg3^{G310E}*; the second most destabilizing mutation based on *in silico* predictions, **Fig. 3C**), and EMS23 (*Erg3^{G195D}*) (**Table EV2**).

4) In line 370 the authors state that "deletion of ERG3 in *N. glabratus* resulted in a higher MIC₈₀ value 256 µg/mL compared to WT", but the corresponding Fig. 6A uses MIC₅₀ as threshold and shows that this is the same for both strains (256 µg/mL).

While different tests can give somewhat different MIC values, the drastic differences between broth microdilution (256 µg/mL for all strains) and Etest (8 µg/mL for the wild type and 2 µg/mL for *erg3Δ* mutants and *erg3Δ gim3Δ* double mutants) in this study are astonishing.

*We agree and are also puzzled by these results. We think that the differences in MIC values between Etest and BDA likely originated from the use of different media in the initial experiments. Specifically, the fluconazole broth microdilution assay in the original version of the manuscript was performed in 2% YPD, whereas all other antifungal susceptibility assays (including the Etest) were carried out on RPMI 1640 medium buffered with MOPS. Now, we repeated the fluconazole broth microdilution assay using the same RPMI+MOPS medium used for the Etest. As shown in the revised Figure 6A, the MIC₅₀ values are now 4 µg/mL for the WT and the *erg3Δ* strain.*

Regardless, the results shown in Fig. 6A and B indicate that ERG3 deletion actually reduces fluconazole resistance in the *N. glabratus* strain used in this study (at least in the Etest) but increases tolerance (trailing growth at fluconazole concentrations above the MIC). Furthermore, MIC levels of *erg3Δ* mutants remain unchanged in the absence of Gim3, but Gim3 is required for the increased fluconazole tolerance of *erg3Δ* mutants. The authors should modify their conclusions accordingly and correctly describe the results. Examples:

- Line 375 "...*erg3Δ* did not show an increased MIC (Etest MIC = 2)...". In fact, the MIC was reduced by 4-fold compared to the wild type (2 µg/mL versus 8 µg/mL).
- Line 381 "Although *erg3Δ* strains did not show higher MIC₅₀ values, they did display enhanced survival at high amphotericin B concentrations". Fig. 6C shows growth, not survival.
- Line 387: "In contrast to the fluconazole and amphotericin B results, GIM3 deletion in the *erg3Δ* background did not lead to a measurable change in MIC by E-test". Table EV5 shows that fluconazole MICs also remained unchanged.

*As mentioned in our reply above, we have now repeated the BDA assay for fluconazole in the same medium as that used for the Etest. MIC₅₀ values were also recalculated per replicate by first normalizing growth to the 0 µg/mL condition and then determining the concentration at which relative growth decreased to 50% (MIC₅₀). We find that *erg3Δ* shows a similar MIC₅₀ value compared to the WT for fluconazole (4 µg/mL, **Figure**).*

Overall, the results show that fluconazole tolerance, rather than resistance, is strongly influenced by GIM3, and this is clearly detected by the E-test (via trailing growth). Amphotericin B susceptibility is increased upon GIM3 deletion in both assay types (Etest and broth dilution

assay). For caspofungin, *ERG3* deletion consistently leads to elevated MICs, while *GIM3* plays no major role. We have adapted our manuscript to make this clear.

Specifically, the revised manuscript now reads (lines 379-398)

“Across these three antifungal drugs, the Etest and broth microdilution assay (BDA) revealed assay-specific susceptibility patterns for the WT, *gim3Δ*, *erg3Δ*, and *erg3Δ gim3Δ* strains. In fluconazole, the BDA test showed a similar MIC_{50} for *erg3Δ* compared to WT ($MIC_{50} = 4$, for both strains, **Fig. 6A**). The Etest on the other hand, revealed a clear difference in tolerance, with *erg3Δ* showing a fourfold lower MIC compared to the wild type (2 $\mu\text{g}/\text{mL}$ versus 8 $\mu\text{g}/\text{mL}$) but developing pronounced trailing growth, a hallmark of azole tolerance (**Fig. 6B**). This trailing completely disappeared in the *erg3Δ gim3Δ* double mutant, indicating that *GIM3* is required for the *erg3Δ*-associated tolerance phenotype.

In amphotericin B, both assays showed that *GIM3* deletion increases drug susceptibility. Etests showed a fourfold MIC reduction for *gim3Δ* (0.125 to 0.032 $\mu\text{g}/\text{mL}$, **Table EV6**), and the BDA confirmed a decrease in MIC_{50} values for this strain (*gim3Δ* = 0.25 $\mu\text{g}/\text{mL}$ versus WT = 0.5 $\mu\text{g}/\text{mL}$, **Fig. 6C**). The *erg3Δ* strain did not exhibit higher MIC_{50} than the WT, but it did show enhanced growth at higher amphotericin B concentration compared to the WT, *gim3Δ*, and *erg3Δ gim3Δ* strains (**Fig. 6C**, p -values = 0.0021, 8.51×10^{-5} , and 0.00083, respectively, two-way ANOVA test, **Table EV5**).

Finally, for caspofungin, both methods consistently showed increased MICs for *erg3Δ* compared to WT and *gim3Δ* (0.25 $\mu\text{g}/\text{mL}$ versus 0.125 $\mu\text{g}/\text{mL}$, 0.125 $\mu\text{g}/\text{mL}$, respectively, in both assays), whereas *GIM3* deletion had no detectable effect in either the single or double mutant (**Figure 6D**, **Table EV6**).

Together, these findings indicate that fluconazole tolerance, rather than classical resistance, is strongly influenced by *GIM3*, and this is readily detected by the Etest (via trailing growth) than by broth microdilution. Amphotericin B susceptibility is increased upon *GIM3* deletion in both assay types. For caspofungin, *ERG3* deletion consistently leads to elevated MICs, while *GIM3* plays no major role.”

Figure 6: Antifungal susceptibility profiles of *N. glabratus* WT, *gim3Δ*, *erg3Δ*, and *erg3Δ gim3Δ* strains. (A) Broth microdilution (BDA) fluconazole susceptibility curves. The BDA did not show significant growth differences between strains across fluconazole concentrations (Table EV5, two-way ANOVA), and the MIC₅₀ for both *erg3Δ* and WT was 4 µg/mL. Deletion of *GIM3*, either alone or in the *erg3Δ* background, did not affect fluconazole MIC₅₀ value. The red dashed line indicates the 50% growth threshold used to determine MIC₅₀. (B) Fluconazole Etest (48h). The *erg3Δ* mutant showed a fourfold lower MIC (2 µg/mL) compared to WT (~8 µg/mL), but produced pronounced trailing growth, indicative of azole tolerance. This trailing phenotype was absent in the *erg3Δ gim3Δ* double mutant, despite an identical MIC (2 µg/mL). (C) Amphotericin B BDA assay. The *gim3Δ* mutant displayed increased susceptibility to amphotericin B, showing a lower MIC₅₀ than the WT (WT MIC₅₀ = 0.125 µg/mL, *gim3Δ* MIC₅₀ = 0.032 µg/mL). The *erg3Δ* strain did not exhibit a higher MIC₅₀ than the WT, but it did show enhanced growth at higher amphotericin B concentrations compared with the WT, *gim3Δ*, and *erg3Δ gim3Δ* strains ($p = 0.0021$, 8.51×10^{-5} , and 8.38×10^{-4} , respectively, two-way ANOVA). (D) Caspofungin BDA assay. The *erg3Δ* strain exhibited higher MIC₅₀ values compared to WT and *gim3Δ* (0.25 µg/mL versus 0.125 µg/mL, 0.125 µg/mL, respectively). Deletion of *GIM3* had no significant effect on caspofungin susceptibility in either the single or double mutant, consistent with the Etest results (Table EV6). The figure further indicates significantly higher growth of *erg3Δ* strains at higher caspofungin concentrations, detailed statistical comparisons and p -values are provided in Table EV5. Table EV6 provides the complete Etest results for all three antifungal drugs, while Table EV5 contains the pairwise p -

values from two-way ANOVA with Tukey's post hoc test comparing relative growth (%) across antifungal treatments among the WT, *gim3Δ*, *erg3Δ*, and *erg3Δ gim3Δ* strains. Statistical analyses shown in the figure were performed using three biological replicates. Data are shown as mean ± s.e.m. ns = not significant; *p < 0.05; **p < 0.01; ***p < 0.001; ****p < 0.0001. Exact genotypes of the strains used in this figure are provided in **Table EV7**.

In my comments to the original manuscript I had addressed only some main points, but the following issues should also be clarified:

5) One thing that I failed to notice in the original manuscript: Why did the authors not introduce an empty vector into the *gim3Δ* mutants to complement their uracil auxotrophy, as in the strains in which GIM3 was reintroduced on a plasmid (see lines 182-186)? This would have been the appropriate control. Could uracil auxotrophy explain the now reported smaller colony size of EMS-treated *gim3Δ* mutants compared to EMS-treated cells containing GIM3 (lines 216-218)? The authors do not explicitly state that such a difference was not observed in the absence of EMS treatment.

*Here it is important to realize that the approach suggested by the reviewer (introducing an empty vector into the *gim3Δ* mutants) prior to the EMS mutagenesis would not have allowed us to identify Gim3-dependent mutants (since these would not be viable in a strain lacking Gim3).*

*Importantly, we do not think the uracil auxotrophy explains the smaller colony size observed for EMS-treated *gim3* mutants compared to EMS-treated cells containing GIM3, since cells were plated on SC medium containing uracil. Moreover, we find that colonies of Gim3-containing cells (*gim3Δ* + GIM3 on plasmid) are actually slightly smaller compared to *gim3Δ* deletion colonies (Wilcoxon, BH-adjusted $p = 4.5 \times 10^{-8}$) on SC medium prior to EMS treatment. These results indicate that the significantly smaller colony size of EMS-treated cells lacking GIM3 is not due to uracil auxotrophy, but instead to fitness effects of the EMS-induced mutations.*

6) The *gim3Δ* mutant was constructed only once from the wild-type strain BY4741 (lines 551-554). What is the difference between the two *gim3Δ* mutants in Fig. EV1 and why did one of them produce more colonies than the other in Fig. EV1A? The same applies to Fig. 1B-D and Fig. EV3. As I understood, these are not the *gim3Δ* mutants that were later derived from the plasmid-bearing strains by FOA treatment (lines 221-225). A table listing all strains with specific names, genotypes, and parents should be added, and the legends to the figures should explicitly state the names of the strains that were used in each particular experiment.

*The two *gim3Δ* strains are **not identical**.*

- *Strain A was created from a *gim3Δ* background that initially carried a plasmid expressing GIM3 from its native promoter.*
- *Strain B was created from the same *gim3Δ* background but initially carried a plasmid expressing GIM3 from the TDH3 promoter.*

*FOA treatment resulted in loss of the GIM3 plasmid, so both final strains are genetically a *gim3Δ*, but they differ in the plasmid that previously complemented them.*

*To make this clear we have re-labeled the figures **Fig. EV1**, **Fig. EV1A**, **Fig. 1B-D**, and **Fig. EV3**:*

- ***“*gim3Δ* (no pGIM3-GIM3)”** for the strain cured of the native promoter plasmid*
- ***“*gim3Δ* (no pTDH3-GIM3)”** for the strain cured of the TDH3 promoter plasmid.*

We also added a Table listing additional information, including a column specifying in which figure data for these strains are shown.

We also added a strain table with full names, genotypes, and parent strains (**Table EV7**):

Code	Strain ID	Genotype	Plasmid history	Parent strain ID	Used in Figures
S. cerevisiae					
MT1	BY4741	MATa his3Δ1 leu2Δ0 met15Δ0 ura3Δ0			Fig 5C-5F (as control), Fig EV2A, Fig EV2D, Fig EV4
MT2	gim3Δ (genomic deletion)	MATa his3Δ1 leu2Δ0 met15Δ0 ura3Δ0 gim3Δ		MT1	
MT3	WT+PV1382	MATa his3Δ1 leu2Δ0 met15Δ0 ura3Δ0	Carries plasmid pV1382	MT1	Fig 1B-1D, Fig2 B, Fig EV1A, Fig EV3A-EV3D
MT4	WT (no PV1382)	MATa his3Δ1 leu2Δ0 met15Δ0 ura3Δ0	Lost plasmid pV1382; plasmid lost by FOA	MT3	Fig 1B-1D, Fig2 B, Fig EV1A, Fig EV3A-EV3D
MT5	GIM3	MATa his3Δ1 leu2Δ0 met15Δ0 ura3Δ0 gim3Δ	Plasmid carry URA3 and GIM3 under native promoter	MT2	Fig 1B-1D, Fig 2B, Fig EV1A, Fig EV1B, Fig EV3A-EV3D
MT6	pTDH3-GIM3	MATa his3Δ1 leu2Δ0 met15Δ0 ura3Δ0 gim3Δ	Plasmid carry URA3 and GIM3 under TDH3 promoter	MT2	Fig 1B-1D, Fig 2B, Fig EV1A, Fig EV1B, Fig EV3A-EV3D
MT7	gim3Δ (no pGIM3-GIM3)	MATa his3Δ1 leu2Δ0 met15Δ0 ura3Δ0 gim3Δ	Previously carried pGIM3-native ; plasmid lost by FOA	MT5	Fig 1B-1D, Fig 2B, Fig EV1A, Fig EV1B, Fig EV3A-EV3D
MT8	gim3Δ (no pTDH3-GIM3)	MATa his3Δ1 leu2Δ0 met15Δ0 ura3Δ0 gim3Δ	Previously carried pGIM3-native ; plasmid lost by FOA	MT6	Fig 1B-1D, Fig 2B, Fig EV1A, Fig EV1B, Fig EV3A-EV3D
MT9-MT50	EMS-treated strains used for Etest (see Table EV2 for more details)	MATa his3Δ1 leu2Δ0 met15Δ0 ura3Δ0	Plasmid carry URA3 and GIM3 under native promoter, or plasmid carry URA3 and GIM3 under TDH3 promoter	MT5 or MT6	MT9-M31 in Fig3A, MT9-M31 in Fig3B MT (9, 10, 12, 13, 14, 15, 17, 18, 19, 20, 23, 24, 25, 26, 27, 31) in Fig 3C, MT12 in EV3E
MT50	Erg3^{WT}+ GIM3	MATa his3Δ1 leu2Δ0 met15Δ0 ura3Δ0 gim3Δ ERG3-6xHis		MT1	Fig 4A-4C, Fig 4E
MT51	Erg3^{WT}+ gim3Δ	MATa his3Δ1 leu2Δ0 met15Δ0 ura3Δ0 gim3Δ ERG3-6xHis		MT2	Fig 4A-4C, Fig 4E
MT52	Erg3^{G195D}+ GIM3	MATa his3Δ1 leu2Δ0 met15Δ0 ura3Δ0 ERG3-6xHis		MT50	Fig 4A-4C, Fig 4E
MT53	Erg3^{G195D}+ gim3Δ	MATa his3Δ1 leu2Δ0 met15Δ0 ura3Δ0 gim3Δ ERG3-6xHis		MT51	Fig 4A-4C, Fig 4E
MT54	Erg3^{T289P}+ GIM3	MATa his3Δ1 leu2Δ0 met15Δ0 ura3Δ0 ERG3-6xHis		MT50	Fig 4A-4C, Fig 4E

MT55	Erg3 ^{T289P} + gim3Δ	MATa his3Δ1 leu2Δ0 met15Δ0 ura3Δ0 gim3Δ ERG3-6xHis		MT51	Fig 4A-4C, Fig 4E
MT56	erg3Δ + GIM3	MATa his3Δ1 leu2Δ0 met15Δ0 ura3Δ0 erg3Δ		MT1	Fig 4B, Fig 4C, Fig 4E
MT57	erg3Δ + gim3Δ	MATa his3Δ1 leu2Δ0 met15Δ0 ura3Δ0 gim3Δ erg3Δ		MT2	Fig 4B, Fig 4C, Fig 4E
MT58- MT133	Evolved BY4741	BY4741, MATa his3Δ1 leu2Δ0 met15Δ0 ura3Δ0		MT1	Fig 5B-5F
N. glabratus					
MT134	WT (ATCC2001)			(CBS 138)	Fig 6A-6D
MT135	gim3Δ			MT134	Fig 6A-6D
MT136	erg3Δ			MT134	Fig 6A-6D
MT137	gim3Δerg3Δ			MT135	Fig 6A-6D

7) If I misunderstood the text and the two *gim3Δ* mutants in Fig. EV1 are the plasmid-cured derivatives of the strains containing *GIM3* on a plasmid (expressed from its own or the *TDH3* promoter), I still not understand the difference in CFUs in Fig. EV1A. The difference between the plasmid-bearing strains could be due to a negative effect of *GIM3* overexpression (*pTDH3-GIM3*) versus expression from its own promoter (*GIM3*), but this difference should be lost after plasmid curing. And why is the difference between *pTDH3-GIM3* and *GIM3* not seen after EMS treatment (Fig. 1C) if *GIM3* overexpression is the cause of the reduced CFU of the former in Fig. EV1A?

*The reviewer is correct: the two *gim3Δ* strains in Fig. EV1 are plasmid-cured derivatives of strains previously expressing *GIM3* from either the native or the *TDH3* promoter. As the reviewer points out, *GIM3* overexpression strains indeed showed slower growth. We hypothesize that the lower CFU observed for the former *pTDH3-GIM3* derivative in Fig. EV1A (strain MT8) likely reflects a carry-over physiological cost of prior overexpression (slower growth before or during plasmid curing), which persisted into the CFU assay.*

EMS treatment has strong cytotoxic effects (compare CFU/mL of Fig. EV1A to CFU/mL of Fig. 1C). We hypothesize that this strong cytotoxic effect could mask baseline fitness variation.

8) Other descriptions in the figures are also confusing. SC-uracil in Fig. 2A indicates synthetic complete medium without uracil. Accordingly, SC-liquid would mean SC medium without liquid (!), and SC-FOA would mean SC medium without FOA, but the latter in fact contains FOA. The minus sign should not have different meanings.

We thank the reviewer for pointing out the inconsistent use of the minus sign. We have standardized all medium abbreviations so that “-” consistently denotes omission of a component (e.g., SC-uracil for SC lacking uracil).

*Media supplemented with a compound are now written as SC + FOA or SC (liquid) where appropriate. However, for genotypes and plasmids, the typical nomenclature denotes a gene driven by a specific promoter as *pXXX-GENEX*. Hence, we have not adapted this nomenclature, since we do not think this will cause confusion.*

All figures and legends have been revised accordingly.

9) Lines 1134-1135: What is the identity of the control strain? How can it grow on medium without uracil (selects for URA3) and also on medium with FOA (selects for *ura3*)?

The control strain here refers to a WT evolved strain that was added to the pinning assay plates after selection for (on medium lacking uracil) or against (on medium with FOA) the plasmid. We have now clarified this in the materials and methods section.

Specifically, the different steps of the protocol are:

- 1. Resistant/ tolerant mutants were pinned onto SC-uracil+ fluconazole 128 µg/mL (Pfizer, #F8929) rectangular agar plates in a 384-pin format and incubated for 72 hours at 30 °C.*
- 2. Colonies were then inoculated into a 384-well plate, with each well containing 80 µL of SC (liquid) media, using the Singer rotor robot.*
- 3. Subsequently, the cells were pinned onto SC-uracil+fluconazole plates and SC+FOA+fluconazole plates to select for and against the plasmid.*
- 4. From the SC-uracil agar plates, strains were pinned into SC-uracil liquid in a 384 well plate to prepare glycerol stocks. A control WT strain, grown separately in SC, was added to specific wells at this step. From the FOA agar plates, strains were pinned into SC liquid (since the plasmid was already lost), and the same control strain was added to the corresponding wells here as well. Because the control was only added during the liquid culture step and not plated on either SC-uracil or FOA, it never underwent FOA selection.*
- 5. Finally, the cells were pinned in a 1536 format (four replicates per resistant strain) onto SC+ fluconazole agar plates and incubated at 30 °C for 48 hours.*

The revised manuscript now reads (lines 696-710):

“These mutants were then pinned onto SC-uracil+fluconazole 128 µg/mL (Pfizer, #F8929) rectangular agar plates in a 384-pin format and incubated for 72 hours at 30 °C. Next, colonies were inoculated into a 384-well plate, with each well containing 80 µL of SC (liquid) media, using the Singer rotor robot. The plates were incubated overnight to allow for a mix of cells with and without the plasmid in each well. Subsequently, the cells were pinned onto SC-uracil+fluconazole plates and SC+FOA+fluconazole plates to select for and against the plasmid. From the SC-uracil agar plates, strains were pinned into SC-uracil liquid in a 384-well plate to prepare glycerol stocks. A control WT strain, grown separately in SC, was added to specific wells at this step. From the FOA agar plates, strains were pinned into SC liquid (since the plasmid was already lost), and the same control strain was added to the corresponding wells here as well. Because the control was only added during the liquid culture step and not plated on either SC-uracil or FOA, it never underwent FOA selection. Finally, the cells were pinned in a 1536 format (four replicates per resistant strain) onto SC+fluconazole agar plates and incubated at 30 °C for 48 hours. This allowed us to measure the colony area of mutant cells with and without the plasmid, thereby comparing the effect of removing GIM3 on mutant growth.”

10) The interpretation of the results shown in Fig. EV3E was not obvious to me. In both the left and right panels the growth inhibition zone appears to be the same for the strains with and without GIM3, and in both cases the strain with GIM3 exhibits more residual growth within the inhibition zone. The identity of the strains should be given for comparison with the data in Table EV2.

The diameter of the inhibition zone is indeed similar, but the number of colonies growing inside the zone is higher when GIM3 is present, indicating increased tolerance in the presence of GIM3.

This interpretation is now explicitly stated in the **Fig. EV3E** legend:

(E) Etest MIC analysis of 35 EMS-derived strains identified as *GIM3*-dependent in **Fig. 2B**. Only 3 strains showed a higher MIC in the presence of *GIM3*, indicating *GIM3*-dependent resistance, while most strains had comparable MIC values with or without *GIM3*. Representative image is shown: the left panel shows a *gim3Δ* strain; the right panel depicts a strain carrying *GIM3*. Zone diameters (MIC) are similar, but residual growth (“trailing”) within the zone is greater when *GIM3* is present. Complete MIC data are available in **Table EV2**. Exact genotypes of the strains used in this figure are provided in **Table EV7**.

We have also updated the legend to include the exact strain names matching **Table EV2**.

11) In lines 315-316 the authors state that they deleted GIM3 from the 100 evolved clones, but in the legend to Fig. 5 (line 1183) and in the methods (lines 822-823 and 825) they state that only 77 evolved clones were tested because 23 went extinct.

To avoid confusion, we have revised lines 315–316 simply to state that GIM3 was deleted in 76 evolved clones, without mentioning the extinct isolates, ensuring consistency with Fig. 5 and the Methods.

The revised manuscript now reads (lines 316-319):

*“We set up a laboratory evolution experiment in which we evolved 76 parallel populations of non-resistant WT *S. cerevisiae* strains in the presence of 64 μg/mL of fluconazole. This concentration was specifically selected to represent a moderate selection pressure (**Fig. 5A**) (for more details, see **Methods**).”*

12) Lines 349-353: 25% of evolved clones (i.e. 19 of the 77 clones?) showed increased caspofungin resistance, of which 57.8% (i.e. 11 of those 19 clones?) exhibited decreased resistance upon GIM3 deletion. How many of these showed decreased fitness also in the absence of selection (30.2%, i.e. 23 of the 77 clones (?), see lines 329-331)? In such clones the apparent importance of Gim3 for caspofungin resistance (tolerance?) might not be true. The

same applies to the 43.4% of evolved clones that lost their fluconazole tolerance upon GIM3 deletion (lines 322-325). I did not understand how many of the 77 evolved clones can yield the stated percentages of 57.8%, 30.2%, and 43.4%.

*The unevolved wild-type control strain was mistakenly included in the dataset as if it were one of the evolved clones. We have now corrected this throughout the analysis. The correct total number of evolved clones is 76, not 77. Consequently, the actual number of *gim3Δ* derivatives tested was 76, and all percentages reported under the different drug conditions were calculated using this denominator.*

Among these 76 clones, 19/76 (25.0%) displayed larger colony size on caspofungin than the unevolved control WT strain. Of those 19 clones, 11/19 (57.9%) show decreased colony size on caspofungin after GIM3 deletion. Notably, 7/11 (63.6%) of these 11 clones also showed a significantly smaller colony size on caspofungin-free medium. These results indicate that for some clones the observed reduced growth in caspofungin could (partially) reflect a general fitness defect.

Across all 76 tested clones, 33/76 (43.4%) lost fluconazole tolerance after GIM3 deletion. Of these 33, 11/33 (33.3) % also exhibited reduced fitness in drug-free medium.

We have now incorporated these explicit counts and percentages in the Results section (lines 360–366) to remove any ambiguity.

“Our results show that 25% of the evolved clones (19/76) displayed larger colony size on caspofungin than the unevolved control WT strain (orange, grey, and green dots above the red dot in Fig. 5F). Of those 19 clones, 11/19 (57.9%) show decreased colony size on caspofungin after GIM3 deletion (orange lines above the red dot in Fig. 5F). Notably, 7 of these 11 clones (63.6%) also showed a significantly smaller colony size on caspofungin-free medium. These results indicate that for some clones the observed reduced growth in caspofungin could (partially) reflect a general fitness defect.”

Raw data on colony size measurement can be found in **Extended Dataset 3**.

Referee #2:

The authors addressed all my concerns and have significantly strengthened the ms. The ms is suitable to be published in EMBO reports in its current format in my opinion

We sincerely thank Referee #2 for their thoughtful review and constructive feedback. We are glad to hear that the revisions have addressed their concerns and that they find the manuscript suitable for publication in its current form. We greatly appreciate their time and support in helping to improve the quality of our work.

Referee #3:

The authors have addressed the comments of the referees. They fail to mention the induction of petite mutants under azole treatment although they saw no small colonies. I think it should be cited to bring this to the attention of others for both species. The additional sensitivity testing is a positive addition. It is possible a mixture exist in their colonies under treatment. They also cite Martel et al 2010 and Jackson et al 2003 claiming drug efflux changes seen in conjunction with *erg3* mutations but this is incorrect.

We now added a discussion paragraph noting that azoles can induce petite mutants:

Lines 534-546

*"Azole exposure is known to induce respiration-deficient ("petite") mutants in *S. cerevisiae* and *N. glabratus* (Zheng et al, 2025; Brun et al, 2004, 2005; Siscar-Lewin et al, 2021; Bouchara et al, 2000; Sanglard et al, 2001). Petite mutations in yeast are typically found in the mitochondrial genome, or in genes related to mitochondrial function and result in respiratory deficiency. Importantly, these mutations also result in smaller colonies on medium containing glucose, since the respiratory defect leads to substantially reduced ATP yield when growing on glucose (Day, 2013; Baruffini et al, 2007; Hess et al, 2009). Mitochondrial dysfunction activates Pleiotropic Drug Resistance (PDR) regulators, increasing efflux and modulating azole response (Ferrari et al, 2009; Tsai et al, 2006; Hallstrom & Moye-Rowley, 2000). In other words, fluconazole can not only induce petite mutants, but these petite mutants can in turn be more fluconazole tolerant. Although we did not perform a dedicated respiration assay (e.g., growth on non-fermentable carbon) in our study, all experiments were conducted on glucose-containing media, and we did not observe small-colony variants typically associated with petites."*

As for the incorrect citation of the Martel et al 2010 and Jackson et al 2003 references, we see that this confusion stems from our rather unfortunate positioning of these references in the text. We have now adapted this to make clearer what exactly is discussed in these two papers:

Lines 439-444

*"Some fluconazole-resistant strains carrying non-synonymous *ERG3* mutations have been described to maintain near-normal sterol profiles (Martel et al, 2010). These are known as *ERG3* leaky mutants, as they retain sufficient function to reduce toxic intermediate build-up while still producing some ergosterol (Jackson et al, 2003)."*

References

- Baruffini E, Lodi T, Dallabona C & Foury F (2007) A Single Nucleotide Polymorphism in the DNA Polymerase Gamma Gene of *Saccharomyces cerevisiae* Laboratory Strains Is Responsible for Increased Mitochondrial DNA Mutability. *Genetics* 177: 1227–1231
- Belle A, Tanay A, Bitincka L, Shamir R & O'Shea EK (2006) Quantification of protein half-lives in the budding yeast proteome. *Proc Natl Acad Sci U S A* 103: 13004–13009
- Bouchara JP, Zouhair R, Le Boudouil S, Renier G, Filmon R, Chabasse D, Hallet JN & Defontaine A (2000) In-vivo selection of an azole-resistant petite mutant of *Candida glabrata*. *J Med Microbiol* 49: 977–984
- Brun S, Bergés T, Poupard P, Vauzelle-Moreau C, Renier G, Chabasse D & Bouchara JP (2004) Mechanisms of Azole Resistance in Petite Mutants of *Candida glabrata*. *Antimicrob Agents Chemother* 48: 1788
- Brun S, Dalle F, Saulnier P, Renier G, Bonnin A, Chabasse D & Bouchara JP (2005) Biological consequences of petite mutations in *Candida glabrata*. *J Antimicrob Chemother* 56: 307–314
- Christiano R, Nagaraj N, Fröhlich F & Walther TC (2014) Global Proteome Turnover Analyses of the Yeasts *S. cerevisiae* and *S. pombe*. *Cell Rep* 9: 1959–1965
- Day M (2013) Yeast Petites and Small Colony Variants: For Everything There Is a Season. *Adv Appl Microbiol* 85: 1–41
- Ferrari S, Ischer F, Calabrese D, Posteraro B, Sanguinetti M, Fadda G, Rohde B, Bauser C, Bader O & Sanglard D (2009) Gain of Function Mutations in CgPDR1 of *Candida glabrata* Not Only Mediate Antifungal Resistance but Also Enhance Virulence. *PLoS Pathog* 5: e1000268
- Guan XL, Souza CM, Pichler H, Dewhurst G, Schaad O, Kajiwarra K, Wakabayashi H, Ivanova T, Castillon GA, Piccolis M, *et al* (2009) Functional interactions between sphingolipids and sterols in biological membranes regulating cell physiology. *Mol Biol Cell* 20: 2083–2095
- Hallstrom TC & Moye-Rowley WS (2000) Multiple signals from dysfunctional mitochondria activate the pleiotropic drug resistance pathway in *Saccharomyces cerevisiae*. *Journal of Biological Chemistry* 275: 37347–37356
- Heese-Peck A, Pichler H, Zanolari B, Watanabe R, Daum G & Riezman H (2002) Multiple Functions of Sterols in Yeast Endocytosis. *Mol Biol Cell* 13: 2664
- Hess DC, Myers C, Huttenhower C, Hibbs MA, Hayes AP, Paw J, Clore JJ, Mendoza RM, Luis BS, Nislow C, *et al* (2009) Computationally Driven, Quantitative Experiments Discover Genes Required for Mitochondrial Biogenesis. *PLoS Genet* 5: e1000407
- Jackson CJ, Lamb DC, Manning NJ, Kelly DE & Kelly SL (2003) Mutations in *Saccharomyces cerevisiae* sterol C5-desaturase conferring resistance to the CYP51 inhibitor fluconazole. *Biochem Biophys Res Commun* 309: 999–1004

- Martel CM, Parker JE, Bader O, Weig M, Gross U, Warrilow AGS, Rolley N, Kelly DE & Kelly SL (2010) Identification and Characterization of Four Azole-Resistant *erg3* Mutants of *Candida albicans*. *Antimicrob Agents Chemother* 54: 4527
- Sanglard D, Ischer F & Bille J (2001) Role of ATP-binding-cassette transporter genes in high-frequency acquisition of resistance to azole antifungals in *Candida glabrata*. *Antimicrob Agents Chemother* 45: 1174–1183
- Siscar-Lewin S, Gabaldón T, Aldejohann AM, Kurzai O, Hube B & Brunke S (2021) Transient mitochondria dysfunction confers fungal cross-resistance against phagocytic killing and fluconazole. *mBio* 12
- Tsai HF, Krol AA, Sarti KE & Bennett JE (2006) *Candida glabrata* PDR1, a Transcriptional Regulator of a Pleiotropic Drug Resistance Network, Mediates Azole Resistance in Clinical Isolates and Petite Mutants. *Antimicrob Agents Chemother* 50: 1384–1392
- Zheng L, Dong Y, Wang J, Jia Y, Wang W, Xu Y & Guo L (2025) Understanding adaptation to fluconazole: comparative insights into tolerance and resistance in *Saccharomyces cerevisiae* and *Candida albicans*. *Front Cell Infect Microbiol* 15: 1519323

Prof. Kevin Verstrepen
VIB-KU Leuven
Belgium

Dear Prof. Verstrepen,

I am very pleased to accept your manuscript for publication in EMBO Reports. Thank you for your contribution to our journal.

You may qualify for financial assistance for your publication charges - either via a Springer Nature fully open access agreement or an EMBO initiative. Check your eligibility: <https://link.springer.com/journal/44319/how-to-publish-with-us>

Yours sincerely,

Yehu Moran
Academic Editor
EMBO Reports

>>> Please note that it is EMBO Reports policy for the transcript of the editorial process (containing referee reports and your response letter) to be published as an online supplement to each paper. If you do NOT want this, you will need to inform the Editorial Office via email immediately. More information is available here: <https://link.springer.com/partners/embo-press/editorial-policies#Peer%20review>